# TRACTABLE MULTI-AGENT REINFORCEMENT LEARNING THROUGH BEHAVIORAL ECONOMICS

**Eric Mazumdar**[*]**, Kishan Panaganti**[*]**, & Laixi Shi**[*]
Department of Computing and Mathematical Sciences
California Insitute of Technology
Pasadena, CA, USA
`{mazumdar,kpb,laixis}@caltech.edu`

## ABSTRACT

A significant roadblock to the development of principled multi-agent reinforcement learning (MARL) algorithms is the fact that desired solution concepts like Nash equilibria may be intractable to compute. We show how one can overcome this obstacle by introducing concepts from behavioral economics into MARL. To do so, we imbue agents with two key features of human decision-making: risk aversion and bounded rationality. We show that introducing these two properties into games gives rise to a class of equilibria—risk-averse quantal response equilibria (RQE)—which are tractable to compute in *all* $n$-player matrix and finite-horizon Markov games. In particular, we show that they emerge as the endpoint of no-regret learning in suitably adjusted versions of the games. Crucially, the class of computationally tractable RQE is independent of the underlying game structure and only depends on agents' degrees of risk-aversion and bounded rationality. To validate the expressivity of this class of solution concepts we show that it captures peoples' patterns of play in a number of 2-player matrix games previously studied in experimental economics. Furthermore, we give a first analysis of the sample complexity of computing these equilibria in finite-horizon Markov games when one has access to a generative model. We validate our findings on a simple multi-agent reinforcement learning benchmark. Our results open the doors for to the development of new decentralized multi-agent reinforcement learning algorithms.

## 1 INTRODUCTION

Machine learning algorithms are increasingly being deployed in dynamic environments in which they interact with other agents like people or other algorithms. Often, these agents have their own goals that may not be aligned with those of the algorithm—making the interactions *strategic*. These interactions are naturally modeled as *games* between rational agents and their ubiquity has driven a surge of interest in learning in games (Cesa-Bianchi and Lugosi, 2006) and multi-agent reinforcement learning (Zhang et al., 2021a) in recent years. Indeed, real-world applications of these problems range from the decentralized control of the smart grid (Mohsenian-Rad et al., 2010), autonomous driving (Kannan et al., 2017), and financial trading (Wellman and Wurman, 1998) to problems of aligning large language models (Munos et al., 2024) and agentic AI (Verma et al., 2024).

When viewed through the lens of game theory, many of these problems can be cast as problems of *equilibrium computation* under varying information structures, where the equilibrium represents a stable outcome for rational agents. The most common equilibrium concept is that of a Nash equilibrium (NE) (Nash, 1950): a solution under which no rational agent has an incentive to unilaterally seek to improve their outcome. Despite its popularity as a solution concept, computing a NE outside of highly structured games is known to be computationally intractable even for two-player matrix games (Daskalakis, 2013). Coupled with a host of negative results on their computation using gradient-based algorithms (Mertikopoulos et al., 2018; Mazumdar et al., 2020), converging to Nash is increasingly viewed as an unreasonable goal for decentralized reinforcement learning algorithms.

---

[*]Alphabetical order.

While relaxations of NE like (coarse) correlated equilibria (CCE) are known to be more tractable to compute—and thus a more attainable goal for learning algorithms— they also have their limitations. Indeed, while CCE arise out of the use of no-regret learning algorithms (Cesa-Bianchi and Lugosi, 2006), the set of CCE can be large (exacerbating the problem of equilibrium selection that arises with NE) and may have support on strictly dominated strategies (Viossat and Zapechelnyuk, 2013), which means that they cannot necessarily be rationalized by individual agents in isolation (Dekel and Fudenberg, 1990). Furthermore, a dynamic versions of CCE—stationary Markov CCE—can also be intractable to compute in general-sum Markov games (Daskalakis et al., 2023b).

Beyond these hardness results, solution concepts like NE and CCE also fail to be predictive of what strategies people play in games (McKelvey and Palfrey, 1995; Erev and Roth, 1998), with people being observed to be imperfect optimizers (Goeree and Holt, 1999; Capra et al., 2002) and risk-averse (Goeree et al., 2003) when confronted with game theoretic scenarios. This aligns with celebrated work in behavioral economics and mathematical psychology which has repeatedly shown that dominant features of human decision-making are a failure to perfectly optimize (Luce, 1959) and risk-aversion (Kahneman and Tversky, 1979; Tversky and Kahneman, 1992).

The first observation is often referred to as *bounded rationality* which posits that individuals are naturally prone to making mistakes and often fail to be perfectly optimal (Luce, 1959). This is often captured in games through the idea of a *quantal response equilibrium* (QRE) (McKelvey and Palfrey, 1995). The second observation can be attributed to the fact that players typically face uncertainty and risk in their decisions. These arise from environmental uncertainties like unknown future events, noise, or even the mere presence of other players. This can lead people to prefer risk-averse strategies, i.e., strategies which give more certain outcomes at the cost of lower expected returns (Gollier, 2001). Interestingly, there is experimental evidence that neither of these properties alone can account for people's patterns of play observed in controlled experiments (Goeree et al., 2003; Goeree and Offerman, 2002), and that models of decision-making that incorporate *both* of these features have the best predictive power (Goeree et al., 2003).

**Contributions:** Motivated by these findings, in this paper we study games in which players are risk-averse and have bounded rationality and study the computational properties of the natural equilibrium concept: a risk-averse quantal response equilibrium (RQE). At first glance, the introduction of these features of human decision-making into games introduces non-linearities that break existing game-theoretic structures. Indeed we show how neither of these features alone give rise to a single class of equilibria that can efficiently computed across all games. However, by relying on dual formulations of risk we show how—for a large range of degrees of risk-aversion and bounded rationality—a class of RQE are computationally tractable in *all* n-player matrix and finite horizon Markov games. Importantly, these conditions are *independent* of the underlying game and only depend on the class of risk metrics and quantal responses under consideration. Thus, RQE not only capture important features of human decision-making but are also more amenable to computation than QRE or NE in matrix and finite-horizon Markov games. This opens the door for the development of theoretically principled decentralized algorithms for multi-agent reinforcement learning (MARL) centered around RQE as opposed to CCE or NE. We refer readers to the discussion of related works in Appendix A for a more in-depth discussion of different equilibria concepts in the context for our work.

To emphasize the practical relevance of this theoretical result we show how the regime for which RQE are computationally tractable captures real-world data from behavioral economics on people's observed pattern of play in 13 different games. This is illustrated in Fig. 1, where the blue region represents the set of RQE that are computationally tractable which is a function of agents' degree of risk-aversion ($\tau_1/\tau_2$), and level of bounded rationality ($\epsilon_1/\epsilon_2$).

Altogether, our results show that imbuing artificial agents with features of human decision-making from behavioral economics gives rise to an expressive class of equilibria in games. Crucially, this new solution concept appears to overcome many of the computational limitations of existing concepts and thus appears to be a promising foundation for the development of principled MARL algorithms.

**Notations:** We use $\Delta_n$ to denote probability simplex of size $n$. In addition, we denote $[N] = \{1, 2, \cdots, N\}$ for any positive integer $N > 0$. We denote $x = \big[x(s,a)\big]_{(s,a)\in\mathcal{S}\times\mathcal{A}} \in \mathbb{R}^{SA}$ (resp. $x = \big[x(s)\big]_{s\in\mathcal{S}} \in \mathbb{R}^S$) as any vector that constitutes certain values for each state-action pair (resp. state).

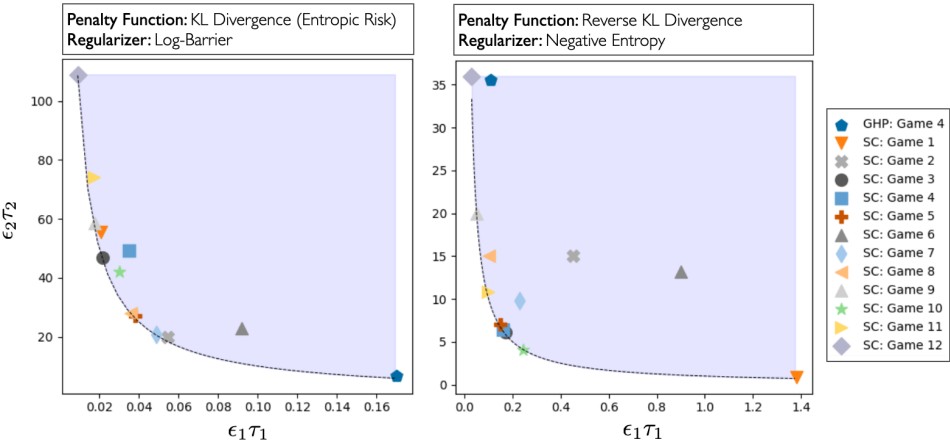

Figure 1: The shaded blue region depicts the regime of risk-aversion and bounded rationality preferences that allow for computationally tractable RQE in all 2-player games as shown in Theorem 3. The markers *GHP: Game 4* (Goeree et al., 2003) and *SC: Game 1-12* (Selten and Chmura, 2008) represent the necessary parameter values required to recreate the average strategy played by people in various 2-player games in observational data up to 1% accuracy.

## 2   RISK-AVERSION AND BOUNDED RATIONALITY IN GAMES

To begin, we focus on $n$-player general-sum finite-action games. In these games, each player $i$ has access to a (finite) action set $\mathcal{A}_i$ with $A_i = |\mathcal{A}_i|$ pure strategies. For each tuple of joint strategies $a = (a_1, ..., a_n) \in \mathcal{A} := \prod_{i=1}^{n} \mathcal{A}_i$ each player has an associated reward or utility $R_i(a)$. As is common in the study of these games, we consider the case where players play over mixed strategies and seek to maximize their expected utility. Player $i$'s expected utility in this case can be written as a function $U_i : \mathcal{P} \to \mathbb{R}$ which can be written as:

$$U_i(\pi_1, ..., \pi_n) = \mathbb{E}_{a \sim \pi}[R_i(a)] \tag{1}$$

where $\pi = (\pi_1, ..., \pi_n) \in \mathcal{P} := \prod_{i=1}^{n} \Delta_{A_i}$ is the joint strategy of all the players. For ease of exposition we often write the utility as $U_i(\pi_i, \pi_{-i})$ where $\pi_{-i} = (\pi_1, ..., \pi_{i-1}, \pi_{i+1}, ..., \pi_n)$ represents the joint strategies of all players *other* than $i$. A natural outcome in this class of games is the notion of a (mixed) Nash equilibrium.

**Definition 1** (Nash Equilibrium). *A (mixed) Nash equilibrium of a $n$-player general-sum finite-action game is a joint strategy $\pi^* = (\pi_1^*, ..., \pi_n^*) \in \mathcal{P}$ such that no player has any incentive to unilaterally deviate—i.e., for all $i = 1, ..., n$: $U_i(\pi_i^*, \pi_{-i}^*) \geq U_i(\pi_i, \pi_{-i}^*) \quad \forall \pi_i \in \Delta_{A_i}$.*

As introduced, $n$-player general-sum finite-action games are well known to admit at least one Nash equilibrium (NE) in mixed strategies (Nash, 1951). While NE are known to always the exist in mixed strategies in the games we consider, they have also been shown to be intractable to compute (Daskalakis, 2013). Moreover, NE may not realistically predict human decision-making behavior. Indeed a preponderance of empirical evidence suggests that people do *not* play their Nash strategies and in fact are potentially bounded rational and risk-averse in their decision-making—inducing new forms of equilibria (see e.g., Luce (1959); Tversky and Kahneman (1992); Goeree et al. (2003); McKelvey and Palfrey (1995)). To formally introduce these features of human decision-making into the problem, we construct generalizations of the expected utility game that allow us to model agents as either imperfect optimizers and risk-averse.

### 2.1   BOUNDED RATIONALITY IN GAMES

In contrast to risk-aversion, which has been under-explored in game theory and multi-agent reinforcement learning, bounded rationality is more common. Many works studying the computational benefits of incorporating it into games (Sokota et al., 2023; Mertikopoulos and Sandholm, 2016; Cen et al., 2021; Leonardos et al., 2021; Evans and Ganesh, 2024; Jacob et al., 2022), with the most common form of bounded rationality found in the literature on learning in games being that of a quantal response. These capture bounded rationality by either assuming that the players are rational in a stochastically perturbed version of the game or equivalently that players' strategies are constrained to the set of *quantal response functions* (McKelvey and Palfrey, 1995; 1998).

**Definition 2.** *(Quantal Response Function) A* quantal response function *is a continuous function* $\sigma : \mathbb{R}^n \to \Delta_n$ *such that for any* $x \in \mathbb{R}^n$: *If* $x_i < x_j$, $\sigma_i(x) > \sigma_j(x)$, *where* $x_k, \sigma_k(x)$ *represent the* $k$-*th components of* $x$ *and* $\sigma$, *respectively.*

These functions restrict players' strategies to subsets of the simplex—effectively smoothing their best responses and preventing them from being complete utility maximizers. One way to restrict a player's strategy to quantal responses is by regularizing their objective with a suitable strongly convex regularizer $\nu_i$ (see, e.g., Föllmer and Schied (2002, Proposition 7), or Sokota et al. (2023); Mertikopoulos and Sandholm (2016)). This yields adjusted utility functions of the form:

$$U_i^{\epsilon_i}(\pi_1, ..., \pi_n) = \mathbb{E}_{a \sim \pi}[R_i(a)] - \epsilon_i \nu(\pi_i), \tag{2}$$

where $\epsilon_i \geq 0$ captures player $i$'s degree of bounded rationality. As $\epsilon_i$ increases their strategy space becomes more constrained, yielding more boundedly rational strategies. When all players achieve a Nash equilibrium in the space of their quantal responses—the class of strategies that can be represented by a fixed class of quantal response functions (cf. Definition 2)—the resulting equilibrium is known as a quantal response equilibrium (QRE).

Despite the many works that focus on computing QRE, to the best of our knowledge there are no classes of QRE that are universally computable across all games. Indeed most works focus on zero-sum or approximately zero-sum games (Sokota et al., 2023; Mertikopoulos and Sandholm, 2016; Leonardos et al., 2021). In more general classes of games the class of QRE or equivalently the level of bounded rationality needed for computational tractability depends on the underlying game structure (e.g., the size of player's action spaces and the magnitude of their rewards) which may not be known a priori (Sun et al., 2024). Furthermore, we note that more work in behavioral economics has consistently highlighted that bounded rationality on its own is not enough to capture the nuances of human decision-making, even for simple games such as matching pennies (Goeree et al., 2003; Tversky and Kahneman, 1992). These findings motivate us to introduce risk-aversion into games.

## 2.2 RISK-AVERSION IN GAMES

To allow agents to have risk preferences we make use of a general class of convex risk metrics from mathematical finance and operations research (Föllmer and Schied, 2002). In this framing we move into a regime where agents seek to *minimize* a measure of risk.

**Definition 3** (Convex Risk Measures). *Let* $\mathcal{X}$ *be the set of functions mapping from a space of outcomes* $\Omega$ *to* $\mathbb{R}$. *A convex measure of risk is a mapping* $\rho : \mathcal{X} \to \mathbb{R}$ *satisfying:*
*1. Monotonicity: If* $X \leq Y$ *almost surely, then* $\rho(X) \geq \rho(Y)$.
*2. Translation Invariance: If* $m \in \mathbb{R}$ *then* $\rho(X + m) = \rho(X) - m$.
*3. Convexity: For all* $\lambda \in (0, 1)$, $\rho(\lambda X + (1 - \lambda)Y) \leq \lambda \rho(X) + (1 - \lambda)\rho(Y)$.

Typically, and as we assume in the remainder of the paper, the set $\mathcal{X}$ is the set of measurable functions defined on a probability space $(\Omega, \mathcal{F}, P)$. Under this assumption, the convex measures of risk allow us to generalize expectations to trade off high variance for and higher utilities. For ease of exposition, we write $\rho_\pi(X)$ to highlight the distribution that the convex risk measure is taken with respect to $\pi$.

Given these definitions, we present two ways of incorporating risk-aversion into games.

***Remark*** 1. A crucial feature of our formulation of risk-aversion is that players are *not* risk-averse to the randomness introduced by their own strategy and only to the uncertainty caused by the environment and their opponents. This is a common approach taken (often implicitly) in the literature on single-agent risk-sensitive and robust decision-making (Shen et al., 2014). It also appears to be necessary to ensure the existence of equilibria introduced in Theorem 2 shortly; otherwise as studied in Fiat and Papadimitriou (2010), equilibria may not exist.

To see why our framing of risk-aversion is natural we introduce the first form of risk that we consider: *action-dependent* risk-aversion.

**Action-dependent Risk Aversion:** In this framing, a player first evaluates the risk associated with each of their pure strategies $a_i$: $\rho_{i, a_{-i} \sim \pi_{-i}}(R(a_i, a_{-i}))$. The player then minimizes their expected risk. We capture this by transforming the player's utilities in (1) into costs $f_i^{act}$ of the form:

$$f_i^{act}(\pi_i, \pi_{-i}) = \mathbb{E}_{\pi_i}\left[\rho_{i, \pi_{-i}}(R_i(a))\right] = \sum_{a_i \in \mathcal{A}_i} \pi_i(a_i) \rho_{i, \pi_{-i}}(R_i(a_i, a_{-i})), \tag{3}$$

where $\rho_{i,\pi_{-i}}$ is used to capture agent $i$'s risk preference which depends on the product distribution of opponents strategies $\pi_{-i}$.

Under action-dependent risk-aversion a player uses mixed strategies only when two pure strategies yield the same level of risk. Thus introducing additional risk-aversion to $\pi_i$ does not change outcomes.

**Aggregate Risk Aversion:** The second form of risk-aversion we consider, *aggregate* risk-aversion is more conservative, but allows for a simpler analysis. To capture aggregate risk aversion, we transform the player's utilities in (1) into costs $f_i^{agg}$ which take the form

$$f_i^{agg}(\pi_i, \pi_{-i}) = \rho_{i,\pi_{-i}}(\mathbb{E}_{\pi_i}[R_i(a)]) = \rho_{i,\pi_{-i}}\left(\sum_{a_i \in \mathcal{A}_i} \pi_i(a_i)R_i(a_i, a_{-i})\right). \tag{4}$$

Aggregate risk-aversion captures the fact that risks may be correlated across pure strategies, and is related (due to convexity of $\rho_i$) by Jensen's inequality to action-dependent risk-aversion such that $f_i^{agg}(\pi) < f_i^{act}(\pi)$. Interestingly, recent work in behavioral economics has shown that this may be a better model of risk-aversion in people (Oprea and Robalino, 2024).

We note that if $\rho_i(X) = \mathbb{E}[-X]$ for all players $i = 1, ..., n$ (which satisfies the requirements in Definition 3), then the new formulations reduce to the original expected utility objective. Thus, the class of risk-averse games can be seen as generalizations of the classic expected utility games (1).

***Remark*** 2. For brevity, as both generalizations have similar implications, we provide details and results of action-dependent risk aversion in our supplementary material and focus on aggregate risk aversion for the remainder of the paper. Thus we denote $f_i = f_i^{agg}$ for the remainder of the paper and analyze $f_i = f_i^{act}$ in the supplementary material.

While at first glance the modified game looks significantly more complex than the previous expected utility maximization setup (cf. (1)), we can rely on a particularly powerful property of convex measures of risk to simplify and expose some structure in this class of problems.

**Theorem 1** (Dual Representation Theorem for Convex Risk Measures (Föllmer and Schied, 2002)). *Suppose that the set $\mathcal{X}$ is the set of functions mapping from a finite set $\Omega$ to $\mathbb{R}$. Then a mapping $\rho : \mathcal{X} \to \mathbb{R}$ is a convex risk measure (cf. Definition 3) if and only if there exists a penalty function $D : \Delta_\Omega \to (-\infty, \infty]$ such that: $\rho(X) = \sup_{p \in \Delta_\Omega} E_p[-X] - D(p)$, where $\Delta_\Omega$ is the set of all probability measures on $\Omega$. Furthermore, the function $D(p)$ can be taken to be convex, lower-semi-continuous, and satisfy $D(p) > -\rho(0)$ for all $p \in \Delta_\Omega$.*

Throughout, we make a simplifying assumption that $D$ is continuous in both arguments, which is satisfied by various widely used risk measures Table 1. We provide more details on penalty functions in Appendix B. Given this result, we derive the aggregate risk-averse game (4) in the following form:

$$f_i(\pi_i, \pi_{-i}) = \sup_{p_i \in \mathcal{P}_{-i}} -\pi_i^T R_i p_i - \frac{1}{\tau_i} D_i(p_i, \pi_{-i}) \ \forall i = 1, .., n, \tag{5}$$

where $\mathcal{P}_{-i} = \mathcal{P}/\Delta_{A_i} \subset \mathbb{R}^{A_{-i}}$, $A_{-i} = \prod_{j \neq i} A_j$, and $R_i \in \mathbb{R}^{A_i \times A_{-i}}$ is player $i$'s payoff matrix.

We note that we differentiate the penalty functions $D_i$ to allow agents to have different risk preferences in different risk metrics. The parameter $\tau_i$ captures a player's degree of risk-aversion. In this form, one can see that in a risk-averse game, the players imagine that intermediate adversaries seek to maximize their cost but are penalized from deviating too far from the opponents' realized strategies. As $\tau_i \to \infty$ they become increasingly risk-averse—and in the extreme, treat their opponents and environment as adversarial.

Since a NE of the risk-averse game can be qualitatively different from that of the original game, we now define a risk-averse Nash equilibrium (RNE). For brevity, we do not introduce aggregate and action-dependent RNE separately, since similar results hold for either formulation.

**Definition 4.** *(Risk-Averse Nash Equilibrium) A risk-averse Nash equilibrium (RNE) of is joint strategy $\pi^* \in \mathcal{P}$ such that no player has any incentive to unilaterally deviate in the risk adjusted game—i.e., for all $i = 1, ..., n$: $f_i(\pi_i^*, \pi_{-i}^*) \leq f_i(\pi_i, \pi_{-i}^*) \ \forall \ \pi_i \in \Delta_{A_i}$*

Note that since players would like to *minimize* risk, the direction of the inequality has changed. The convexity and continuity of the penalty function guarantees that the risk-averse games admit at least one RNE. A general statement and proof are provided in Theorem 7.

**Theorem 2.** *All aggregate risk-averse games (5) admit at least one RNE.*

The existence of a RNE is a consequence of the fact that under our risk formulations, players are not risk averse to the randomness of their own by playing a mixed strategy. This is in stark contrast to previous works which consider games in which players are risk-averse to all randomness (including their own), and in which a RNE may not exist (Fiat and Papadimitriou, 2010). Thus, our specific risk formulations are crucial to our later results.

Finally, we note that the additional convexity induced by the introduction of risk aversion guarantee is not enough to ensure the computational tractability of the risk-averse Nash equilibrium (see e.g., (McMahan et al., 2024)). This is not surprising given the fact that computing NE in general convex games can be intractable in general.

## 3 RISK-AVERSE QUANTAL RESPONSE EQUILIBRIA IN MATRIX GAMES

Since neither bounded rationality nor risk-aversion alone guarantees computational tractability of their associated solution concepts, we consider the equilibrium concept that arises out of their combination and show that it can often be efficiently computed. Interestingly, this echoes work in behavioral economics in which the combination of risk aversion and bounded rationality has also been found to be a better predictor of human play (Goeree et al., 2003) than either of the properties alone.

Towards this goal, in the final form of the risk-adjusted game we consider, players' costs are given by:

$$f_i^{\epsilon_i}(\pi_i, \pi_{-i}) = f_i(\pi_i, \pi_{-i}) + \epsilon_i \nu_i(\pi_i) \ \ \forall i = 1, .., n, \tag{6}$$

where $\nu_i$ is a strictly convex regularizer which gives rise to a set of quantal responses. The natural outcome of this game is what we term as a risk-averse quantal-response equilibrium (RQE).

**Definition 5** (Risk-Averse Quantal Response Equilibrium (RQE)). *A risk-averse quantal response equilibrium of a $n$-player general-sum finite-action game is a joint strategy $\pi^* \in \mathcal{P}$ such that for each player $i = 1, ..., n$:*

$$f_i^{\epsilon_i}(\pi_i^*, \pi_{-i}^*) \leq f_i^{\epsilon_i}(\pi_i, \pi_{-i}^*) \ \ \forall \ \pi_i \in \Delta_{A_i}.$$

For any set of convex regularizers $\nu_i$, it is easy to observe that the game remains a convex game and thus RQE (as defined) will always exist in all matrix games.

### 3.1 CONDITIONS FOR THE COMPUTATIONAL TRACTABILITY OF RQE

Given our definition of RQE, we now demonstrate that if players' risk preferences (i.e., their degree of risk aversion) and families of quantal response functions (i.e., their bounded rationality parameters) satisfy a simple relationship, the game admits a RQE that can be efficiently computed using arbitrary no-regret learning algorithms (e.g., gradient-play or mirror descent) in a decentralized manner. Importantly this result is *independent* of the underlying payoffs $\{R_i\}_{i=1}^n$.

To derive our results on the computational tractability of RQE under aggregate risk-aversion, we first introduce a related $2n$-player game in which we associate to each original player $i$ an adversary whose strategy is denoted as $p_i \in \mathcal{P}_{-i}$. Let $p = (p_1, ..., p_n) \in \bar{\mathcal{P}} = \prod_{i=1}^n \mathcal{P}_{-i}$. In this new game, each original player's loss function takes the form:

$$J_i(\pi_i, \pi_{-i}, p) = -\pi_i^T R_i p_i - \frac{1}{\tau_i} D_i(p_i, \pi_{-i}) + \epsilon_i \nu_i(\pi_i). \tag{7}$$

For each player $p_i$ we associate them to a new loss function which we denote:

$$\bar{J}_i(\pi, p_i, p_{-i}) = \pi_i^T R_i p_i + \frac{1}{\tau_i} D_i(p_i, \pi_{-i}) - \sum_{j \neq i} \xi_{i,j} \nu_j(\pi_j). \tag{8}$$

In this $2n$ player game, the $i$-th player's adversary, whose strategy is $p_i$ seeks to minimize their loss $\bar{J}_i$ which is strategically the same as maximizing $J_i$.

Given the definition of the $2n$-player game we show how no-regret learning algorithms can be used to compute a RQE. To prove this, we first define coarse correlated equilibria (CCE).

**Definition 6.** *A coarse correlated equilibrium of the $2n$-player game is a probability measure $\sigma$ on $\mathcal{P} \times \bar{\mathcal{P}}$ such that for all $i = 1, ..., n$:*

$$\mathbb{E}_{(\pi, p) \sim \sigma}[J_i(\pi, p)] \leq \mathbb{E}_{(\pi_{-i}, p) \sim \sigma}[J_i(\pi_i', \pi_{-i}, p)] \quad \forall \pi_i' \in \Delta_{A_i}$$
$$\mathbb{E}_{(\pi, p) \sim \sigma}[\bar{J}_i(\pi, p)] \leq \mathbb{E}_{(\pi, p_{-i}) \sim \sigma}[\bar{J}_i(\pi, p_i', p_{-i})] \quad \forall p_i' \in \mathcal{P}_{-i}.$$

CCE are the natural outcome of no-regret learning algorithms, and we show that CCEs of the $2n$-player game coincide with RQE of the original $n$-player matrix game. This is a phenomenon known as equilibrium collapse which is well known to happen in zero-sum games and certain generalizations of zero-sum games (Cai et al., 2016; Kalogiannis and Panageas, 2023). For brevity we present our results on 2-player matrix games now and extend the results to $n$-player setting in Appendix C.3.

To simplify our results we define $H_1(p_1, \pi_2) = \frac{1}{\tau_1} D_1(p_1, \pi_2) - \xi_1 \nu_2(\pi_2)$ and $H_2(p_2, \pi_1) = \frac{1}{\tau_2} D_2(p_2, \pi_1) - \xi_2 \nu_1(\pi_1)$. Let $\xi_1^* > 0$ and $\xi_2^* > 0$ be the smallest values of $\xi_1$ and $\xi_2$ such that $H_1(p_1, \pi_2)$ and $H_2(p_2, \pi_1)$ are concave in $\pi_2$ and $\pi_1$ for all $p_1$ and $p_2$ respectively. These parameters capture the player's relative degrees of risk aversion, and as we show, for certain instantiations of risk measures and quantal response functions $\xi_i^* = \frac{1}{\tau_i}$ for $i = 1, 2$. The following theorem gives conditions on $\xi_1^*, \xi_2^*$ and $\epsilon_1, \epsilon_2$, under which RQEs can be computed through no-regret learning. The proof is postponed to Appendix C.2.

**Theorem 3.** *Assume the penalty functions that give rise to the players' risk preferences $D_1(\cdot, \cdot)$ and $D_2(\cdot, \cdot)$ are jointly convex in both their arguments. If $\sigma$ is a CCE of the four player game with $\xi_{1,2} = \xi_1^*$ and $\xi_{2,1} = \xi_2^*$, and $\epsilon_1 \epsilon_2 \geq \xi_1^* \xi_2^*$, then $\hat{\pi}_1 = \mathbb{E}_\sigma[\pi_1]$ and $\hat{\pi}_2 = \mathbb{E}_\sigma[\pi_2]$ constitute an RQE of the risk-averse game.*

This theorem gives a range of risk aversion and bounded rationality parameters under which an RQE can be computationally tractable using no-regret learning. This range is independent of the structure in the underlying game—making the class of tractably computable equilibria universal to all games. In the case where $\xi_i^* = \frac{1}{\tau_i}$, we observe that for any $\epsilon_1, \epsilon_2 > 0$, as $\tau_i \to \infty$, the game becomes solvable. This captures the fact that for any degree of bounded rationality one can compute a players' security strategy using no-regret learning—recovering existing results on the computational tractability of min-max strategies.

To validate this result, we show that the class of computationally tractable RQE is sufficiently rich to capture the aggregate strategies played by people in matrix games studied in laboratory experiments in behavioral economics to showcase how human behavior can differ from Nash equilibrium strategies (Selfridge, 1989; Goeree et al., 2003; Selten and Chmura, 2008). In particular, we showcase in Fig. 1 that for some choice of parameters ($\epsilon_j$ and $\tau_j$'s) satisfying conditions in Theorem 3, we recover human behaviors in various matching pennies games investigated in Tables 2 and 3. We provide more details about the games and experiments in Appendix F.1.

## 4 RISK-AVERSE QUANTAL RESPONSE EQUILIBRIA IN MARKOV GAMES

In this section, we extend our results to finite-horizon Markov games, or stochastic games (Shapley, 1953) which allow us to model dynamic games played out over Markov decision processes (MDP).

A Markov game can be seen as a sequential matrix game with stochastic dynamics. In this paper, we consider general-sum finite-horizon Markov games involving $n$ players, represented as $\mathcal{MG} = \{H, \mathcal{S}, \{\mathcal{A}_i\}_{i \in [n]}, \{R_{i,h}, P_{i,h}\}_{i \in [n], h \in [H]}\}$. Here, $H$ is the time horizon length of the Markov game and $\mathcal{S} = \{1, 2, \cdots, S\}$ represents the state space of the underlying MDP with size $S$. We also adopt the same notation as in Section 2, and let $\{\mathcal{A}_i\}_{i \in [n]}$ represent the action spaces of each player, each with cardinality $|\mathcal{A}_i| = A_i$. Similarly, the joint action space is given by $\mathcal{A}$, a joint action profile is $a \in \mathcal{A}$, and $\mathcal{P}$ is the product policy space. Similarly, $R_{i,h}$ represents the utility or reward function of the $i$-th player at time step $h$ for any $(i, h) \in [n] \times [H]$ and $R_{i,h}(s, \boldsymbol{a})$ represents the immediate reward (or utility) received by the $i$-th player given state-action pair $(s, \boldsymbol{a})$. For simplicity, we assume $R_{i,h}$ is deterministic. Lastly, the dynamics are captured by transition kernels $P_{i,h} : \mathcal{S} \times \mathcal{A} \mapsto \Delta_S$ where $P_{i,h}(s' \mid s, \boldsymbol{a})$ which capture probability transitioning from current state $s$ to the next state $s'$ conditioned on the joint action $\boldsymbol{a}$.

**Markov policies and value functions.** In the classical setup of finite-horizon Markov games, players are assumed to play over the space of Markov policies, where at any state $s$ at any time

step $h \in [H]$, the action selection rule depends only on the current state $s$, and is independent of previous trajectories and other players. Specifically, the $i$-th player or agent executes actions according to a policy $\pi_i = \{\pi_{i,h} : \mathcal{S} \mapsto \Delta_{A_i}\}_{1 \le h \le H}$, with the probability of selecting action $a$ in state $s$ at time step $h$ given by $\pi_{i,h}(a \,|\, s)$. Since we are operating in a finite horizon regime, it is natural to assume the policies are time-dependent. For simplicity, we define the joint policy of all agents as a product policy defined as $\pi = (\pi_1, \ldots, \pi_n) : \mathcal{S} \times [H] \mapsto \mathcal{P}$ and the joint policy space as $\Pi$. As such, the joint action profile $\boldsymbol{a}$ of all agents is drawn from distribution specified by $\pi_h(\cdot \,|\, s) = (\pi_{1,h}, \pi_{2,h} \ldots, \pi_{n,h})(\cdot \,|\, s) \in \mathcal{P}$ conditioned on state $s$ at time step $h$. For any given $\pi$, we employ $\pi_{-i} : \mathcal{S} \times [H] \mapsto \mathcal{P}_{-i}$ to represent the policies of all agents excluding the $i$-th agent.

## 4.1 Risk-Averse Markov Games

Given these definitions, we now generalize Markov games by allowing agents to be risk-averse to both the uncertainties arising from other agents' strategies and from the stochastic dynamics. This results in a new formulation of risk-averse Markov games (RAMGs). To address the two sources of risk-aversion we allow agents to have different risk preferences for their opponents and for the environment. To do so we fix two penalty functions $\{D_{\mathsf{pol},i}(\cdot, \cdot)\}_{i \in [n]}$ and $\{D_{\mathsf{env},i}(\cdot, \cdot)\}_{i \in [n]}$, which in turn give rise to two risk metrics. This allows us to define the following two functions which separately capture the risk associated with other players' actions and the environment respectively:

$$\forall i \in [n]: \quad f_{\mathsf{pol},i}^{\pi}(Q_i) = \sup_{p_i \in \mathcal{P}_i} -\pi_i^T Q_i p_i - D_{\mathsf{pol},i}(p_i, \pi_{-i}),$$

$$\forall i \in [n]: \quad f_{\mathsf{env},i}(R, P, V) = \inf_{\widetilde{P} \in \Delta_S} \left[ R + \widetilde{P}V + D_{\mathsf{env},i}(\widetilde{P}, P) \right], \tag{9}$$

for any joint policy $\pi$, where $Q_i \in \mathbb{R}^{A_i \times \sum_{j \ne i} A_j}$, $R \in \mathbb{R}^A$, $V \in \mathbb{R}^S$ will be used to represent cumulative payoff matrices and $P \in \Delta_S$, represents the dynamics of the MDP. Note that for simplicity, we choose the same $\{D_{\mathsf{pol},i}\}_{i \in [n]}$ and $\{D_{\mathsf{env},i}\}_{i \in [n]}$ for all time steps $h \in [H]$, though this can be considered to be time dependent as well.

Given the definition of these two penalty functions $f_{\mathsf{pol},i}^{\pi}(\cdot)$ and $f_{\mathsf{env},i}(\cdot)$, we can now define an agent's cumulative risk in a RAMG. For a risk-averse agent $i$ their goal is now to minimize their own long-term risk, which can be captured by a risk-averse value function $\{V_{i,h}(\pi)\}_{h \in [H]}$, where $V_{i,h}(\pi) : \mathcal{S} \mapsto \mathbb{R}$ for any joint strategy $\pi \in \Pi$. This can be defined recursively as follows: for a given joint policy $\pi$, for all $(h, s) \in [H] \times \mathcal{S}$,

$$V_{i,h}(\pi; s) = f_{\mathsf{pol},i}^{\pi} \left( Q_{i,h}(\pi; s, :) \right) \tag{10}$$

where $Q_{i,h}(\pi; s, \boldsymbol{a}) = f_{\mathsf{env},i} \left( R_{i,h}(s, \boldsymbol{a}), P_{h,s,\boldsymbol{a}}, V_{i,h+1}(\pi) \right)$ and $P_{h,s,\boldsymbol{a}} := P_h(\cdot \,|\, s, \boldsymbol{a}) \in \mathbb{R}^{1 \times S}$.

To parse this definition, we observe that $Q_{i,h}(\pi; s, :) \in \mathbb{R}^{A_i \times \sum_{j \ne i} A_j}$ represents a payoff matrix for the $i$-th player at state $s$ and time step $h$, where the value at the $a_i$-th row and the $a_{-i}$-th column is specified by $Q_{i,h}(\pi; s, \boldsymbol{a})$ with $\boldsymbol{a} = (a_i, a_{-i})$ which captures an agents' cumulative risk if joint action $\boldsymbol{a}$ is executed at state $s$ and time $h$, and the policy $\pi$ is followed subsequently. This can also be captured concisely through a recursive definition:

$$V_{i,h}(\pi; s) = f_{\mathsf{pol},i}^{\pi} \circ f_{\mathsf{env},i} \left( R_{i,h}(s, :), P_{h,s,:}, \left[ f_{\mathsf{pol},i}^{\pi} \circ f_{\mathsf{env},i} \left( R_{i,h+1}, P_{h+1}, \cdots \right) \right] \right).$$

We remark that this definition reduces to the classical setup of multi-agent reinforcement learning (Zhang et al., 2021a) when agents are risk-neutral and to the well studied setup of risk-sensitive reinforcement learning (Shen et al., 2014) when there is only one agent.

As before, we focus on the computation of risk-averse quantal response equilibria in Markov games. To do, we constrain players to quantal responses by regularizing their value functions as follows:

$$\forall (i, h, s) \in [n] \times [H] \times \mathcal{S}: \quad V_{i,h}^{\epsilon_i}(\pi; s) = V_{i,h}(\pi; s) + \epsilon_i \nu_i(\pi_i), \tag{11}$$

where $\{\epsilon_i \ge 0\}_{i \in [n]}$ capture players' degrees of bounded rationality. To define Markov RQE we denote $\pi_{i,-h} := \{\pi_{i,t}\}_{t=1,2,3,\cdots,h-1,h+1,\cdots,H}$ as agent $i$'s policy over all times other than $h$.

**Definition 7** (Markov RQE). *A product policy $\pi = \pi_1 \times \cdots \times \pi_n \in \Pi$ is said to be a risk-averse quantal response equilibrium of RAMG if:*

$$\forall (i, s, h) \in [n] \times \mathcal{S} \times [H]: \quad V_{i,h}^{\epsilon_i}(\pi; s) \le \min_{\pi_h' : \mathcal{S} \mapsto \Delta_{\mathcal{A}_i}} V_{i,h}^{\epsilon_i} \left( (\pi_h', \pi_{i,-h}) \times \pi_{-i}; s \right), \tag{12}$$

*where $\nu_i(\cdot)$ is a strictly convex function over the simplex.*

### 4.2 Computing and Approximating Markov RQE

Given our setup we now investigate how to compute Markov RQE, both with full information over the dynamics and rewards and without. For the full-information regime we propose a modified form of dynamic programming, in which we work backwards from time $h = H$ and then recursively compute the RQE until step $h = 0$. Towards this goal, at each time step $h$, for all $(i, h, s, \boldsymbol{a}) \in [n] \times [H] \times \mathcal{S} \times \mathcal{A}$, we construct the payoff matrices for the underlying for all agents at any state $s$ at time step $h$ as

$$\forall (i, s, h) \in [n] \times \mathcal{S} \times [H] : \ \widehat{Q}_{i,h}(s, \boldsymbol{a}) = R_{i,h}(s, \boldsymbol{a}) + \inf_{\widetilde{P} \in \Delta_S} \widetilde{P} \widehat{V}_{i,h+1} + D_{\mathsf{env},i}(\widetilde{P}, P_{h,s,\boldsymbol{a}}). \quad (13)$$

These payoff matrices capture both the future payoffs as well as the risk associated with the stochastic transitions in the MDP associated with a joint action $\boldsymbol{a}$. Together, they define a matrix game at state $s$ and time $h$, for which we can compute a RQE using the results presented in Section 2. We denote the routine for computing a RQE of a matrix game as $\mathsf{RQE}(\cdot)$. Proceeding recursively, we compute the RQE at time $t + 1$ and use the resulting policies to define the underlying payoff matrices at time $t$. We summarize the algorithm for computing Markov RQE in Algorithm 1 in the appendix. The following theorem guarantees that the procedure will output a Markov RQE.

**Theorem 4.** *For any RAMG $\mathcal{MG}$, assume the penalty functions that give rise to the players' risk preferences $\{D_{\mathsf{pol},i}(\cdot, \cdot)\}$ are jointly convex in both of their arguments. If, for all $(s, h) \in \mathcal{S} \times [H]$, $\epsilon_i \geq \sum_{j \neq i} \xi_{j,i}^*$ for all $(i, j) \in [n] \times [n]$, where $\xi_{i,j}^*$ is defined as in Theorem 9 for given $D_{\mathsf{pol},i}(\cdot, \cdot)$ and $\nu_j$, then the output policy $\widehat{\pi}$ from Algorithm 1 is a Markov RQE of $\mathcal{MG}$.*

A consequence of this result is that a class of Markov RQE are computationally tractable to compute via no-regret learning in finite-horizon Markov games. We remark that this is in stark contrast to both Markov Nash equilibria and Markov quantal response equilibria which cannot efficiently be computed in general-sum Markov games (Daskalakis et al., 2023b).

**Computing Markov RQE in unknown Markov Games.** The previous results focused on the case when the dynamics and rewards of the RAMG are known. In practice, especially in multi-agent reinforcement learning, the environments can be complex and unknown and must be *learned* by interacting with (i.e., sampling from) the environments. In this subsection, we focus on such scenarios and provide the theoretical guarantees for Algorithm 1.

We focus on a fundamental problem setup which assumes access to a generative model or a simulator (Kearns and Singh, 1998; Agarwal et al., 2020). This allows us to collect $N$ independent samples for each state-action pair generated based on the true environment $\{R_{i,h}, P_h\}_{i \in [n], h \in [H]}$:

$$\forall (s, \boldsymbol{a}, j) \in \mathcal{S} \times \mathcal{A} \times [n] : \quad s_{i,s,\boldsymbol{a},h} \overset{i.i.d}{\sim} P_h(\cdot \,|\, s, \boldsymbol{a}), \quad r_{j,s,\boldsymbol{a},h}^i = R_{j,h}(s, \boldsymbol{a}), \quad i \in [N]. \quad (14)$$

The total sample size is, therefore, $N_{\mathsf{all}} := NS \prod_{i \in [n]} A_i$.

For problems we propose a model-based approach to the computation of Markov RQE which first constructs an empirical reward function and nominal transition kernel based on the collected samples and then applies Algorithm 1 to learn a Markov RQE. First, the empirical reward function and transition kernel $\widehat{P} \in \mathbb{R}^{S \prod_{i=1}^{n} A_i \times S}$ are constructed from the empirical frequency of state transitions,

$$\forall (s, \boldsymbol{a}, j, h) : \widehat{R}_{j,h}(s, \boldsymbol{a}) = r_{j,s,\boldsymbol{a},h}^1 \quad \text{and} \quad \widehat{P}_h(s' \,|\, s, \boldsymbol{a}) = \frac{1}{N} \sum_{i=1}^{N} \mathbb{1}\{s_{i,s,\boldsymbol{a}} = s'\}. \quad (15)$$

Then with such empirical reward and transition, we can apply the oracle summarized in Algorithm 1 to compute Markov RQE using model-based MARL. The following theorem provides the first finite-sample guarantees for the computation or Markov RQE.

**Theorem 5.** *For any RAMG $\mathcal{MG}$, we consider penalty functions $\{D_{\mathsf{env},i}(\cdot, \cdot)\}_{i \in [n]}$ are L-Lipschitz w.r.t the $\ell_1$ norm of the second argument with any fixed first argument. Applying Algorithm 1 with the estimated reward $\{\widehat{R}_{j,h}\}$ and transition kernels $\{\widehat{P}_h\}_{h \in [H]}$ as input, the output solution $\widehat{\pi}$ is an $\delta$-RQE of $\mathcal{MG}$. Namely, we have $\max_{(i,s,h) \in [n] \times \mathcal{S} \times [H]} \Big\{ V_{i,h}^{\epsilon_i}(\widehat{\pi}; s) - \min_{\pi_h' : \mathcal{S} \mapsto \Delta_{\mathcal{A}_i}} V_{i,h}^{\epsilon_i}((\pi_h', \widehat{\pi}_{i,-h}) \times$*

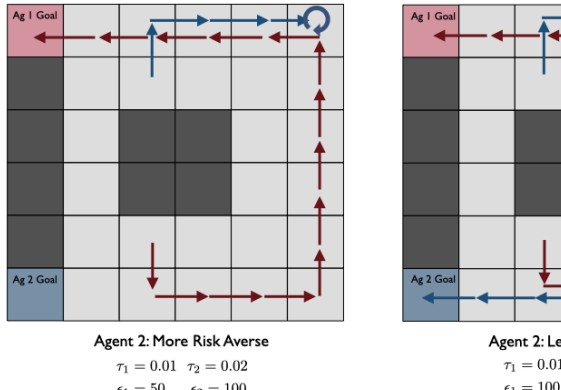

Figure 2: *Cliff Walk Description and Results*: The *Cliff Walk* grid-world is depicted here with color codes of grids: black is the cliff, blue and pink are agents/players 1 and 2's goals respectively. The agents' proximity to one another increases the randomness in the dynamics making falling into one of the cliff states more likely. Agent 2's policy in the left figure showcases more risk-aversion and less bounded rationality than on the right.

$\widehat{\pi}_{-i}; s)\Big\} \leq \delta$ *as long as the total number of samples satisfies*

$$N_{\text{all}} = NS \prod_{i\in[n]} A_i \geq 8S \prod_{i\in[n]} A_i HL \sqrt{\frac{S}{N} \log\left(2SH \prod_{i\in[n]} A_i/\delta\right)}.$$

The proof is postponed to Appendix E. We remark that $L$ can be some constant for various penalty function, such as $L = 1$ when $\{D_{\text{env},i}\}_{i\in[n]}$ are defined as any $\ell_p$ norm including total variation (TV). We also note that our result suffers from what is known as the curse of mutliagency (Bai et al., 2020) through the dependence on $\prod_{i\in[n]} A_i$.

### 4.3 EXPERIMENTS AND EVALUATION

We consider a grid-world problem to test our algorithm (Algorithm 1) and showcase the effects of risk-aversion and bounded rationality in games.

**Cliff Walk Environment description:** A grid consists of some tiles representing a cliff where they will remain stuck for all time and goal states for agents as well as goal states of agents. The cliff is the black grid with rewards $-2$. Agents/players are rewarded $0$ for taking each step and $1$ for reaching their respective goals. Agents actions are {up,down,left,right} and they are followed with probability $p_{\text{d}} = 0.9$ with random movements happening otherwise. To introduce multi-agent effects we reduce $p_{\text{d}}$ to $0.5$ when the agents are least a grid cell apart— making the likelihood of falling into the cliff higher. The episode horizon $H = 200$ and the joint state space is the tuple of players' positions.

**Results Discussion:** We present two results in Fig. 2. For both results, agent 1 starts at the 5th row and 3rd column of the cliff-walking grid-world, and agent 2 starts at the 2nd row and 3rd column grid position. In both figures, the red and blue paths depict the maximum likelihood paths taken by agents 1 and 2 respectively. Agent 2's policy in the left figure showcases more risk-aversion and less bounded rationality resulting in them preferring to hide far from obstacles than run the risk of falling off the path. On the right, agent 2 successfully reaches their goal. The equilibrium strategy of agent 1 changes in both scenarios: a more risk-seeking agent 2 forces agent 1—in an effort to minimize risk—to wait until the path is clear to attempt the journey to its goal.

## 5 CONCLUSION

By incorporating risk aversion and bounded rationality into agents' decision-making processes, we introduced a new class of equilibria for matrix games and finite-horizon Markov games: RQE. RQE were shown to align well with observed human behavior and we provided theoretical results showing that classes of RQE are tractably computable in all finite horizon Markov games. We also provided sample complexity results in the generative modeling setting for multi-agent reinforcement learning. Altogether, our results open the doors to the principled development of new MARL algorithms.

ACKNOWLEDGMENT

The work of LS is supported in part by the Resnick Institute and Computing, Data, and Society Postdoctoral Fellowship at California Institute of Technology. KP acknowledges support from the 'PIMCO Postdoctoral Fellow in Data Science' fellowship at the California Institute of Technology. EM acknowledges support from NSF Award 2240110.

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
