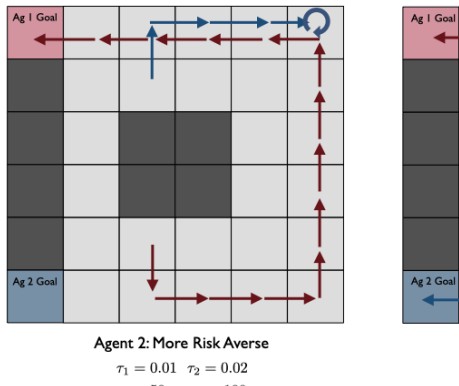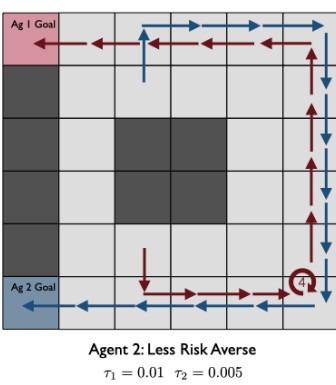

**Agent 2: More Risk Averse**
$\tau_1 = 0.01$  $\tau_2 = 0.02$
$\epsilon_1 = 50$    $\epsilon_2 = 100$

**Agent 2: Less Risk Averse**
$\tau_1 = 0.01$  $\tau_2 = 0.005$
$\epsilon_1 = 100$    $\epsilon_2 = 200$

Figure 2: *Cliff Walk Description and Results*: The *Cliff Walk* grid-world is depicted here with color codes of grids: black is the cliff, blue and pink are agents/players 1 and 2's goals respectively. The agents' proximity to one another increases the randomness in the dynamics making falling into one of the cliff states more likely. Agent 2's policy in the left figure showcases more risk-aversion and less bounded rationality than on the right.

$\widehat{\pi}_{-i}; s) \Big\} \leq \delta$ *as long as the total number of samples satisfies*

$$N_{\mathsf{all}} = NS \prod_{i \in [n]} A_i \geq 8S \prod_{i \in [n]} A_i HL \sqrt{\frac{S}{N} \log \left(2SH \prod_{i \in [n]} A_i / \delta\right)}.$$

The proof is postponed to Appendix E. We remark that $L$ can be some constant for various penalty function, such as $L = 1$ when $\{D_{\mathsf{env},i}\}_{i \in [n]}$ are defined as any $\ell_p$ norm including total variation (TV). We also note that our result suffers from what is known as the curse of mutliagency (Bai et al., 2020) through the dependence on $\prod_{i \in [n]} A_i$.

### 4.3 EXPERIMENTS AND EVALUATION

We consider a grid-world problem to test our algorithm (Algorithm 1) and showcase the effects of risk-aversion and bounded rationality in games.

**Cliff Walk Environment description:** A grid consists of some tiles representing a cliff where they will remain stuck for all time and goal states for agents as well as goal states of agents. The cliff is the black grid with rewards $-2$. Agents/players are rewarded $0$ for taking each step and $1$ for reaching their respective goals. Agents actions are {up,down,left,right} and they are followed with probability $p_{\mathsf{d}} = 0.9$ with random movements happening otherwise. To introduce multi-agent effects we reduce $p_{\mathsf{d}}$ to $0.5$ when the agents are least a grid cell apart— making the likelihood of falling into the cliff higher. The episode horizon $H = 200$ and the joint state space is the tuple of players' positions.

**Results Discussion:** We present two results in Fig. 2. For both results, agent 1 starts at the 5th row and 3rd column of the cliff-walking grid-world, and agent 2 starts at the 2nd row and 3rd column grid position. In both figures, the red and blue paths depict the maximum likelihood paths taken by agents 1 and 2 respectively. Agent 2's policy in the left figure showcases more risk-aversion and less bounded rationality resulting in them preferring to hide far from obstacles than run the risk of falling off the path. On the right, agent 2 successfully reaches their goal. The equilibrium strategy of agent 1 changes in both scenarios: a more risk-seeking agent 2 forces agent 1—in an effort to minimize risk—to wait until the path is clear to attempt the journey to its goal.

## 5 CONCLUSION

By incorporating risk aversion and bounded rationality into agents' decision-making processes, we introduced a new class of equilibria for matrix games and finite-horizon Markov games: RQE. RQE were shown to align well with observed human behavior and we provided theoretical results showing that classes of RQE are tractably computable in all finite horizon Markov games. We also provided sample complexity results in the generative modeling setting for multi-agent reinforcement learning. Altogether, our results open the doors to the principled development of new MARL algorithms.

ACKNOWLEDGMENT

The work of LS is supported in part by the Resnick Institute and Computing, Data, and Society Postdoctoral Fellowship at California Institute of Technology. KP acknowledges support from the 'PIMCO Postdoctoral Fellow in Data Science' fellowship at the California Institute of Technology. EM acknowledges support from NSF Award 2240110.

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

---

**Algorithm 1:** Computation method of RQE for risk-averse Markov games (RAMGs).

---

1: **Input:** reward function $\{R_{i,h}\}_{i\in[n]\times h\in[H]}$, transition kernel $\{P_h\}_{h\in[H]}$.
2: **Initialization:** $\widehat{Q}_{i,h}(s,a) = 0$, $\widehat{V}_{i,h}(s) = 0$ for all $(s, \boldsymbol{a}, h) \in \mathcal{S} \times \mathcal{A} \times [H+1]$.
3: **for** $h = H, H-1, \cdots, 1$ **do**
4:     **for** $i = 1, 2, \cdots, n$ and $s \in \mathcal{S}, \boldsymbol{a} \in \mathcal{A}$ **do**
5:         Set $\widehat{Q}_{i,h}(s, \boldsymbol{a})$ according to (13).
6:     **end for**
7:     **for** $s \in \mathcal{S}$ **do**
8:         Get $\pi_h(s) = \{\pi_{i,h}(s)\}_{1\le i\le n} \leftarrow \mathsf{RQE}\left(\left\{f_{\mathsf{pol},i}^{\pi,\epsilon_i}(\widehat{Q}_{i,h}(s,:))\right\}_{1\le i\le n}\right)$.
9:         Set $\widehat{V}_{i,h}(s) = f_{\mathsf{pol},i}^{\pi_h,\epsilon_i}\left(\widehat{Q}_{i,h}(s,:)\right)$.
10:     **end for**
11: **end for**    **Output:** $\widehat{\pi} = \{\pi_h\}_{1\le h\le H}$.

---

# A   RELATED WORKS

We put our work into context, and discuss related works here.

**Computational tractability of game theoretic solution concepts.** This work proposes a new solution concept for game theoretic settings that is computationally tractable, yet retains many of the desirable properties of classical equilibrium concepts. This general question emerged from the finding that computing a Nash equilibrium—perhaps the most natural solution concept for a game between rational self-interested agents—is PPAD-hard (Daskalakis, 2013), even for two-player general-sum matrix games. Despite this negative result, a large amount of subsequent work has focused on understanding the classes of games in which one can compute, approximate, or learn Nash equilibria efficiently. This is often done by assuming additional structure on the players' utilities and their relationships to one another, with large classes of games being zero-sum or competitive games, zero-sum polymatrix games (Cai et al., 2016; Kalogiannis and Panageas, 2023), monotone games (Golowich et al., 2020), smooth games (Roughgarden, 2015), or socially concave games (Even-Dar et al., 2009).

In games without such structure however, the natural targets for computation and learning became correlated (Moulin and Vial, 1978) and coarse correlated equilibria (Aumann, 1974; 1987)(CE and CCE respectively), both of which can be shown to emerge as the endpoint of no-regret learning and are thus considered to be computationally tractable targets for the design of learning algorithms. Despite this desirable property, the two concepts have significant drawbacks. Indeed both CE and CCE require some form of coordination between players to implement, introduce a highly nontrivial equilibrium selection problem (Cesa-Bianchi and Lugosi, 2006), and may have support on dominated strategies (Viossat and Zapechelnyuk, 2013). Furthermore, in the dynamic game context of Markov games, stationary CE and CCE are also computationally intractable to compute (Daskalakis et al., 2023b).

More recently, a new equilibrium concept—a smoothed Nash equilibrium— has been proposed as an alternative to these other equilibrium concepts (Daskalakis et al., 2023a) and motivated by similar considerations of individual and independent rationalizability and computational tractability. By applying ideas from smoothed analysis to the problem of computing Nash equilibria the authors show that one can efficiently find approximate classes of smoothed Nash equilibria—though to the best of our knowledge this cannot be done in a decentralized way.

Our approach is orthogonal and is rooted in giving MARL agents a foundation rooted in behavioral economics by imbuing them with a realistic feature of human decision-making: risk-aversion. The question of computational tractability of risk-averse Nash equilibria has been analyzed in (Fiat and Papadimitriou, 2010). The work shows that if agents are risk-averse with respect to all the randomness in the game (including their own) then a risk-averse Nash equilibrium may not even exist in mixed strategies, and even understanding if such equilibria exist can be NP-complete. Our formulation overcomes this by incorporating risk-aversion in a different way. Indeed, we show that when agents are risk-averse *only to the randomness introduced to their opponents (and the environment)* then the

risk-averse Nash equilibria will *always* exist. We note that such formulations of risk-aversion are common in the literature on risk-sensitive control (Shen et al., 2013; Borkar, 2023) and risk-sensitive reinforcement learning (Shen et al., 2014) where agents are implicitly presumed to be risk-averse only to the randomness that is outside their control (i.e., the environment). Furthermore we show that introducing bounded rationality into the game allows a class of risk-averse quantal response equilibria (RQE) to be computationally tractable in *all* finite action and finite-horizon Markov games.

**Predictive power of equilibrium concepts.** Another driving force in moving beyond the Nash and correlated equilibrium concepts stems from their lack of predictive power in experimental settings (see e.g., (Brown and Rosenthal, 1990; O'Neill, 1987; McKelvey and Palfrey, 1992; Erev and Roth, 1998; McKelvey and Palfrey, 1995)). To address this, a line of work originating in economics seeks to understand the natural solution concepts in game where players have behaviorally plausible restrictions to their strategy spaces, and to study whether such equilibria were better predictors of human play than Nash or (coarse) correlated equilibria (Goeree et al., 2003; Goeree and Offerman, 2002; Ho et al., 2004). The most common restriction is that players have *bounded rationality*,— i.e., they may fail to perfectly optimize—a model with roots in mathematical psychology (Luce, 1959). Under this restriction, a natural equilibrium concept that emerged was that of a quantal response equilibrium (QRE) which induces bounded rationality by either assuming that the players are rational in a stochastically perturbed version of the game or equivalently that they optimize a regularized version of their utility (McKelvey and Palfrey, 1995; 1998; Mertikopoulos and Sandholm, 2016). Beyond their use as a better model for human decision-making in games, QRE have also increasingly been adopted as a solution concept in multi-agent reinforcement learning and learning in games (Sokota et al., 2023; Mertikopoulos and Sandholm, 2016; Cen et al., 2021; Leonardos et al., 2021; Evans and Ganesh, 2024; Jacob et al., 2022) due to their links with KL and entropy regularized reinforcement learning. Despite these developments QRE are not computable in all games. Indeed the class of QRE or equivalently the level of bounded rationality needed for computational tractability depends on the underlying game structure which may not be known a priori. In contrast we show that the addition of risk aversion allows for the *same* class of quantal response equilibria to be computationally tractable to compute in all finite action games and finite-horizon Markov games. Furthermore we show that this class of risk-averse QRE is nontrivial and can capture human data better than risk-neutral QRE—a finding which is in line with findings in behaviorial economics (Goeree et al., 2003; Goeree and Offerman, 2002).

**Risk-averse and robust multi-agent reinforcement learning.** Our work builds on and provides an additional justification for risk-sensitive (multi-agent) reinforcement learning. This line of work has roots going back to seminal work by Jacobson on risk-sensitive control (Jacobson, 1973), and more recently in risk-sensitive reinforcement learning (Shen et al., 2014). In these works, the aim is to find a controller or policy for a system that accounts for stochasticity or uncertainty in the environment or system in a more nuanced way than risk-neutral approaches like optimal control or reinforcement learning (Borkar, 2023). Due to classic duality results (see e.g., (Panaganti et al., 2022; Zhang et al., 2024)) this line of work is closely related to the literature on robust control and distributionally robust reinforcement learning (Iyengar, 2005; Panaganti and Kalathil, 2022; Xu* et al., 2023) which seeks to find solutions that are robust to worst case environmental disturbances.

Our work rigorously extends these formulations to the multi-agent regime though it is not the first to consider risk-aversion in MARL. Indeed, risk-sensitive MARL has been the focus of several recent works (e.g., (Yekkehkhany et al., 2020; Wang et al., 2024; Gao et al., 2021; Slumbers et al., 2023)). Several provide rigorous definitions of risk-averse equilibria and some guarantees on their computation by assuming structure on the risk-averse game. Oftentimes this is done by assuming that the risk-averse game is itself zero-sum (Yekkehkhany et al., 2020), monotone (Wang et al., 2024), or that it satisfies other strong conditions (Gao et al., 2021). Other works are more empirical in nature (Ganesh et al., 2019; Eriksson et al., 2022; Zhang et al., 2021b; Qiu et al., 2021; Shen et al., 2023; Slumbers et al., 2023), showing the promise of risk-averse algorithms for MARL.

Another line of work originates in the literature on robust and risk-sensitive control where the focus has largely been on understanding the solution to stochastic dynamic games (Borkar, 2023; Basar, 1999; Moon et al., 2019). In these contexts, the focus has often remained on exponential forms of risk and on characterizing properties of the Nash equilibrium solutions—but questions of computational tractability are mostly side-stepped or the problems are analyzed under conditions on the risk-adjusted

game like weak coupling (Basar, 1999) or zero-sum structures (Moon et al., 2019) which may not arise from risk-averse problems.

One last closely related line of work is the emerging literature on robust multi-agent reinforcement learning (Zhang et al., 2020; He et al., 2023; Shi et al., 2024; Blanchet et al., 2024). Once again due to duality arguments, these works can be seen as tackling a similar problem to the risk-averse MARL problem. The focus of these previous works, however, is on robustness in the face of only environmental uncertainties (and not opponent strategies), and questions of existence and computational tractability are either assumed away or the focus is on extensions of correlated equilibrium concepts. A recent related work in this literature analyzed the computational tractability of robust Nash equilibria in Markov games, but only provided strong guarantees on the zero-sum regime, showing that computing such equilibria in general is PPAD-hard (McMahan et al., 2024).

To the best of our knowledge, no previous work in either of these literatures highlights the broad benefits afforded by risk-aversion in MARL in terms of computational tractability of equilibria. In our work we show that risk-aversion (and by extension distributional robustness) to all external randomness, when combined with bounded rationality yields a computationally tractable class of individually rationalizable equilibria in *all* finite-horizon $n$-player Markov games. Furthermore we show that these equilibria can be computed using no-regret learning algorithms.

## B   DETAILS OF RISK-AVERSE AND BOUNDED RATIONAL MATRIX GAMES

In the next two subsections, we provide detailed technical results pertaining to modeling agents as both risk-averse and imperfect optimizers—embedding agents with human decision-making capabilities. These technical results serve as a more rigorous and detailed understanding of our main methodology choices presented in Section 2.

### B.1   RISK-AVERSION IN MATRIX GAMES

In the case of matrix games, we consider the case where agents are risk averse with respect to the mixing or randomness introduced into the game by their opponents. In Markov games, we will also consider the case where agents are risk-averse with respect to the underlying dynamics. To emphasize, we assume players are not risk-averse to their own randomness. We note that is a common approach taken in the literature on risk-sensitive and robust decision-making (Shen et al., 2014) and it is necessary since if agents are risk-averse to their own randomness then an equilibrium may cease to exist (Fiat and Papadimitriou, 2010). We refer to related works for more discussion.

We now recall our generalization of the original game that differ in how risk is incorporated into the problem: *aggregate risk aversion* (4). We also formulate the *action-dependent risk aversion* generalization of the original game here in (17).

**Aggregate Risk Aversion Game:**   The player's utilities take the form

$$f_i(\pi_i, \pi_{-i}) = \rho_{i,\pi_{-i}}(\mathbb{E}_{\pi_i}[R_i(a)]) = \rho_{i,\pi_{-i}}\left(\sum_{a_i \in \mathcal{A}_i} \pi_i(a_i) R_i(a_i, a_{-i})\right) \tag{16}$$

where $\rho_{i,\pi_{-i}}$ is used to denote the potentially different risk preference of agent $i$ which depends on the product distribution of opponents strategies $\pi_{-i}$.

**Action-dependent Risk Aversion Game:**   The player's utilities take the form

$$f_i(\pi_i, \pi_{-i}) = \mathbb{E}_{\pi_i}\left[\rho_{i,\pi_{-i}}(R_i(a))\right] = \sum_{a_i \in \mathcal{A}_i} \pi_i(a_i) \rho_{i,\pi_{-i}}\left(R_i(a_i, a_{-i})\right) \tag{17}$$

where again $\rho_{i,\pi_{-i}}$ is used to denote the potentially different risk preference of agent $i$ which depends on the product distribution of opponents strategies $\pi_{-i}$.

We remark that in both of these formulations, if $\rho_i(X) = \mathbb{E}[-X]$ for all players $i = 1, ..., n$ (which satisfies the requirements in Definition 3) then the new formulation reduces to the original expected utility objective. Thus, both can be seen as generalizations of the classic setup described in the previous section.

| Risk-measure | Penalty function $D(p, q)$ |
|---|---|
| Entropic Risk (Ahmadi-Javid, 2012) | Kullback-Leibler (KL): $KL(p, q) = \sum_i p_i \log \left( \frac{p_i}{q_i} \right)$ |
| (Föllmer and Schied, 2002) | Reverse KL (RKL): $\sum_i q_i \log \left( \frac{q_i}{p_i} \right)$ |
| $\phi$-Entropic Risk (Ahmadi-Javid, 2012) | $\phi$-Divergence: $\sum_i p_i \phi \left( \frac{p_i}{q_i} \right)$ |
| Utility-based shortfall (Föllmer and Schied, 2002) | Utility-based Shortfall $(u)$: $\inf_{\lambda > 0} \frac{1}{\lambda} [r + \sum_i p_i u^\star (\lambda \frac{q_i}{p_i})]$ |

Table 1: We list several widely-used convex risk-measures with its penalty function $D(p, q)$. Here, $p, q$ are distributions of the same finite dimension, $\phi$ and $u$ are differentiable convex functions on $\mathbb{R}$ satisfying $\phi(1) = 0$ and $\phi(t) = +\infty$ for $t < 0$. The utility $u$ is equipped with risk tolerance level of $r$, and its penalty function depends on its convex conjugate $u^\star$.

We also remark that in both of these formulations, due to monotonicity and linearity of expectation both can be seen as generalizations of risk-sensitive decision-making investigated in the literature on risk-sensitive control (Borkar, 2023) and risk-sensitive RL (Shen et al., 2014). To see this reduction, one can simply take the other agents to be part of an unknown environment.

We now rely on a particularly powerful property of convex measures of risk to simplify and expose some structure for these two risk-adjusted games. In particular, we make use of the following *dual* representation theorem for convex risk measures.

**Theorem 6** (Dual Representation Theorem for Convex Risk Measures (Föllmer and Schied, 2002))**.** *Suppose that the set $\mathcal{X}$ is the set of functions mapping from a finite set $\Omega$ to $\mathbb{R}$. Then a mapping $\rho : \mathcal{X} \to \mathbb{R}$ is a convex risk measure (cf. Definition 3) if and only if there exists a penalty function $D : \Delta_\Omega \to (-\infty, \infty]$ such that: $\rho(X) = \sup_{p \in \Delta_\Omega} E_p[-X] - D(p)$, where $\Delta_\Omega$ is the set of all probability measures on $\Omega$. Furthermore, the function $D(p)$ can be taken to be convex, lower-semi-continuous, and satisfy $D(p) > -\rho(0)$ for all $p \in \Delta_\Omega$.*

When the set $\mathcal{X}$ is again the set of measurable functions defined on a probability space, one can choose the penalty function $D$ to represent a notion of of distance from the probability law or distribution of the random variable $\pi$. In such cases, the dual representation theorem takes the form:

$$\rho_\pi(X) = \sup_{p \in \Delta_\Omega} E_p[-X] - D(p, \pi),$$

where $D(p, \pi)$ is convex in $p$ for a fixed $\pi$. We also make a simplifying assumption that $D$ is continuous in both its arguments, which is satisfied by various widely-used risk measures. This general form allows us to draw connections with a large class of risk and robustness metrics that are based around $\phi$-divergences. We provide examples of common risk measures in Table 1.

Given this reformulation tool, we recover the alternate aggregate risk-averse game (5) in the following form:

$$f_i(\pi_i, \pi_{-i}) = \sup_{p_i \in \mathcal{P}_{-i}} -\pi_i^T R_i p_i - D_i(p_i, \pi_{-i}) \tag{18}$$

where $\mathcal{P}_{-i} = \mathcal{P}/\Delta_{A_i} \subset \mathbb{R}^{A_{-i}}$, $A_{-i} = \sum_{j \neq i} A_j$, and $R_i \in \mathbb{R}^{A_i \times A_{-i}}$ is player $i$'s payoff matrix. Similarly, the action-dependent risk-averse game also takes the form:

$$f_i(\pi_i, \pi_{-i}) = \sum_{j \in \mathcal{A}_i} \pi_i(j) \left( \sup_{p_{i,j} \in \mathcal{P}_{-i}} -\langle R_{i,j}, p_{i,j} \rangle - D_i(p_{i,j}, \pi_{-i}) \right), \tag{19}$$

where $R_{i,j}$ corresponds to the $jth$ row of $R_i$.

As noted before, different penalty functions $D_i$ allow different agents to have different risk preferences. In this form, one can see that in a risk-averse game, the players imagine that intermediate adversaries seek to maximize their cost but are penalized from deviating too far from the opponents' realized strategies. Thus, agents in risk-averse games introduce a certain amount of worst-case thinking into their strategies.

We have thus introduced a risk-averse Nash equilibrium (RNE) concept (see Definition 4). The convexity and continuity of the penalty function guarantees that the risk-averse games admit at least one RNE.

**Theorem 7.** *There always exists a RNE for all aggregate and action-dependent risk-averse games presented in* (5) *and* (19) *respectively.*

*Proof.* To begin, we show that $f_i(\pi_i, \pi_{-i})$ is convex in $\pi_i$ for all $\pi_i \in \mathcal{P}_{-i}$ in (5). This follows from the fact that $D_i(\cdot, \cdot)$ is assumed to be convex and continuous in its first argument. Invoking Danskin's theorem guarantees us that $f_i$ is convex in $\pi_i$ for all fixed values of $\pi_{-i}$. Since the probability simplex is compact and convex, the game satisfies the conditions of a convex game (Rosen, 1965) and thus a Nash equilibrium must exist. The action-dependent risk aversion regime follows from the linearity of $f_i$ in $\pi_i$ in (19) and a similar invocation of the existence of Nash in convex games (Rosen, 1965). □

We note that even though risk preferences already help convexify player's objectives, the addition of risk can serve to weaken existing structures in the player's cost function. To illustrate this, we show that even in two-player zero-sum games where $R_1 = R = -R_2^T$, the risk-averse game loses any zero-sum structure and may cease to even be strictly competitive in the sense that one player's gain is the other's loss. Thus, additional convexity induced by the introduction of risk aversion guarantee is not enough to ensure the computational tractability of the Nash equilibrium (see e.g., (McMahan et al., 2024)).

**Example 1.** *Consider a 2-player zero-sum game where $R_1 = R = -R_2^T$ where players have aggregate risk aversion in the entropic risk metric with different degrees of risk aversion $\tau_1$ and $\tau_2$. Under these conditions, the players loss functions take the following form:*

$$f_1(\pi_1, \pi_2) = \sup_{p_1 \in \mathcal{A}_2} -\pi_1^T R p_1 - \frac{1}{\tau_1} KL(p_1, \pi_2) = \frac{1}{\tau_1} \log \Big( \sum_{1 \leq j \leq A_2} \pi_2(j) \exp(-\tau_1 [R\pi_1]_j) \Big)$$

$$f_2(\pi_1, \pi_2) = \sup_{p_2 \in \mathcal{A}_1} \pi_2^T R^T p_2 - \frac{1}{\tau_2} KL(p_2, \pi_1) = \frac{1}{\tau_2} \log \Big( \sum_{1 \leq j \leq A_1} \pi_1(j) \exp(\tau_2 [R^T \pi_2]_j) \Big).$$

*Even instantiating $R = \mathbb{I}_2$, $\tau_1 = 10$, and for any $\tau_2 > 0$, both $f_1(\pi_1, \pi_2) > f_1(\pi_1', \pi_2)$ and $f_2(\pi_1, \pi_2) > f_2(\pi_1', \pi_2)$ holds for the regions $\pi_1, \pi_2 \in \Delta_2$ satisfying $\pi_2(1) \in (0.1, 0.5)$ and $0.75 - \pi_1(1) > \pi_1'(1) > \pi_1(1)$, which implies that the game is not a strictly competitive game.*

Note that the previous example introduces degrees of risk aversion into the game through the parameters $\tau_1 > 0$ and $\tau_2 > 0$. As $\tau$ increases, the game becomes less reliant on the regularization term, which makes the adversary more powerful and results in more conservative game playing by the player. As $\tau$ goes to zero we recover the risk neutral regime (Ahmadi-Javid, 2012).

B.2    BOUNDED RATIONALITY IN MATRIX GAMES

Since risk aversion on its own is not sufficient to guarantee computational tractability of NE, we introduced human decision-making into the game: bounded rationality. To incorporate bounded rationality into agents, we resort to the notion of a *quantal response function* (see Definition 2).

Clearly, when players responses are constrained to be *quantal* responses, they cannot be perfect maximizers since the $\arg\max$ function does not satisfy the first desiderata of a quantal response function. Common quantal response functions include the logit response function:

$$\sigma(x) = \frac{\exp(-\frac{1}{\epsilon} x_i)}{\sum_{j=1}^n \exp(-\frac{1}{\epsilon} x_j)}, \tag{20}$$

where the sign is to account for the fact that agents may be *minimizing* their loss function.

To incorporate quantal responses into our risk-averse game, we introduced regularization to the player's losses in (6). Recalling:

$$f_i^{\epsilon_i}(\pi_i, \pi_{-i}) = f_i(\pi_i, \pi_{-i}) + \epsilon_i \nu_i(\pi_i) \tag{21}$$

where $\nu_i$ is strictly convex over the simplex and controls the class of quantal responses available to player $i$ and $\epsilon_i$ controls the agent's degree of bounded rationality. This can be shown to be equivalent to constraining the player's responses to quantal responses (see, e.g., (Föllmer and Schied, 2002, Proposition 7), or (Sokota et al., 2023; Mertikopoulos and Sandholm, 2016)).

**Example 2.** *If players are constrained to* logit *response functions, one can reflect this by incorporating a negative entropy regularizer $\nu(\pi) = \sum_i p_i \log(p_i)$. Another class of quantal response functions would be generated by making use of e.g., a log-barrier regularizer $\nu(\pi) = -\sum_i \log(p_i)$. Both of these regularizers give rise to quantal response functions that satisfy Definition 2.*

This game now incorporates two key properties of human decision-making: risk aversion and bounded rationality on the part of the agents. The natural outcome of this game is what we termed a risk-averse quantal-response equilibrium (RQE).

## C  FURTHER RESULTS ON COMPUTATIONAL TRACTABILITY OF RQE IN AGGREGATE RISK-AVERSE MATRIX GAMES

In this section we provide further results on the aggregate risk aversion formulation of the risk-averse problem. This section provides a proof of the main result of Theorem 3 presented in the main body of the paper as well as the generalization of the results on computationaly tractability to $n$-players. A key step in the proof is relating the Nash equilibrium of the $2n$-player convex game presented where players have utilites of the form (8) and (7) and RQE of the desired game (4). We then use this result to prove Theorem 3.

### C.1  RELATING NASH OF THE $2n$-PLAYER GAME AND RQE

We first note that this $2n$ player game is a convex game played over compact convex action sets and thus a Nash equilibrium must exist (Rosen, 1965). To relate outcomes in this new game to that of the original game, we show that the strategies played by the original players in Nash equilibria of the $2n$-player game coincide with the RQE.

**Proposition 1.** *If $(\pi^*, p^*)$ is a Nash equilibrium of the $2n$-player game, then $\pi^*$ is a RQE of Game (6). Furthermore, if $\pi^*$ is a RQE of Game (6), then $(\pi^*, p^*)$ is a Nash equilibrium of the $2n$-player game, where:*

$$p_i^* = \arg \max_{p_i \in \mathcal{P}_{-i}} -\pi_i^{*T} R_i p_i - D_i(p_i, \pi_{-i}^*). \tag{22}$$

*Proof.* To prove this result, we rely on the definitions of Nash and RQE. We begin by proving the first claim of the proposition. Recall that a Nash equilibrium is a joint strategy $(\pi^*, p^*) \in \mathcal{P} \times \bar{\mathcal{P}}$ such that, for all $i = 1, ..., n$:

$$J_i(\pi_i^*, \pi_{-i}^*, p^*) \leq J_i(\pi_i', \pi_{-i}^*, p^*) \ \forall \ \pi_i' \in \Delta_{A_i} \tag{23}$$

$$\bar{J}_i(\pi^*, p_i^*, p_{-i}^*) \leq \bar{J}_i(\pi^*, p_i', p_{-i}^*) \ \forall \ p_i' \in \mathcal{P}_{-i}$$

Noting that each $J_i(\pi_i^*, \pi_{-i}^*, p_i^*, p_{-i}^*)$ does not depend on $p_{-i}^*$, to show the forward direction, we start by taking the supremum of the right hand side of (23) over $p_i \in \mathcal{P}_{-i}$. Thus, we find that, for all $i$:

$$J_i(\pi_i^*, \pi_{-i}^*, p^*) \leq \sup_{p_i \in \mathcal{P}_{-i}} J_i(\pi_i', \pi_{-i}^*, p_i, p_{-i}^*) \ \forall \ \pi_i' \in \Delta_{A_i}$$

$$= f_i^{\epsilon_i}(\pi_i', \pi_{-i}^*) \ \forall \ \pi_i' \in \Delta_{A_i},$$

where in the second line we used the fact that for any $\pi, p_{-i}$, by definition,

$$\sup_{p_i \in \mathcal{P}_{-i}} J_i(\pi, p_i, p_{-i}) = f_i^{\epsilon_i}(\pi).$$

It remains to show that for any $i = 1, ..., n$, $J_i(\pi_i^*, \pi_{-i}^*, p^*) = f_i^{\epsilon_i}(\pi_i^*, \pi_{-i}^*)$. This follows from the fact that the simplex is compact. Indeed for any fixed $\pi$, the function $J_i(\pi, p_i, p_{-i})$ is concave in $p_i$. Thus, the supremum is attained at $p_i^*$ since

$$p_i^* = \arg \min_{p_i \in \mathcal{P}_{-i}} \bar{J}_i(\pi^*, p_i, p_{-i}^*) = \arg \max_{p_i \in \mathcal{P}_{-i}} J_i(\pi^*, p_i, p_{-i}^*).$$

Since the same argument holds for all $i$, we have shown that if $(\pi^*, p^*)$ is a Nash equilibrium of the $2n$-player game, then:

$$f_i^{\epsilon_i}(\pi_i^*, \pi_{-i}^*) \leq f_i^{\epsilon_i}(\pi_i, \pi_{-i}^*) \ \forall \ \pi_i \in \Delta_{A_i}$$

which is the definition of a RQE.

To prove the second claim, suppose that $\pi^*$ is a RQE. By definition, we have that if

$$p_i^* = \arg \max_{p_i \in \mathcal{P}_{-i}} -\pi_i^{*T} R_i p_i - D_i(p_i, \pi_{-i}^*).$$

then $p_i^*$ by construction satisfies the condition of a Nash equilibrium of the $2n$-player game:

$$\bar{J}_i(\pi^*, p_i^*, p_{-i}^*) \leq \bar{J}_i(\pi^*, p_i', p_{-i}^*) \ \forall \ p_i' \in \mathcal{P}_{-i}.$$

It remains to show that $\pi^*$ also satisfies the necessary conditions on $J_i$. To see that this must hold, by definition, we must have that $(\pi_i^*, p_i^*)$ satisfies:

$$J_i(\pi_i^*, p_i^*, \pi_{-i}^*, p_{-i}^*) = f_i^{\epsilon_i}(\pi_i^*, \pi_{-i}^*) = \min_{\pi_i \in \Delta_{A_i}} \max_{p_i \in \mathcal{P}_{-i}} J(\pi_i, p_i, \pi_{-i}^*, p_{-i}^*)$$

Further manipulations allow us to show that:

$$\begin{aligned}
J_i(\pi_i^*, \pi_{-i}^*, p^*) &= f_i^{\epsilon}(\pi_i^*, \pi_{-i}^*) \\
&= \min_{\pi_i \in \Delta_{A_i}} \max_{p_i \in \mathcal{P}_{-i}} J(\pi_i, p_i, \pi_{-i}^*, p_{-i}^*) \\
&= \min_{\pi_i \in \Delta_{A_i}} J(\pi_i, p_i^*, \pi_{-i}^*, p_{-i}^*) \\
&\leq J(\pi_i', p_i^*, \pi_{-i}^*, p_{-i}^*) \ \forall \ \pi_i' \in \Delta_{A_i}
\end{aligned}$$

Since this holds true for all $i$, this completes the proof. $\qquad\square$

## C.2 Proof of Theorem 3 and immediate corollaries

Given the results showing that finding a Nash equilibrium of the $2n$-player convex game allows us to compute RQE, we present the proof of Theorem 3. We also provide two corollaries that specialize the results to different risk-metrics and quantal response functions.

**Theorem 8** (Restatement of Theorem 3). *Assume the penalty functions that give rise to the players' risk preferences $D_1(\cdot, \cdot)$ and $D_2(\cdot, \cdot)$ are jointly convex in both their arguments. If $\sigma$ is a CCE of the four player game with $\xi_1 = \xi_1^*$ and $\xi_2 = \xi_2^*$, and*

$$\frac{\epsilon_1}{\xi_1^*} \geq \frac{\xi_2^*}{\epsilon_2},$$

*then $\hat{\pi}_1 = \mathbb{E}_\sigma[\pi_1]$ and $\hat{\pi}_2 = \mathbb{E}_\sigma[\pi_2]$ constitute a RQE of the original game.*

The proof follows by showing that CCE of the $2n$-player convex game coincide with Nash equilibria of the $2n$-player convex game, and then invoking results from the previous section.

*Proof.* To prove this, we show that $(\hat{\pi}_1, \hat{p}_1, \hat{\pi}_2, \hat{p}_2)$ is a Nash equilibrium of the four player game and then invoke Proposition 1 (where $\hat{\pi}_1 = \mathbb{E}_\sigma[\pi_1]$, $\hat{p}_2 = \mathbb{E}_\sigma[p_2]$, $\hat{\pi}_2 = \mathbb{E}_\sigma[\pi_2]$, and $\hat{p}_2 = \mathbb{E}_\sigma[p_2]$).

To begin, we focus on $J_1$ and $\bar{J}_1$. By symmetry, the same arguments hold for $J_2$ and $\bar{J}_2$. We first note that $J_1$ is (strictly) convex in $\pi_1$ for all fixed $p_1, \pi_2, p_2$ and jointly concave in $p_1, \pi_2, p_2$ for all fixed $\pi_1$. Now, starting with the definition of CCE for $\sigma$ and via Jensen's inequality, we have that:

$$\begin{aligned}
\mathbb{E}_\sigma[J_1(\pi_1, p_1, \pi_2, p_2)] &\leq \mathbb{E}_{(p_1, \pi_2, p_2) \sim \sigma}[J_1(\pi_1', p_1, \pi_2, p_2)] \\
&= \mathbb{E}_{p_1 \sim \hat{p}_1}\left[\mathbb{E}_{\pi_2 \sim \sigma \mid p_1}\left[\mathbb{E}_{p_2 \sim \sigma \mid p_1, \pi_2}[J_1(\pi_1', p_1, \pi_2, p_2)]\right]\right] \\
&\leq \mathbb{E}_{p_1 \sim \hat{p}_1}\left[\mathbb{E}_{\pi_2 \sim \sigma \mid p_1}\left[J_1(\pi_1', p_1, \pi_2, \mathbb{E}_{\sigma \mid p_1, \pi_2}[p_2])\right]\right] \\
&\leq \mathbb{E}_{p_1 \sim \hat{p}_1}\left[J_1(\pi_1', p_1, \mathbb{E}_{\sigma \mid p_1}[\pi_2], \mathbb{E}_{\sigma \mid p_1}[p_2])\right] \\
&\leq J_1(\pi_1', \hat{p}_1, \hat{\pi}_2, \hat{p}_2) \quad \forall \pi_1' \in \Delta_{A_1}.
\end{aligned} \tag{24}$$

Similarly, we note that for $\xi_1 = \xi_1^*$, $\bar{J}_1$ is jointly concave in $\pi_1, \pi_2, p_2$, such that:

$$\mathbb{E}_\sigma[\bar{J}_1(\pi_1, p_1, \pi_2, p_2)] \leq \bar{J}_1(\hat{\pi}_1, p'_1, \hat{\pi}_2, \hat{p}_2) \quad \forall p'_1 \in \Delta_{A_2}.$$

Letting $\hat{z} = (\hat{\pi}_1, \hat{p}_1, \hat{\pi}_2, \hat{p}_2)$ and $z = (\pi_1, p_1, \pi_2, p_2)$ to simplify notation, we can now take a weighted sum of the four utility functions with $\lambda \in (0, 1)$ to find that:

$$\frac{\lambda}{2}\left(J_1(\hat{z}) + \bar{J}_1(\hat{z})\right) + \frac{1-\lambda}{2}\left(J_2(\hat{z}) + \bar{J}_2(\hat{z})\right)$$

$$\geq \mathbb{E}_\sigma\left[\frac{\lambda}{2}\left(J_1(z) + \bar{J}_1(z)\right) + \frac{1-\lambda}{2}\left(J_2(z) + \bar{J}_2(z)\right)\right]$$

$$= \frac{1}{2}\left(\lambda\epsilon_1 - (1-\lambda)\xi_2^*\right)\mathbb{E}_\sigma\left[\nu_1(\pi_1)\right] + \frac{1}{2}\left((1-\lambda)\epsilon_2 - \lambda\xi_1^*\right)\mathbb{E}_\sigma\left[\nu_2(\pi_2)\right].$$

Choosing $\lambda = \xi_2^*/(\epsilon_1 + \xi_2^*)$, we can further simplify to find that:

$$\mathbb{E}_\sigma\left[\frac{\lambda}{2}\left(J_1(\hat{z}) + \bar{J}_1(\hat{z})\right) + \frac{1-\lambda}{2}\left(J_2(\hat{z}) + \bar{J}_2(\hat{z})\right)\right]$$

$$= \frac{1}{2}\left(\frac{\epsilon_1\epsilon_2}{\epsilon_1 + \xi_2^*} - \frac{\xi_1^*\xi_2^*}{\epsilon_1 + \xi_2^*}\right)\mathbb{E}_\sigma\left[\nu_2(\pi_2)\right]$$

$$\geq \frac{1}{2}\left(\frac{\epsilon_1\epsilon_2}{\epsilon_1 + \xi_2^*} - \frac{\xi_1^*\xi_2^*}{\epsilon_1 + \xi_2^*}\right)\nu_2(\hat{\pi}_2)$$

$$= \frac{\lambda}{2}\left(J_1(\hat{z}) + \bar{J}_1(\hat{z})\right) + \frac{1-\lambda}{2}\left(J_2(\hat{z}) + \bar{J}_2(\hat{z})\right),$$

where we used the fact that $\frac{\epsilon_1}{\xi_1^*} \geq \frac{\xi_2^*}{\epsilon_2}$ by assumption and invoke Jensen's inequality for $\nu_2$ at the second inequality. Thus we have shown that:

$$\frac{\lambda}{2}\left(J_1(\hat{z}) + \bar{J}_1(\hat{z})\right) + \frac{1-\lambda}{2}\left(J_2(\hat{z}) + \bar{J}_2(\hat{z})\right) \tag{25}$$

$$= \frac{\lambda}{2}\left(\mathbb{E}_\sigma\left[J_1(z)\right] + \mathbb{E}_\sigma\left[\bar{J}_1(z)\right]\right) + \frac{1-\lambda}{2}\left(\mathbb{E}_\sigma\left[J_2(z)\right] + \mathbb{E}_\sigma\left[\bar{J}_2(z)\right]\right). \tag{26}$$

By Eq. (24), we observe:

$$\mathbb{E}_\sigma\left[J_1(z)\right] \leq J_1(\hat{z}), \mathbb{E}_\sigma\left[\bar{J}_1(z)\right] \leq \bar{J}_1(\hat{z}), \mathbb{E}_\sigma\left[J_2(z)\right] \leq J_2(\hat{z}), \mathbb{E}_\sigma\left[\bar{J}_2(z)\right] \leq \bar{J}_2(\hat{z}). \tag{27}$$

We note a fact: $\lambda a + (1-\lambda)b = \lambda c + (1-\lambda)d$ and $a \leq c, b \leq d$ implies $a = c, b = d$ for any $a, b, c, d \in \mathbb{R}$. Using this fact for Eqs. (26) and (27), we have:

$$\mathbb{E}_\sigma\left[J_1(z)\right] = J_1(\hat{z}) \leq J_1(\pi'_1, \hat{p}_1, \hat{\pi}_2, \hat{p}_2) \quad \forall \pi'_1 \in \Delta_{A_1}$$

$$\mathbb{E}_\sigma\left[\bar{J}_1(z)\right] = \bar{J}_1(\hat{z}) \leq \bar{J}_1(\hat{\pi}_1, p'_1, \hat{\pi}_2, \hat{p}_2) \quad \forall p'_1 \in \Delta_{A_2}$$

$$\mathbb{E}_\sigma\left[J_2(z)\right] = J_2(\hat{z}) \leq J_2(\hat{\pi}_1, \hat{p}_1, \pi'_2, \hat{p}_2) \quad \forall \pi'_2 \in \Delta_{A_2}$$

$$\mathbb{E}_\sigma\left[\bar{J}_2(z)\right] = \bar{J}_2(\hat{z}) \leq \bar{J}_2(\hat{\pi}_1, \hat{p}_1, \hat{\pi}_2, p'_2) \quad \forall p'_2 \in \Delta_{A_1}$$

Thus we have shown that $\hat{z} = (\hat{\pi}_1, \hat{p}_1, \hat{\pi}_2, \hat{p}_2)$ is a Nash equilibrium for the 4-player game. By invoking Propostion 1 we derive our result that $(\hat{\pi}_1, \hat{\pi}_2)$ must be a RQE for the original risk-averse game. $\qquad\square$

We remark that the result does not necessarily guarantee uniqueness, though by exploiting the connections between socially convex games and monotone games (Gemp and Mahadevan, 2017) such a result would follow.

We now present two corollaries that specialize the results to specific risk metrics and quantal responses. In the first we look at the case where players make use of the entropic risk and log-barrier reguarlizers.

**Corollary 8.1.** *Suppose the players are risk-averse in the entropic risk metric with parameters $\tau_1$ and $\tau_2$ respectively, meaning that their risk-averse losses are given by:*

$$f_1(\pi_1, \pi_2) = \sup_{p_1 \in \mathcal{A}_2} -\pi_1^T R_1 p_1 - \frac{1}{\tau_1} KL(p_1, \pi_2) \qquad f_2(\pi_1, \pi_2) = \sup_{p_2 \in \mathcal{A}_1} \pi_2^T R_2 p_2 - \frac{1}{\tau_2} KL(p_2, \pi_1)$$

*If they respond in the space of quantal responses generated by the log-barrier regularizers with parameters $\epsilon_1$ and $\epsilon_2$ respectively, and if $\epsilon_1\tau_1 \geq \frac{1}{\epsilon_2\tau_2}$ then, for any $R_1, R_2$ the players can compute a RQE by using no-regret learning.*

In the second corollary we look at the case when players make use of the reverse KL penalty function and logit quantal responses.

**Corollary 8.2.** *Suppose the players are risk-averse and make use of the reverse-KL as a penalty function to give rise to their risk metric with parameters $\tau_1$ and $\tau_2$ respectively. Their risk-averse losses are given by:*

$$f_1(\pi_1, \pi_2) = \sup_{p_1 \in \mathcal{A}_2} -\pi_1^T R_1 p_1 - \frac{1}{\tau_1} RKL(p_1, \pi_2) \ \ f_2(\pi_1, \pi_2) = \sup_{p_2 \in \mathcal{A}_1} \pi_2^T R_2 p_2 - \frac{1}{\tau_2} RKL(p_2, \pi_1)$$

*If they respond in the space of quantal responses generated by the negative entropy regularizor with parameters $\epsilon_1$ and $\epsilon_2$ respectively, and if $\epsilon_1 \tau_1 \geq \frac{1}{\epsilon_2 \tau_2}$ then, for any $R_1, R_2$ the players can compute a RQE by using no-regret learning.*

*Proof.* The proof of these corollaries result comes from the fact that for $\xi^* = \frac{1}{\tau}$, the function $H(p, \pi) = \frac{1}{\tau} RKL(p, \pi) - \xi \nu(\pi)$ is concave in $\pi$ for all $p$ if $\xi \geq \frac{1}{\tau}$. Thus, choosing $\xi^* = \frac{1}{\tau}$ and invoking Theorem 3 completes the proof. □

Note that the proof of this corollary is the same as that of Corollary 8.1 and so we only provide one proof for both.

### C.3 COMPUTING RQE IN n-PLAYER GENERAL-SUM GAMES

We now extend our result to the computation of RQE in $n$-player games. This requires stronger assumptions on the players' risk preferences and bounded rationality parameters. Nevertheless, we once again show that a large class of RQE is computationally tractable in this class of games.

To do so we now define $H_i(p_i, \pi_{-i}) = D_i(p_i, \pi_{-i}) - \sum_{j \neq i} \xi_{i,j} \nu_j(\pi_j)$. For all $i, j \in \{1, ..., n\}$ let $\xi_{i,j}^* > 0$ be the smallest values of $\xi_{i,j}$ such that $H_i(p_i, \pi_{-i})$ is concave in $\pi_j$. Again, the parameters $\xi_{i,j}^*$ capture the player's degrees of risk aversion. The following theorem gives a general condition under which an RQE is computable using no-regret learning.

**Theorem 9.** *Assume the penalty functions that give rise to the players' risk preferences $D_i(\cdot, \cdot)$ are jointly convex in both of their arguments. If $\sigma$ is a CCE of the $2n$-player game with $\xi_{i,j} = \xi_{i,j}^*$ for all $i, j \in \{1, ..., n\}$, and for all $i = 1, ..., n$ we have $\epsilon_i \geq \sum_{j \neq i} \xi_{j,i}^*$, then $\hat{\pi} = \mathbb{E}_\sigma[\pi]$ is a RQE of the risk-averse $n$-player game.*

*Proof.* To prove this, we show that $(\hat{\pi}, \hat{p})$ is a Nash equilibrium of the $2n$-player game and then invoke Proposition 1 (where $\hat{\pi} = \mathbb{E}_\sigma[\pi], \hat{p} = \mathbb{E}_\sigma[p]$).

We focus on $J_i$ and $\bar{J}_i$. We first note that $J_i$ is (strictly) convex in $\pi_i$ for all fixed $p, \pi_{-i}$ and jointly concave in $p, \pi_{-i}$ for all fixed $\pi_i$ by assumption. Thus, via Jensen's inequality, we have that:

$$\mathbb{E}_\sigma[J_i(\pi, p)] \leq J_i(\pi_i', \hat{\pi}_{-i}, \hat{p}) \ \ \forall \pi_i' \in \Delta_{A_i}$$

Similarly, we note that for $\xi_{i,j} = \xi_{i,j}^*$, $\bar{J}_i$ is jointly concave in $\pi, p_{-i}$, such that:

$$\mathbb{E}_\sigma[\bar{J}_i(\pi, p)] \leq \bar{J}_i(\hat{\pi}, p_i', \hat{p}_{-i}) \ \ \forall p_i' \in \mathcal{P}_{-i}.$$

We can now take a sum of the $2n$ utility functions to find that:

$$\sum_{i=1}^{n} J_i(\hat{\pi}, \hat{p}) + \bar{J}_i(\hat{\pi}, \hat{p}) \geq \sum_{i=1}^{n} \mathbb{E}_\sigma[J_i(\pi, p)] + \mathbb{E}_\sigma[\bar{J}_i(\pi, p)]$$

$$= \sum_{i=1}^{n} \left( \epsilon_i - \sum_{j \neq i} \xi_{j,i}^* \right) \mathbb{E}_\sigma[\nu_i(\pi_i)]$$

$$\geq \sum_{i=1}^{n} \left( \epsilon_i - \sum_{j \neq i} \xi_{j,i}^* \right) \nu_i(\hat{\pi}_i)$$

$$= \sum_{i=1}^{n} J_i(\hat{\pi}, \hat{p}) + \bar{J}_i(\hat{\pi}, \hat{p})$$

where we used the assumed condition on $\epsilon_i$ to guarantee convexity of the functions of $\pi_i$ in the second line, allowing us to use Jensen's inequality to derive the third line.

Thus we have shown that:

$$\sum_{i=1}^{n} J_i(\hat{\pi}, \hat{p}) + \bar{J}_i(\hat{\pi}, \hat{p}) = \sum_{i=1}^{n} \mathbb{E}_\sigma[J_i(\pi, p)] + \mathbb{E}_\sigma[\bar{J}_i(\pi, p)]$$

This implies that:

$$J_i(\hat{\pi}, \hat{p}) = \mathbb{E}_\sigma[J_i(\pi, p)] \le J_i(\pi_i', \hat{\pi}_{-i}, \hat{p}) \quad \forall \pi_i' \in \Delta_{A_i}$$
$$\bar{J}_i(\hat{\pi}, \hat{p}) = \mathbb{E}_\sigma[\bar{J}_i(\pi, p)] \le \bar{J}_i(\hat{\pi}, p_i', \hat{p}_{-i}) \quad \forall p_i' \in \mathcal{P}_{-i}.$$

Thus we have shown that $(\hat{\pi}, \hat{p})$ is a Nash equilibrium for the $2n$-player game. By invoking Proposition 1 we can observe that $\hat{\pi}$ must be a RQE for the risk-averse game. $\qquad\square$

# D  COMPUTATIONAL TRACTABILITY OF RQE IN ACTION-DEPENDENT RISK-AVERSE MATRIX GAMES

As in the case of aggregate risk-aversion, to prove our results we again introduce an auxiliary game that we relate to our risk-averse game of interest. In this case, the loss of the original players is given by:

$$J_i(\pi_i, \pi_{-i}, p) = \sum_{j \in \mathcal{A}_i} \pi_i(j) \left( -\langle R_{i,j}, p_{i,j} \rangle - D_i(p_{i,j}, \pi_{-i}) \right) + \epsilon_i \nu_i(\pi_i). \tag{28}$$

We now associate each player $i$ to its intermediate adversary $p_i$. For each player $p_i$ their loss function is given by:

$$\bar{J}_i(\pi, p_i) = \sum_{j \in \mathcal{A}_i} \pi_i(j) \left( \langle R_{i,j}, p_{i,j} \rangle + D_i(p_{i,j}, \pi_{-i}) \right) - \sum_k \xi_{i,k} \nu_k(\pi_k), \tag{29}$$

where $p_i = \{p_{i,j}\}_{j \in \mathcal{A}_i}$ where $p_{i,j} \in \mathcal{P}_{-i}$. This is once again a convex game since each player's loss is convex in its own argument. Define $\xi_{i,k}^* \ge 0$ as the minimum value of $\xi_{i,k}$ needed for $\bar{J}_i(\pi, p_i)$ to be concave in $\pi$ for all values of $p_i$ for all $i$. Note that due to the structure of $\bar{J}$ the values of $\xi_{i,k}^*$ only depend on properties of the risk metrics under consideration which are captured in $D_i$, and *not* on the payoff structure $R_i$.

The following proposition relates the Nash equilibrium of the $2n$-player convex game to the RQE of the action-dependent risk averse game.

**Proposition 2.** *If $(\pi^*, p^*)$ is a Nash equilibrium of the $2n$-player game, then $\pi^*$ is a RQE of Game (6). Furthermore, if $\pi^*$ is a RQE of the action-dependent risk averse game, then $(\pi^*, p^*)$ is a Nash equilibrium of the $2n$-player game, where:*

$$p_{i,j}^* = \arg \min_{p_{i,j} \in \mathcal{P}_{-i}} \bar{J}_{i,j}(\pi^*, p_{i,j}) \tag{30}$$

The proof of this result follows by exactly the same arguments as the proof of Proposition 1 and is therefore omitted for brevity.

Given these results we now prove the equivalent of Theorem 9 for the action-dependent formulation. We note that by a more careful accounting of terms, a stronger guarantee is possible in the 2-player regime similar to Theorem 3.

**Theorem 10.** *Assume the penalty functions that give rise to the players' risk preferences $D_i(\cdot, \cdot)$ are jointly convex in both their arguments. If $\sigma$ is a CCE of the $n$-player game and for each $i$:*

$$\epsilon_i \ge \sum_j \xi_{i,j}^*$$

*then the marginal strategies $\hat{\pi}_i = \mathbb{E}_\sigma[\pi_i]$ constitute a RQE of the action-dependent risk-averse game.*

Note that one point of departure for this result from the previous results is that $\sigma$ is now the CCE of the original 2-player convex game defined on the objective functions $f_1^{\epsilon_1}$ and $f_2^{\epsilon_2}$.

*Proof.* To begin, let $\sigma$ be a CCE of the convex game played on $f_i^{\epsilon_i}$, where:

$$f_i^{\epsilon_i}(\pi_i, \pi_{-i}) = \sum_{j \in \mathcal{A}_i} \pi_i(j) \left( \sup_{p_{i,j} \in \mathcal{P}_{-i}} -\langle R_{i,j}, p_{i,j} \rangle - D_i(p_{i,j}, \pi_{-i}) \right) + \epsilon_i \nu(\pi_i).$$

By definition of a CCE, we must have that:

$$\mathbb{E}_\sigma[f_i^{\epsilon_i}(\pi_i, \pi_{-i})] \le \mathbb{E}_\sigma[f_i^{\epsilon_i}(\pi_i', \pi_{-i})] \ \forall \pi_i' \in \Delta_{A_i},$$

for all $i = 1, ..., n$. Given $\sigma$, we can define a new distribution $\sigma'$ as $(\pi, p^*(\pi))$ where

$$p_i^*(\pi) = \arg\min_{p_i \in \mathcal{P}_{-i}} \bar{J}_i(\pi, p_i),$$

with $\xi_{i,j} = \xi_{i,j}^*$. We now claim that, by construction, $\sigma'$ is a CCE of the $2n$-player game. To see this, we observe that

$$\mathbb{E}_{\sigma'}[J_i(\pi_i, \pi_{-i}, p)] = \mathbb{E}_\sigma[f_i^{\epsilon_i}(\pi_i, \pi_{-i})] \le \mathbb{E}_\sigma[f_i^{\epsilon_i}(\pi_i', \pi_{-i})] = \mathbb{E}_{\sigma'}[J_i(\pi_i', \pi_{-i}, p)] \ \forall \pi_i \in \Delta_{A_i},$$

where the first equality follows by construction of $\sigma'$, the second from the definition of a CCE, and the third from the fact that the value of $p^*(\pi)$ only depends on $\pi_{-i}$ and not $\pi_i'$.

Similarly, we can show the same result for all the players $p_i$. By simply applying Jensen's inequality, we can see that

$$\mathbb{E}_{\sigma'}[\bar{J}_i(\pi, p_i)] = \mathbb{E}_{\pi \sim \sigma} \left[ \min_{p_{i,j}} \bar{J}_i(\pi, p_{i,j}) \right] \le \mathbb{E}_{\sigma'}[\bar{J}_i(\pi, p_i')] \ \forall p_i' \in \mathcal{P}_{-i}.$$

Thus we can observe that $\sigma'$ is a CCE of the $2n$-player game. To show that the marginals of $\pi$ in this CCE are Nash equilibria of the convex game (and thus RQE), we proceed as before. Using the fact that $D(p_{i,j}, \pi_{-i})$ is jointly convex in each of its arguments, we can apply Jensen's inequality to find that:

$$\mathbb{E}_{\sigma'}[J_i(\pi_i, \pi_{-i}, p)] \le J_i(\pi_i', \hat{\pi}_{-i}, \hat{p}) \ \forall \pi_i' \in \Delta_{A_i},$$

where $\hat{p}_{i,j} = \mathbb{E}_{\sigma'}[p_{i,j}]$ and $\hat{\pi}_i = \mathbb{E}_{\sigma'}[\pi_i]$. Similarly, by our choice of $\xi_{i,j} = \xi_{i,j}^*$ we can find that:

$$\mathbb{E}_{\sigma'}[\bar{J}_i(\pi, p_i)] \le \bar{J}_i(\hat{\pi}, p_i') \ \forall p_i' \in \mathcal{P}_{-i}.$$

Now, we can observe that:

$$\begin{aligned}
\sum_i J_i(\hat{\pi}_i, \hat{\pi}_{-i}, \hat{p}) + \bar{J}_i(\hat{\pi}, \hat{p}_i) &\ge \mathbb{E}_{\sigma'} \left[ \sum_i J_i(\pi_i, \pi_{-i}, p) + \bar{J}_i(\pi, p_i) \right] \\
&= \sum_i \mathbb{E}_{\sigma'} \left[ \left( \epsilon_i - \sum_j \xi_{i,j}^* \right) \nu_i(\pi_i) \right] \\
&\ge \sum_i \left( \epsilon_i - \sum_j \xi_{i,j}^* \right) \nu_i(\hat{\pi}_i) \\
&= \sum_i J_i(\hat{\pi}_i, \hat{\pi}_{-i}, \hat{p}) + \bar{J}_i(\hat{\pi}, \hat{p}_i),
\end{aligned}$$

where the third inequality follows by the Jensen's inequality. By the same rationale as in the proof of Theorem 9 we can now conclude that $\hat{\pi}$ is a Nash equilibrium-joint strategy profile-of the $2n$ player game and thus (due to Proposition 2) an RQE of the action-dependent risk averse game. $\square$

Similar to the aggregate risk regime, one can instantiate the previous theorem with specific risk-metrics and quantal response functions to illustrate the class of action-dependent RQE that are guaranteed to be computationally tractable. Importantly, this class once again only depends on the levels of risk averse and quantal response but *not* on the underlying structure of the game. However, note that the requirements are more stringent since the requirements on the $\xi^*$'s are stronger since they must ensure joint convexity of $\bar{J}_i$. Nevertheless further algebraic manipulations would allow one to recover analogues of Corollaries 8.1 and 8.2 for the action-dependent risk case as well. For brevity we leave these as exercises to the reader.

# E  PROOF OF THEOREM 5

Armed with the estimated reward and transition kernel in (15), we can construct an empirical MG $\widehat{\mathcal{MG}} = \{H, \mathcal{S}, \{\mathcal{A}_i\}_{i \in [n]}, \{\widehat{R}_{i,h}, \widehat{P}_{i,h}\}_{i \in [n], h \in [H]}\}$. Analogously, for any joint policy $\pi$, we denote the corresponding risk-averse loss functions or the payoff matrices as $\{\widehat{V}_{i,h}^{\epsilon_i}(\pi)\}$ and $\{\widehat{Q}_{i,h}^{\epsilon_i}(\pi)\}$, respectively.

To begin with, recall the goal is to show that

$$\max_{(i,s,h) \in [n] \times \mathcal{S} \times [H]} \left\{ V_{i,h}^{\epsilon_i}(\widehat{\pi}; s) - \min_{\pi_h': \mathcal{S} \mapsto \Delta_{\mathcal{A}_i}} V_{i,h}^{\epsilon_i}((\pi_h', \widehat{\pi}_{i,-h}) \times \widehat{\pi}_{-i}; s) \right\} \leq \delta. \tag{31}$$

For convenience, we denote

$$\forall (i,h,s) \in [n] \times [H] \times \mathcal{S}: \quad V_{i,h}^{\star,\epsilon_i}(\widehat{\pi}_{-i}; s) = \min_{\pi_{i,h}': \mathcal{S} \mapsto \Delta_{\mathcal{A}_i}} V_{i,h}^{\epsilon_i}((\pi_{i,h}', \widehat{\pi}_{i,-h}) \times \widehat{\pi}_{-i}; s) \tag{32}$$

and define the best-response policy $\pi_i^\star := \{\pi_{i,h}^\star : \mathcal{S} \mapsto \Delta_{\mathcal{A}_i}\}_{h \in [H]}$ so that

$$\forall (i,h,s) \in [n] \times [H] \times \mathcal{S}: \quad \pi_{i,h}^\star(s) := \operatorname{argmin}_{\pi_i': \mathcal{S} \mapsto \Delta_{\mathcal{A}_i}} V_{i,h}^{\epsilon_i}((\pi_{i,h}', \widehat{\pi}_{i,-h}) \times \widehat{\pi}_{-i}; s). \tag{33}$$

Then, for any $i \in [n]$, the gap $V_{i,1}^{\epsilon_i}(\widehat{\pi}) - V_{i,1}^{\star,\epsilon_i}(\widehat{\pi}_{-i})$ can be decomposed as follows:

$$V_{i,h}^{\epsilon_i}(\widehat{\pi}) - V_{i,h}^{\star,\epsilon_i}(\widehat{\pi}_{-i})$$

$$= V_{i,h}^{\epsilon_i}(\widehat{\pi}) - \widehat{V}_{i,h}^{\epsilon_i}(\widehat{\pi}) + \left( \widehat{V}_{i,h}^{\epsilon_i}(\widehat{\pi}) - \widehat{V}_{i,h}^{\epsilon_i}((\pi_{i,h}^\star, \widehat{\pi}_{i,-h}) \times \widehat{\pi}_{-i}) \right)$$

$$\quad + \left( \widehat{V}_{i,h}^{\epsilon_i}((\pi_{i,h}^\star, \widehat{\pi}_{i,-h}) \times \widehat{\pi}_{-i}) - V_{i,h}^{\epsilon_i}((\pi_{i,h}^\star, \widehat{\pi}_{i,-h}) \times \widehat{\pi}_{-i}) \right)$$

$$\leq V_{i,h}^{\epsilon_i}(\widehat{\pi}) - \widehat{V}_{i,h}^{\epsilon_i}(\widehat{\pi}) + \left( \widehat{V}_{i,h}^{\epsilon_i}((\pi_{i,h}^\star, \widehat{\pi}_{i,-h}) \times \widehat{\pi}_{-i}) - V_{i,h}^{\epsilon_i}((\pi_{i,h}^\star, \widehat{\pi}_{i,-h}) \times \widehat{\pi}_{-i}) \right)$$

$$\leq \|V_{i,h}^{\epsilon_i}(\widehat{\pi}) - \widehat{V}_{i,h}^{\epsilon_i}(\widehat{\pi})\|_\infty \mathbf{1} + \left\| \widehat{V}_{i,h}^{\epsilon_i}((\pi_{i,h}^\star, \widehat{\pi}_{i,-h}) \times \widehat{\pi}_{-i}) - V_{i,h}^{\epsilon_i}((\pi_{i,h}^\star, \widehat{\pi}_{i,-h}) \times \widehat{\pi}_{-i}) \right\|_\infty \mathbf{1} \tag{34}$$

where the first inequality holds by applying Theorem 4 with the estimated RAMG $\widehat{\mathcal{MG}}$ so that $\widehat{\pi}$ is a RQE of $\widehat{\mathcal{MG}}$, i.e.,

$$\widehat{V}_{i,h}^{\epsilon_i}(\widehat{\pi}) \leq \min_{\pi_{i,h}': \mathcal{S} \mapsto \Delta_{\mathcal{A}_i}} \widehat{V}_{i,h}^{\epsilon_i}((\pi_{i,h}', \widehat{\pi}_{i,-h}) \times \widehat{\pi}_{-i}) \leq \widehat{V}_{i,h}^{\epsilon_i}((\pi_{i,h}^\star, \widehat{\pi}_{i,-h}) \times \widehat{\pi}_{-i}). \tag{35}$$

To continue, we divide the proof into several key steps.

**Step 1: developing the recursion.** To control the two terms in (34), we consider that for any joint policy $\pi$ and time step $(s, h) \in \mathcal{S} \times [H]$,

$$V_{i,h}^{\epsilon_i}(\pi; s) - \widehat{V}_{i,h}^{\epsilon_i}(\pi; s)$$

$$\overset{\text{(i)}}{=} g_{\text{pol},i}^\pi (Q_{i,h}(\pi; s, :)) - g_{\text{pol},i}^\pi \left( \widehat{Q}_{i,h}(\pi; s, :) \right)$$

$$\overset{\text{(ii)}}{=} \sup_{p_i \in \mathcal{P}_i} -\pi_i(s)^T g_{\text{env},i} (R_{i,h}(s,:), P_{h,s,:}, V_{i,h+1}(\pi)) p_i - D_{\text{pol},i}(p_i, \pi_{-i}(s))$$

$$\quad - \left[ \sup_{p_i \in \mathcal{P}_i} -\pi_i(s)^T g_{\text{env},i} \left( R_{i,h}(s,:), \widehat{P}_{h,s,:}, \widehat{V}_{i,h+1}(\pi) \right) p_i - D_{\text{pol},i}(p_i, \pi_{-i}(s)) \right]$$

$$\leq \sup_{p_i \in \mathcal{P}_i} \left| -\pi_i(s)^T \left[ g_{\text{env},i} (R_{i,h}(s,:), P_{h,s,:}, V_{i,h+1}(\pi)) - g_{\text{env},i} \left( R_{i,h}(s,:), \widehat{P}_{h,s,:}, \widehat{V}_{i,h+1}(\pi) \right) \right] p_i \right|$$

$$\leq \left\| g_{\text{env},i} (R_{i,h}(s,:), P_{h,s,:}, V_{i,h+1}(\pi)) - g_{\text{env},i} \left( R_{i,h}(s,:), \widehat{P}_{h,s,:}, \widehat{V}_{i,h+1}(\pi) \right) \right\|_\infty, \tag{36}$$

where (i) holds by the definition in (11) and (10), and (ii) follows from the definition of $g_{\text{pol},i}^\pi(\cdot)$ in (9).

To continue, we know that for any $(s, \boldsymbol{a}) \in \mathcal{S} \times \mathcal{A}$:

$$
\left| g_{\mathsf{env},i}\left( R_{i,h}(s, \boldsymbol{a}), P_{h,s,\boldsymbol{a}}, V_{i,h+1}(\pi) \right) - g_{\mathsf{env},i}\left( R_{i,h}(s, :), \widehat{P}_{h,s,\boldsymbol{a}}, \widehat{V}_{i,h+1}(\pi) \right) \right|
$$

$$
= \left| R_{i,h}(s, \boldsymbol{a}) - \sup_{\widetilde{P} \in \Delta_S} -\widetilde{P} V_{i,h+1}(\pi) - D_{\mathsf{env},i}(\widetilde{P}, P_{h,s,\boldsymbol{a}}) \right.
$$

$$
\left. - \left( R_{i,h}(s, \boldsymbol{a}) - \sup_{\widetilde{P} \in \Delta_S} -\widetilde{P} \widehat{V}_{i,h+1}(\pi) - D_{\mathsf{env},i}(\widetilde{P}, \widehat{P}_{h,s,\boldsymbol{a}}) \right) \right|
$$

$$
= \left| - \sup_{\widetilde{P} \in \Delta_S} \left( -\widetilde{P} V_{i,h+1}(\pi) - D_{\mathsf{env},i}(\widetilde{P}, P_{h,s,\boldsymbol{a}}) \right) + \sup_{\widetilde{P} \in \Delta_S} \left( -\widetilde{P} \widehat{V}_{i,h+1}(\pi) - D_{\mathsf{env},i}(\widetilde{P}, \widehat{P}_{h,s,\boldsymbol{a}}) \right) \right|
$$

$$
\stackrel{(i)}{\leq} \sup_{\widetilde{P} \in \Delta_S} \left| - \left( -\widetilde{P} V_{i,h+1}(\pi) - D_{\mathsf{env},i}(\widetilde{P}, P_{h,s,\boldsymbol{a}}) \right) + \left( -\widetilde{P} \widehat{V}_{i,h+1}(\pi) - D_{\mathsf{env},i}(\widetilde{P}, \widehat{P}_{h,s,\boldsymbol{a}}) \right) \right|
$$

$$
\stackrel{(ii)}{=} \sup_{\widetilde{P} \in \Delta_S} \left| \widetilde{P} \left( V_{i,h+1}(\pi) - \widehat{V}_{i,h+1}(\pi) \right) \right| + \sup_{\widetilde{P} \in \Delta_S} \left| D_{\mathsf{env},i}(\widetilde{P}, P_{h,s,\boldsymbol{a}}) - D_{\mathsf{env},i}(\widetilde{P}, \widehat{P}_{h,s,\boldsymbol{a}}) \right|
$$

$$
\leq \left\| V_{i,h+1}(\pi) - \widehat{V}_{i,h+1}(\pi) \right\|_\infty + L \left\| P_{h,s,\boldsymbol{a}} - \widehat{P}_{h,s,\boldsymbol{a}} \right\|_1. \tag{37}
$$

where the first equality holds by (13), (i) holds by the supreme operator is 1-Lipschitz, and the last inequality holds by the assumption that $D_{\mathsf{env},i}(\cdot)$ is $L$-Lipschitz with respect to the $\ell_1$ norm for the second argument, with any fixed first argument. We mention an important note here. As remarked earlier, similar sample complexity guarantees can be shown when $\{D_{\mathsf{env},i}\}_{i \in [n]}$ are defined as $\phi$-divergence and allude to analyses ideas in (Panaganti and Kalathil, 2022; Xu* et al., 2023; Shi et al., 2023) to modify this step.

Plugging in (37) back to (36) and applying the results for all $s \in \mathcal{S}$ yields

$$
\left\| V_{i,h}^{\epsilon_i}(\pi) - \widehat{V}_{i,h}^{\epsilon_i}(\pi) \right\|_\infty \leq \left\| V_{i,h+1}(\pi) - \widehat{V}_{i,h+1}(\pi) \right\|_\infty + L \underbrace{\max_{(s,\boldsymbol{a}) \in \mathcal{S} \times \mathcal{A}} \left\| P_{h,s,\boldsymbol{a}} - \widehat{P}_{h,s,\boldsymbol{a}} \right\|_1}_{=:l_h}
$$

$$
= \left\| V_{i,h+1}^{\epsilon_i}(\pi) - \widehat{V}_{i,h+1}^{\epsilon_i}(\pi) \right\|_\infty + L l_h. \tag{38}
$$

Applying above fact recursively for $h, h+1, \cdots, H$, we arrive at

$$
\left\| V_{i,h}^{\epsilon_i}(\pi) - \widehat{V}_{i,h}^{\epsilon_i}(\pi) \right\|_\infty \leq \left\| V_{i,h+2}^{\epsilon_i}(\pi) - \widehat{V}_{i,h+2}^{\epsilon_i}(\pi) \right\|_\infty + L l_h + L l_{h+1}
$$

$$
\leq \cdots \leq \left\| V_{i,H+1}^{\epsilon_i}(\pi) - \widehat{V}_{i,H+1}^{\epsilon_i}(\pi) \right\|_\infty + L \sum_{t=h}^{H} l_t
$$

$$
\leq L \sum_{t=h}^{H} l_t, \tag{39}
$$

where the last inequality holds since $V_{i,H+1}^{\epsilon_i}(\pi) = \widehat{V}_{i,H+1}^{\epsilon_i}(\pi) = V_{i,h+1}(\pi) = \widehat{V}_{i,h+1}(\pi) = 0$ for any policy $\pi$.

**Step 2: controlling the errors $\{l_t\}$.** The remainder of the proof will focus on controlling (39). Applying (Auer et al., 2008, Lemma 17) over all $(h, s, \boldsymbol{a}) \in [H] \times \mathcal{S} \times \mathcal{A}$, we achieve the union bound that with probability at least $1 - \delta$,

$$
\max_{(h,s,\boldsymbol{a}) \in [H] \times \mathcal{S} \times \mathcal{A}} \left\| P_{h,s,\boldsymbol{a}} - \widehat{P}_{h,s,\boldsymbol{a}} \right\|_1 \leq \sqrt{\frac{14S}{N} \log\left( \frac{2S \prod_{i \in [n]} A_i H}{\delta} \right)}. \tag{40}
$$

Applying (40), we arrive at with probability at least $1 - \delta$,

$$
\forall h \in [H], \; l_h = \max_{(s,\boldsymbol{a}) \in \mathcal{S} \times \mathcal{A}} \left\| P_{h,s,\boldsymbol{a}} - \widehat{P}_{h,s,\boldsymbol{a}} \right\|_1 \leq \sqrt{\frac{14S}{N} \log\left( \frac{2S \prod_{i \in [n]} A_i H}{\delta} \right)}. \tag{41}
$$

Inserting (41) into (39) gives

$$\left\| V_{i,h}^{\epsilon_i}(\pi) - \widehat{V}_{i,h}^{\epsilon_i}(\pi) \right\|_\infty \le L \sum_{t=h}^{H} l_t \le HL \sqrt{\frac{14S}{N} \log\left(\frac{2S \prod_{i \in [n]} A_i H}{\delta}\right)}. \tag{42}$$

Finally, recalling (34) yields

$$V_{i,1}^{\epsilon_i}(\widehat{\pi}) - V_{i,1}^{\star,\epsilon_i}(\widehat{\pi}_{-i}) \le \| V_{i,h}^{\epsilon_i}(\widehat{\pi}) - \widehat{V}_{i,h}^{\epsilon_i}(\widehat{\pi}) \|_\infty \mathbf{1} \tag{43}$$

$$+ \left\| \widehat{V}_{i,h}^{\epsilon_i}((\pi_{i,h}^\star, \widehat{\pi}_{i,-h}) \times \widehat{\pi}_{-i}) - V_{i,h}^{\epsilon_i}((\pi_{i,h}^\star, \widehat{\pi}_{i,-h}) \times \widehat{\pi}_{-i}) \right\|_\infty \mathbf{1}$$

$$\le 8HL \sqrt{\frac{S}{N} \log\left(\frac{2S \prod_{i \in [n]} A_i H}{\delta}\right)} \mathbf{1}, \tag{44}$$

where the last inequality holds by applying (42) with $\pi = \widehat{\pi}$ or $\pi = (\pi_{i,h}^\star, \widehat{\pi}_{i,-h}) \times \widehat{\pi}_{-i}$. □

## F EXPERIMENT DETAILS

We provide more results and details of our experiments in this section. We provide our code in the following **Github link** https://github.com/kishanpb/Risk-averse-Quantal-Equilibria that contains instructions to reproduce all results in this paper.

### F.1 MATRIX GAMES

| Game 1 | | | | Game 2 | | | | Game 3 | | |
|---|---|---|---|---|---|---|---|---|---|---|
| | L | R | | | L | R | | | L | R |
| U | 10 10 | 0 18 | | U | 9 4 | 0 13 | | U | 8 6 | 0 14 |
| D | 9 9 | 10 8 | | D | 6 7 | 8 5 | | D | 7 7 | 10 4 |
| **Game 4** | | | | **Game 5** | | | | **Game 6** | | |
| | L | R | | | L | R | | | L | R |
| U | 7 4 | 0 11 | | U | 7 2 | 0 9 | | U | 7 1 | 1 7 |
| D | 5 6 | 9 2 | | D | 4 5 | 8 1 | | D | 3 5 | 8 0 |
| **Game 7** | | | | **Game 8** | | | | **Game 9** | | |
| | L | R | | | L | R | | | L | R |
| U | 10 12 | 4 22 | | U | 9 7 | 3 16 | | U | 8 9 | 3 17 |
| D | 9 9 | 14 8 | | D | 6 7 | 11 5 | | D | 7 7 | 13 4 |
| **Game 10** | | | | **Game 11** | | | | **Game 12** | | |
| | L | R | | | L | R | | | L | R |
| U | 7 6 | 2 13 | | U | 7 4 | 2 11 | | U | 7 3 | 3 9 |
| D | 5 6 | 11 2 | | D | 4 5 | 10 1 | | D | 3 5 | 10 0 |

Table 2: Payoff Matrices from (Selten and Chmura, 2008). In each column, the number above (below) is the payoff for player 1 (player 2).

|   | L | R |
|---|---|---|
| U | 200 | 160 |
|   | 160 | 10 |
| D | 370 | 10 |
|   | 200 | 370 |

Table 3: Game 4 Payoff Matrix from (Goeree et al., 2003). In each column, the number above (below) is the payoff for player 1 (player 2).

**Matching pennies matrix games:** Two players in matching pennies simultaneously choose heads or tails, and a player wins (the other player loses) a payoff if their choices *match*. The detailed payoff matrices are provided in Tables 2 and 3. These games from (Goeree et al., 2003; Selten and Chmura, 2008) are focused on showcasing risk-averse solutions by developing payoffs strategically such that either player deviating from their choice of plays will cause hefty damage in terms of payoff to the other player. So, both players prefer a 'safer' choice of plays, thus highlighting both risk-averse and bounded rational preferences.

**Algorithm and Result details:** We use the penalty functions KL and reverse KL with the regularizers log barrier and negative entropy, as mentioned in Table 1, for our experiments. To be precise, we consider the following two experimental setups: (a) For $j = 1, 2$, we let $\nu_j(p) = -\sum_i \log(p_i)$ and $D_j(p, q) = (1/\tau_j)\text{KL}(p, q)$. (b) For $j = 1, 2$, we let $\nu_j(p) = \sum_i p_i \log(p_i)$ and $D_j(p, q) = (1/\tau_j)\text{KL}(q, p)$. We note that the parameters $\tau_j^{-1}$ play the same role as $\xi_j$'s (formalized in Corollaries 8.1 and 8.2), i.e., the players take more risk neutral decisions as $\tau_j \to 0$. With the perfect information, we use the vanilla projected gradient descent with constant stepsizes (Beck, 2017) as the no-regret algorithm to arrive at the CCE $\sigma$ of the four player game. We use consistent stepsizes between $10^{-4}$ and $10^{-3}$ for all our runs iterating through $10^4$ steps of gradient descent. We notice 2% deviations in sup-norm for the algorithm policies in about 20 runs.

## F.2 MARKOV GAMES

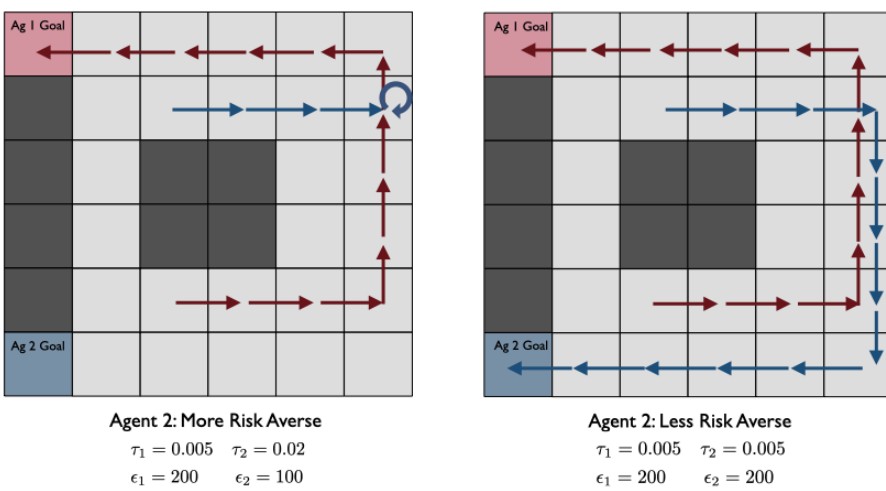

Figure 3: Cliff-Walk results for the $\ell_1$ environmental uncertainty metric.

**Algorithm details:** We use the penalty function KL with the log-barrier regularizer, as mentioned in Table 1 and Example 2, for our experiments. We also use KL divergence for the environmental uncertainty metric $D_{\text{env},i}$ with the same bounded rational parameter $\tau_i^{-1}$ used by RQE. We use the full information of this grid-world to evaluate the Q-values described in Algorithm 1. Additionally, as we use the matrix games solver for RQE, we have similar statistical deviations for different training runs. In Appendix F.2, we also showcase environmental uncertainty results with $\ell_1$ metric.

Here, we showcase results corresponding to $\ell_1$ metric for the environmental uncertainty metric $D_{\mathrm{env},i}$ for the grid-world problem. We consider the same *Cliff-Walk* environment described in Section 4.3 under some minor modifications discussed here. The rewards for falling into the cliff is now $-100$ and for reaching the goal is 20. Agents/players are rewarded a small negative reward of $-0.1$ for taking each step. The episode horizon $H$ is 100. The algorithm details is the same as described in Section 4.3.

We present two results in Fig. 3 that has similar implications as in Fig. 2. We make one important observation about these agents equipped with $\ell_1$ environmental uncertainty metric. These agents are less risk-averse towards the cliff in horizontal grid axis compared to the results Fig. 2 corresponding to the KL environmental uncertainty metric. We do not investigate the effects of different environmental uncertainty metrics in this work and postpone to address in future research directions.