# OpenReview forum: "Tractable Multi-Agent Reinforcement Learning through Behavioral Economics"
_ICLR.cc/2025/Conference — ICLR 2025 Oral_

### Official Review · Reviewer_JYbT · 2024-10-27

**Soundness:** 3
**Presentation:** 3
**Contribution:** 2
**Rating:** 8
**Confidence:** 3

**Summary:**

The authors propose an algorithm for multi-agent game and reinforcement learning inspired by behavioral economics, with risk aversion and bounded rationality. They study a new class of equilibria called risk-averse quantal response equilibria (RQE), which is computationally tractable and with more human characteristics. Moreover, it provides an analysis of sample complexity for computing RQE in finite-horizon Markov games using generative models

Especially, these equilibria are independent of specific game structures and instead hinge on the degree of agents' risk aversion and bounded rationality. The authors also validate the effectiveness of RQE by showing its alignment with human behavior in experimental game studies.

**Strengths:**

- The authors introduces games with risk-averse agents and bounded rationality, allowing for the computation of risk-averse quantal response equilibrium (RQE), which is realistic and computationally feasible.
- RQE can be computed in polynomial time in all $n$-player matrix games under specific risk aversion and bounded rationality conditions, irrespective of the underlying payoffs.
- The authors present a sampling-based algorithm with finite-sample guarantees for learning approximate RQE in unknown environments.

**Weaknesses:**

- While the paper applies duality and risk aversion concepts, these techniques have been previously explored in the context of game theory and multi-agent reinforcement learning, raising questions about the technical novelty of the contributions.
- Extending the findings to finite-horizon Markov Decision Processes (MDPs) may be somewhat straightforward, as the process involves relatively conventional methods. While this extension adds scope, it may not significantly deepen the theoretical insights or complexity of the approach.
- The proposed algorithm requires access to a simulator, which is a strong assumption and may limit the practical applicability of the method, especially in real-world scenarios where such simulators are unavailable or costly to obtain.

**Questions:**

- Typo on L161: To allow agents to have risk preferences we make u a general class of convex
- What challenges arise in adopting conditions weaker than requiring a simulator?

---

> ### Author Response · Authors · 2024-11-24
> **Reply to Reviewer JYbT: Part One**
>
> We sincerely thank the reviewer for acknowledging our contributions and providing constructive suggestions.
>
> > **Q1** What is the technical contribution? Duality and risk aversion concepts have been previously explored in game theory and multi-agent reinforcement learning (MARL).
>
>
>
> **A1** Thanks for raising this insightful question. From the view of risk-averse games with duality (both matrix games and Markov games, i.e., MARL), this work propose a new solution concept --- a natural equilibrium (RQE) for risk-averse games that is tractable to be computed for all finite games and finite-horizon Markov games (MARL).  **Risk aversion on its own gives rise to a convex game but cannot guarantee computational tractabiltiy** of the resulting equilibrium notion (risk-averse Nash equilibria).
>
>
> The key is to combine risk-aversion with **bounded rationality**, another important observation in behavirol economics. Bounded rationality on its own is also well known to help with computational tractability (as is pointed out by reviewer BGUx), but the degree of bounded rationality required to ensure this holds depends on the game structure (i.e., the relationship between players' payoffs and their magnitude). In contrast, the class of RQE we prove is computationally tractable is completely independent of the game structure. Furthermore, RQE is expressive in that it can recreate peoples' patterns of play in experimental data as observed in Figure 1.
>
> Two benefits distinguish RQEs from previous other equilibrium notions such as Nash equlibrium and Quantal Response Equilibriums (only consider bounded rationality), offering a fresh perspective on learning games with tractability: 1) predictive of human decision-making; 2) guarantees for computationally tractability. To the best of our knowledge, this is the first result of its kind and while we only scratch the surface of their analysis we are excited at the protential that these equilibria represent for designing principled algorithms for multi-agent reinforcement learning.
> We achieve this by incorporatin another observation from behaviral economics that
>
>
>
>
> > **Q2** Extending the findings to finite-horizon Markov Decision Processes (MDPs) may be somewhat straightforward, as the process involves relatively conventional methods. While this extension adds scope, it may not significantly deepen the theoretical insights or complexity of the approach.
> >
>
>
> **A2** The reviewer is correct that much of the technical contributions on computational tractability primarily focus on matrix games. However, introducing uncertainty in environment dynamics, as Markov games are sequential in nature, requires new techniques from high-dimensional statistics to address the randomness in dynamics when the environment is unknown. In Section 4, extending one-shot games to the finite-horizon setting significantly broadens the scope of this work, encompassing sequential decision-making and games with unknown structures. This extension offers a fresh perspective by introducing a new equilibrium concept for learning MARL with tractability, expanding its relevance beyond the game theory and economics community to broader fields such as machine learning and reinforcement learning.

---

> ### Author Response · Authors · 2024-11-24
> **Reply to Reviewer JYbT: Part Two**
>
> > **Q3** Questions about the sampling mechanism --- a simulator.
> > * The proposed algorithm requires access to a simulator, which is a strong assumption and may limit the practical applicability of the method, especially in real-world scenarios where such simulators are unavailable or costly to obtain.
> > * What challenges arise in adopting conditions weaker than requiring a simulator?
>
> **A3** We appreciate this insightful question.
>
> * **Simulator (a generative model) is a widely-used fundamental setting in RL and MARL literature, and even this setting is heavily understudied when considering additional risk-aversion or robustness.** Sampling through a simulator is a fundamental setting with a long line of research for decades  in RL ranging from all kinds of single-agent RL and multi-agent RL problems [1-2]. The widely studied data collection settings can be categoried into  "simulator" settings, "online," and "offline". Simulator setting plays an essential role in shaping the theoretical foundation of RL and are critically needed before addressing more complex or practical sampling settings (such as online).
>
>     However, even the fundamental simulator setting for robust MARL is understudeid and open with only a few prior works [3-4]. This motivates us to take a step in this open question, where our theoretical findings are potential to be extended to more settings such as online, offline, or function approximation robust MARL.
> * The more complex setting introduces additional challenges, such as when doing offline sampling, we encounter insufficient coverage of the entire state and action spaces in offline scenarios [3]; and when doing online sampling, we need to balance the trade-off between exploitation and exploration during the interactive learning process, where data collection and agent strategy updates occur alternately.
>
>
> >> [1] Clavier, Pierre, Erwan Le Pennec, and Matthieu Geist. "Towards minimax optimality of model-based robust reinforcement learning." arXiv preprint arXiv:2302.05372 (2023).\
> >> [2] Kearns, M., & Singh, S. (1998). Finite-sample convergence rates for Q-learning and indirect algorithms. Advances in neural information processing systems, 11.\
> >> [3] Jose Blanchet, Miao Lu, Tong Zhang, and Han Zhong. Double pessimism is provably efficient
> for distributionally robust offline reinforcement learning: Generic algorithm and robust partial coverage. Advances in Neural Information Processing Systems, 36, 2024\
> [4] Laixi Shi, Eric Mazumdar, Yuejie Chi, and Adam Wierman. Sample-efficient robust multi-agent reinforcement learning in the face of environmental uncertainty. In Forty-first International Conference on Machine Learning, 2024
>
> > **Q4** Typo on L161: To allow agents to have risk preferences we make u a general class of convex
>
> **A4** Thank you so much for the detailed review. We will fix it in the updated revision as: 'we make a general class...'

---

> > ### Author Response · Authors · 2024-11-26
> >
> > Dear Reviewer JYbT,
> >
> > We extend our gratitude for dedicating your valuable time to reviewing our paper. Your insightful suggestions have greatly contributed to enhancing the quality of our work, and we eagerly anticipate the possibility of receiving further feedback from you. **As the reviewers-authors discussion period has been graciously extended by the ICLR committee (Dec 2/3 AoE), we kindly request your reply to our responses to ensure that all your concerns have been adequately addressed in our initial rebuttal response.** Moreover, we remain enthusiastic about engaging in further discussions to address any additional questions or considerations you may have.
> >
> > Best regards,
> >
> > Authors

---

> > > ### Comment · Reviewer_JYbT · 2024-11-28
> > >
> > > Thanks for the detailed clarification! The authors' response addressed my questions well, and I'm raising my score to 8.

---

> > > > ### Author Response · Authors · 2024-12-01
> > > >
> > > > Dear Reviewer JYbT,
> > > >
> > > > Thanks so much for your efforts in reviewing our paper and your positive score/feedback! We will further improve our paper following your suggestions during revision.
> > > >
> > > > Thanks,
> > > >
> > > > Authors

---

### Official Review · Reviewer_E3pZ · 2024-11-03

**Soundness:** 3
**Presentation:** 4
**Contribution:** 3
**Rating:** 8
**Confidence:** 3

**Summary:**

This paper discusses multi-agent learning in a setting where the agents are risk-averse. By introducing risk aversion and quantal response into the agents' behaviour, the authors simplify computations to achieve a risk-averse quantal response equilibrium (RQE). This is accomplished via some clever tooling, applying a dual representation from [1] to convert the problem into a convex optimization problem under certain values of risk aversion. The trade-off is that it introduces n additional players into the game. The theory from the repeated game setting is then extended to the Markov game setting, where it establishes equivalence to the repeated game setting, quantifies the learning error, and implies convergence to RQE under specific parameter settings. A toy example is presented to convince the reader that the technology is functional.

[1] Föllmer, Hans, and Alexander Schied. "Convex measures of risk and trading constraints." Finance and stochastics 6 (2002): 429-447.

**Strengths:**

- The authors perform a rigorous analysis and provide convincing arguments for a well-defined system.
- The work could have real economic applications.
- There is good novelty in applying the dual formulation of [1] to this multi-agent setting and constructing the loss functions in Eq. (6) and (7).

**Weaknesses:**

- The notation could be a bit confusing at times; the difference between $\pi$ and $p_i$ is not clearly defined and can be hard for the reader to follow.
- Typo in line 160: "we make u a general class..."
- The experiments lack breadth in demonstrating the algorithm’s efficacy across diverse problem settings. Only one cliff walk example is demonstrated, and very standard simple benchmarks are used for the repeated game.

**Questions:**

- Could we not also have a baseline comparison with standard algorithms, such as Nash Q-learning in the MG, or other MARL algorithms (e.g., multi-agent PPO), to assess its performance against established algorithms?
- Similarly, could we extend the MG setting to more than two players in the toy examples? This could significantly increase computation time, even with the proposed, more efficient, algorithm.
- How would this accommodate non-convex compositions of risk-averse functions? Is there a simple extension, or would it be more complex?
- What economic scenarios could the RAMG apply to, given that the concept of risk aversion originally stems from economics?

---

> ### Author Response · Authors · 2024-11-24
> **Reply to Reviewer E3pZ: Part One**
>
> We sincerely appreciate reviewer E3pZ for recognizing the contributions of this work and providing various insightful suggestions.
>
> > **Q1.** Suggestions on presentation
> > * The difference between $\pi$ and $p_i$ is not clearly defined.
> > * Typo in line 160: "we make u a general class..."
>
>
>
> **A1.** Thanks for raising these points. We will specify them in the updated version. For the reviewer's convenience, $\pi$ is usually represent a possible $\pi_{-i} \in \Delta_{\mathcal{A}_{-i}}$, an element in the joint action space of all agents except the $i$-th agent.
>
>
>
> > **Q2.** Suggestions on the experimental results
> > * The experiments lack breadth in demonstrating the algorithm’s efficacy across diverse problem settings. Only one cliff walk example is demonstrated, and very standard simple benchmarks are used for the repeated game.
> > * Could we not also have a baseline comparison with standard algorithms, such as Nash Q-learning in the MG, or other MARL algorithms (e.g., multi-agent PPO), to assess its performance against established algorithms?
> > * Similarly, could we extend the MG setting to more than two players in the toy examples? This could significantly increase computation time, even with the proposed, more efficient, algorithm.
>
>
> **A2** Thanks for raising this question for the experiments.
> Even though we are inspired by the concept of bounded rationality from behavioral economics, we want to highlight that this is used to rather enable the computation tractability of the risk-averse quantal equilibria solution concept. Thus, we believe risk-averseness of agents' behavior plays critical role in the **cliff-walking environment** we built and in the standard **matrix-games promoting risk-averse solutions**. We emphasize that our algorithms uses full information about the environments to reach the solutions shown in the three figures; thus, our evaluation criteria is more about showcasing risk-averseness than comparing performances in the learning setting. We validated our risk-averseness and bounded rational parameters' conditions in our Corollary 7.1-7.2. We believe our results on cliff-walking example provide important insights for **future scalable RL algorithm** development for large-scale environment experiments. These are important future directions that are of interest in the multi-agent RL community.
>
> > **Q3.** How would this accommodate non-convex compositions of risk-averse functions? Is there a simple extension, or would it be more complex?
>
>
> **A3** Thank you for raising this point. We focus on convex risk metrics for our risk-averse functions, motivated by the advantages of their dual representation, which simplifies the inner optimization problem by introducing a regularization term [1]. However, our results cannot directly extend to non-convex risk-averse functions (won't guarantee a dual representation), as non-convex optimization is generally more challenging and does not guarantee convergence to the global optimum. Exploring non-convex risk metrics is an intriguing direction for future research.
>
> >> [1] Föllmer, Hans, and Alexander Schied. "Convex measures of risk and trading constraints." Finance and stochastics 6 (2002): 429-447.
>
> > **Q4.** What economic scenarios could the RAMG apply to, given that the concept of risk aversion originally stems from economics?
>
>
> **A4**  Thanks for raising this important question. RAMGs are of widely interests not only in behavorial economics, but also in optimal control, operations research, finance, neuroscience, etc.
>
> >> [1] Mihatsch, Oliver, and Ralph Neuneier. "Risk-sensitive reinforcement learning." Machine learning 49 (2002): 267-290.

---

> > ### Comment · Reviewer_E3pZ · 2024-11-25
> >
> > I thank the author's for their insightful response. My positive score remains the same.

---

> > > ### Author Response · Authors · 2024-11-26
> > >
> > > Dear Reviewer E3pZ,
> > >
> > > Thanks so much for your efforts in reviewing our paper and your positive feedback! We will further improve our paper following your suggestions during revision.
> > >
> > > Thanks,
> > >
> > > Authors

---

### Official Review · Reviewer_GTtm · 2024-11-04

**Soundness:** 3
**Presentation:** 4
**Contribution:** 3
**Rating:** 8
**Confidence:** 4

**Summary:**

The paper investigates incorporating risk aversion and bounded rationality in order to tractably compute risk-averse quantal response equilibria (RQE) for n-player matrix and finite-horizon Markov games. The paper proposes computing the RQE is advantageous because it is tractable under certain conditions independent of the game structure and it models human behavior better. The conditions for tractability of RQE is stated with respect to degree of risk aversion and degree of bounded rationality. The connection between using no-regret learning algorithms and computing the coarse correlated equilibrium (CCE) of the 2n-player game and computing the RQE of the n-player game is provided. The method and the definitions are extended to Markov games and the experiment results on a toy Cliff Walk grid-world game is given which support the claims of the paper.

**Strengths:**

* Novel contribution while bringing a theoretically grounded view on risk aversion and bounded rationality to Markov games which can eventually be applied to more complex MARL scenarios. Adjusting the games so that RQE is achieved tractably by no-regret learning and having the tractability not depend on the underlying game structure is useful.
* Claims are supported extensively by proofs.
* The experiment on Cliff Walk demonstrates the risk averseness well.
* Paper is well written in general.

**Weaknesses:**

* There is only one example game for Markov game results and it is very toy.
* Bounded rationality is less obvious than risk averseness on Cliff Walk experiments. Interpreting bounded rationality from the experiments is difficult.
* Paper has some minor typos (example line 160).

**Questions:**

* What is the significance of the choice of aggregate risk aversion and action-dependent risk aversion? When should we pick either one?
* Is there any advantage of converging to the RQE rather than optimizing to the optimal solutions, since the policies are to be used by artificial agents and the purpose is not to model human behavior? Is it only to be used because it is more tractable or do you foresee another significant use for it due to risk aversion and bounded rationality modeling?

---

> ### Author Response · Authors · 2024-11-24
> **Reply to Reviewer GTtm: Part One**
>
> We sincerely appreciate the reviewer’s thoughtful review and recognition of our work’s theoretical and experimental contributions.
>
> > **Q1.** Suggestions on the experimental results**
> > * There is only one toy example game for Markov game results.
> > * Bounded rationality is less obvious than risk averseness on Cliff Walk experiments. Interpreting bounded rationality from the experiments is difficult.
>
> **A1:** Thanks for raising this question. Even though we are inspired by the concept of bounded rationality from behavioral economics, we want to highlight that this is used to rather enable the computation tractability of the risk-averse quantal equilibria solution concept. Thus, we believe risk-averseness of agents' behavior takes critical role in the cliff-walking environment we built. We emphasize that our multi-agent Markov game algorithm uses full-information of the environment to reach at the solutions shown in the two figures. We validated our risk-averseness and bounded rational parameters' conditions in our Corollary 7.1-7.2. We believe our results on cliff-walking example provide important insights for future scalable algorithm development for large-scale environment experiments. These are important future directions that is of interest in the multi-agent RL community.
>
>
> > **Q2.** What is the significance of the choice of aggregate risk aversion and action-dependent risk aversion? When should we pick either one?
>
> **A2:**
>
>
> Thanks for this question. Using the same $\rho_{i,\pi_{-i}}(\cdot)$，the Aggregate Risk Aversion formulation (2) is more conservative (always not bigger) than Action-dependent Risk Aversion formulation (3), follows by the Jensen's inequality. Intuitively this means the Aggregate Risk Aversion formulation in (2) considers the regularized worst-case when the agent i let the 'adversarial' know its strategy during the attack. This additional information can help the 'adversarial' to choose a harder case for agent i than the latter formulation in (3) where the 'adversarial' won't know agent i's policy. Hence, it is left to practitioners to choose the appropriate conservatism they need for their applications -- for e.g. (2) in communication games where adversarial agents often get agents' information and (3) in traffic networks where adversarial agents is the nature affecting the weather and so on.

---

> > ### Author Response · Authors · 2024-11-24
> > **Reply to Reviewer GTtm: Part Two**
> >
> > > **Q3.** Is there any advantage of RQE rather than the original optimal solutions without considering risk aversion and bounded rationality modeling? Since the policies are to be used by artificial agents and the purpose is not to model human behavior? Is it only to be used because it is more tractable or do you foresee another significant use for it due to risk aversion and bounded rationality modeling?
> >
> >
> > **A3:** Thanks for raising this important question! **Strategive games involving human are ubiquitously** in human society and engineering sytems, such as finance and friendship, autonomous driving, and human-robotics interactions. So it brings us to consider realistic games with human decision-making features.
> >
> > Our goal in this work was to understand the impact of introducing features of human decision-making from behavioral economics into problems of learning in games and multi-agent reinforcement learning. Surprisingly, we were able to show that the **cominbation** of bounded rationality and risk-aversion gives rise to a single class of equilibria that are computationally tractable across all n-player matrix games and finite-horizon Markov games.
> >
> > * **The advantages of the proposed RQE compard to prior works: predictive of human decision-making and computationally tractable.** Crucially this result requires both bounded rationality and risk aversion to go through. Risk aversion on its own gives rise to a convex game but cannot guarantee computational tractabiltiy of the resulting equilibrium notion (risk-averse Nash equilibria). Bounded rationality on its own is well known to help with computational tractability (as is pointed out by reviewer BGUx), but the degree of bounded rationality required to ensure this holds depends on the game structure (i.e., the relationship between players' payoffs and their magnitude). In contrast, the class of RQE we prove is computationally tractable is completely independent of the game structure. Furthermore, this class is expressive in that it can recreate peoples' patterns of play in experimental data as observed in Figure 1.
> >
> >
> >     These two benefits distinguish RQEs from previous other equilibrium notions such as Nash equlibrium and Quantal Response Equilibriums (only consider bounded rationality), offering a fresh perspective on learning games with tractability. **To the best of our knowledge, this is the first result of its kind and while we only scratch the surface of their analysis we are excited at the protential that these equilibria represent for designing principled algorithms for multi-agent reinforcement learning.**
> >
> > > **Q4.** Paper has some minor typos (example line 160).
> >
> > **A4** Thank you. We will fix it in a revision as: 'we make a general class...'

---

> > > ### Author Response · Authors · 2024-11-26
> > >
> > > Dear Reviewer GTtm,
> > >
> > > **We extend our gratitude for dedicating your valuable time to reviewing our paper and your positive score.** Your insightful suggestions have greatly contributed to enhancing the quality of our work, and we eagerly anticipate the possibility of receiving further feedback from you. **As the reviewers-authors discussion period has been graciously extended by the ICLR committee (Dec 2/3 AoE), we kindly request your reply to our responses to ensure that all your concerns have been adequately addressed in our initial rebuttal response.** Moreover, we remain enthusiastic about engaging in further discussions to address any additional questions or considerations you may have.
> > >
> > > Best regards,
> > >
> > > Authors

---

> > > > ### Comment · Reviewer_GTtm · 2024-11-27
> > > >
> > > > I thank the authors for all the insightful clarifications. My concerns have been well addressed, and I am keeping my positive score.

---

> > > > > ### Author Response · Authors · 2024-12-01
> > > > >
> > > > > Dear Reviewer GTtm,
> > > > >
> > > > > Thanks so much for your efforts in reviewing our paper and your positive score/feedback! We will further improve our paper following your suggestions during revision.
> > > > >
> > > > > Thanks,
> > > > >
> > > > > Authors

---

### Official Review · Reviewer_BGux · 2024-11-05

**Soundness:** 3
**Presentation:** 2
**Contribution:** 3
**Rating:** 8
**Confidence:** 4

**Summary:**

Motivated by behavioral economics, this work proposes to investigate the combination of bounded rationality and risk aversion to define a regularized game leading to a new risk averse nash equilibrium concept. The main motivation of the paper is to introduce tractable equilibria in all n-player matrix and finite horizon Markov games. Notably, the introduced class of games is ‘tractable’ independently of the game structure while only depending on both levels of bounded rationality and risk aversion. The paper further proposes illustrations to motivate the games from the behavioral economics viewpoint and performs a few simulations to test the algorithms on simple examples. A sample complexity for finite Markov games under the generative model further completes the theoretical results.

**Strengths:**

- Writing is clear overall (I have some concerns for some aspects though, see next section) and the paper is well-organized. The proofs are well exposed in the appendix.
- Investigating the tractability of games combining bounded rationality and risk aversion is interesting. In particular I find the idea of achieving tractability independently from the payoffs interesting.
- The paper covers matrix games and their extension to finite horizon Markov games. More importantly the results hold for general sum games.
- Modelling the behavior of decision making agents and their strategic interaction beyond expected utility theory is interesting on its own.

**Weaknesses:**

- My main concern about this paper is the following question: What exactly makes the introduced games tractable? The paper argues that it is the combination of bounded rationality and risk aversion. I find the focus on tractability rather confusing.
Risk aversion has its own motivation but it is unclear if it is required here for tractability. To be precise, I think the contribution in this paper is that you can achieve tractability (for computing another solution concept  than NE/CCE relating to a modified game) independently of the knowledge of payoff functions (their bounds I guess). More precisely, I think bounded rationality is already enough to allow for tractability but it should require some assumptions on the payoffs which this paper avoids. Please correct me if I have missed something here. The paper does not exactly present the contribution this way. In particular, I think the following statements would gain in clarity:

-- l. 111-112: ‘RQE not only incorporates features of human decision-making but are also more amenable to computation than QRE …’ Why? There is no extended discussion about this whereas I believe it is essential to clarify the contributions.

-- l. 235-238: While risk aversion might not be enough, I think bounded rationality should be enough (e.g. QRE as demonstrated in a number of works) under relevant assumptions on the structure of the game (mainly conditions on the bounded rationality level compared to the magnitude of the utilities/rewards). I find the presentation order confusing here. You start by introducing risk aversion and show its shortcomings as for the tractability of computing equilibria in such risk averse games to motivate the need of something else but equilibria should become tractable only with bounded rationality.

-- From the title and the abstract (and throughout the paper), the emphasis is on tractability, as if both are actually needed to ‘achieve’ tractability. If this is not the case as bounded rationality is the main enabler, I would maybe highlight less this computational aspect and clarify that the combination brings you the possibility to achieve tractability under a condition on both levels of bounded rationality and risk aversion which is independent of the rewards. How does this condition compare to the one for say QREs? What’s the price to pay and advantages (if any) for RQNEs compared to QREs (beyond the advantage of incorporating risk aversion)?

- Equilibrium collapse is briefly mentioned in l. 294. I would like to see a more in-depth description of how the results of this paper connect to the prior work regarding this phenomenon. It seems that the extended game you introduce seems to have some zero sum structure under a perhaps more stringent assumption than what you set (\epsilon_1 = \xi_{2,1} and \epsilon_2 = \xi_{1,2}, a condition satisfying your condition), under which J_1 + J_2 + \bar{J}_1 + \bar{J}_2 = 0. This can be seen from setting \lambda = 1/2 in your proof of Theorem 7 p. 21, l. 1133-1146. Is this an observation that connects to prior work about zero sum games and equilibrium collapse? It is also worth mentioning that setting \lambda = 1/2 leads to a condition that is consistent with your extension to the n-player setting (l. 1226-1232).

- I think the introduction and motivation of (6) and (7) needs some improvement. Could you give more motivation and intuition?

- The notations $\xi_1^*, \xi_2^*$ are quite confusing, especially that you also use the notations $\tau_1, \tau_2$ in Fig. 1 without defining them in the main part. What do you mean precisely by $\tau_1, \tau_2$ satisfy the conditions of Theorem 2 (l. 328)? How do the \tau s and \xi s relate?I believe one issue in the presentation is that you chose to hide \tau in the definition of the regularizer D without putting it in front of D (as it appears from the appendix). Maybe a solution is just to add the examples to the main part to clarify this and further comment on the relationship between the \xi s and \tau s.

- Comments about experiments: While the paper focuses on RQREs, experiments do not seem to illustrate the concept. What are the RQREs in this game and why? Are they different from NEs? Why would they be more reasonable?

- The introduction (and some other parts of the work e.g. l. 772-780), highlights the limitations of CCEs. Then Theorem 2 assumes access to a CCE of the 4-player game (which can be computed with a no-regret algorithm). While this CCE is for the modified (regularized) game and not the original one, why would this alleviate all the limitations that are inherent to CCEs in general and that you highlight in the paper? More specifically, the facts that the set of CCEs can be large, may have support on strictly dominated strategies and that stationary CCEs are PPAD-hard to compute in dynamic general sum games (l. 47-51). For the last point, does it mean that your output is also a nonstationary CCE since you also rely on no-regret algorithms (l. 8 of algorithm 1).

- Dependence on the action space size: a host of theoretical results in the MARL literature have devoted efforts to ‘break’ the ‘curse of multi-agency’ and go from the product of action spaces to the sum. Your sample complexity is exponential in the number of players. While I understand that space might have been constraining, at least a comment is needed here on the theorem. The very last point of the conclusion mentions scalability but this is not clearly delineating the limitations of the result and why.

- I have several other questions that I would like the authors to address for clarifying their contributions and the technical novelty.

Minor:
- l. 47-48: ‘the set of CCE can be large (introducing an additional problem of equilibrium selection)’: This equilibrium solution problem is already largely relevant for Nash equilibria, I find the argument confusing here. This is even more confusing since you also rely on CCE computation (for a modified game though) in the paper.
- l. 305-307: H notations do not seem to be used in the main part.
- l. 377, eq. (9): \tilde P V: the notation is a bit confusing here, I guess it is the usual dynamic programming notation but \tilde{P} \in \Delta_S and V \in R^S is a bit unclear, \tilde{P} is usually a matrix.
- (24) l. 1045: for the argmax to be unique I guess you need D_i to be strictly convex w.r.t. p_i here.
- Proposition 1 p. 20: bounded rationality does not seem to play any role in obtaining this result, this is probably worth mentioning.

**Questions:**

1. Could you please elaborate on how do you obtain figure 1? l. 69-71: ‘The markers … represent the necessary parameter values required to recreate the average strategy played by people in various 2-player games  in observational data…’
Could you please clarify this?  How do you find the values of the bounded rationality and risk aversion levels (or their ratio) only looking at behavioral data? Are you saying that you test several such ratios, computing the equilibria that you define in the risk averse and bounded rationality game (which ones? There are not a priori unique?) and then the deployment of the obtained equilibrium policy cpincides with the ‘average strategy played by people’ (whatever that exactly means). As Fig. 1 plays an important role in motivating your approach (at least as a motivating illustration for the games you consider), I would expect a more rigorous and precise description here in order to strengthen the message and clarify to.

2. Are both risk aversion formulations (2) and (3) related by Jensen’s inequality? Any implications?

3. Where is continuity w.r.t. both arguments of D needed? Does this exclude the examples of D considered in this work? It seems to exclude the ones considered in the appendix (because of continuity w.r.t. second argument that is likely not needed). While I understand that the space limitation is constraining, I think examples should appear in the main part.

4. Could you rather state your condition as $\epsilon_1 \epsilon_2 \geq \xi_1^* \xi_2^*$ to include zero values of the parameters? This comes from your specific choice of $\lambda$ that cancels one of the terms in your proof. Note that a symmetric choice for lambda is also possible (cancelling the other term). Otherwise the condition looks non-symmetric (I can set $\epsilon_1 = 0$ but not $\epsilon_2 = 0$, similarly for \xi_1^*, \xi_2^*) whereas the game is symmetric. Is positivity of all these parameters needed? Comments about the corner cases would be welcome here.

5. More generally, it seems that one of the key argument in the proof of the main result (theorem 7) (if not the only key argument) is the joint convexity invoked. Under which condition do CCEs and NEs coincide more generally? How does your result fit with such known existing results.

6. l. 373: You put a continuity assumption for matrix games (l. 207) and consider lower-semicontinuity in l. 373. Any specific reason for this discrepancy? Which ones of the examples you provide in table 1 satisfies continuity w.r.t. both arguments?

7. Definition 7: Why is there a deviation w.r.t. the horizon step and not just $\pi’_i, \pi_{-i}$? Is this standard? If I set $\epsilon_i = 0$ and remove the regularizers, do I recover the standard definition of NEs in finite horizon Markov games?

8. Theorem 3: The condition you obtain for RAMG is not a generalization of the matrix games (two-player) one. Does this show some fundamental difference between the two player and multiplayer (\geq 3) settings?

9. Eq. 19 p. 17: If you exclude the action a_i of player i from being subject to \rho_i, \pi_{-i}, I guess you would get a less stringent assumption than what you currently have. Is this right? Essentially, excluding k=i in (31).

10. Can you provide a proof of Corollary 7.1?

11. Proof of Theorem 4: Do the examples you provide satisfy the L lipschitzness of D wrt the l_1 norm for the 2nd argument, the first one being fixed?

Minor:
- You cite Tversky and Kahneman (1992) in l. 154-155: ‘a preponderance of empirical evidence suggests that people do not play their Nash strategies …’ To the best of my knowledge, there is no single occurrence of the word ‘Nash equilibrium’ in this paper. The reference is relevant  though for the risk aversion discussion at least for single agents and beyond as a reference in behavioral economics.
- \sigma is introduced as a probability distribution over the product space P \times \bar{P}, then you also use it only for $\pi$ s (l. 317), perhaps a slight abuse of notation here.
- I think Theorem 6 p. 18 should appear in the main part to make existence very clear. You report an existence result in theorem 1 for RNEs but you do not state it for the equilibrium which are the focus of the work. I understand that space imitation might have been the issue, but I think this is important. Suggestion: In that case maybe defer Theorem 1 to the appendix and move Theorem 6 to the main part?
- l. 392: ‘non-linear sum over’ is this a sum? The terminology is confusing here, is it rather a composition or a recursion?
- Please be precise in the statement of Theorem 4, this is a high probability result, the result holds with probability at least 1-\delta.
- l. 452-453: the sentence needs to be revised, something missing.
- l. 515 ‘and goal states for agents as well as goal states of agents’, meaning?
- l. 518: ‘To introduce multi-agent effects’, what do you mean exactly by these effects? Collisions? Are they possible in your implementation?
- l. 529: why would agent 1 wait? Does agent 2 make it riskier for agent 1 to move if they move together? Can you explain?
- l. 988: what do you mean by a strictly competitive game?
- p. 21, l. 1132- …: The 1/2 does not seem to be useful in the proof.
- p. 29: ‘to arrive at the CCE \sigma of the four player game’, is there a unique one?

---

> ### Author Response · Authors · 2024-11-24
> **Reply to Reviewer BGux: Part One**
>
> Before addressing the reviewer’s questions and suggestions, thank you for the thorough review and many constructive comments, which we feel have really helped us improve our presentation. When writing the paper we had to cut some exposition and move some to the appendix due to space constraints but you bring up many valid points that we will clarify and expand on in the revision. Below we address your individual comments:
>
> > **Q1:** My main concern about this paper is the following question: What exactly makes the introduced games tractable? The paper argues that it is the combination of bounded rationality and risk aversion. I find the focus on tractability rather confusing. Risk aversion has its own motivation but it is unclear if it is required here for tractability. To be precise, I think the contribution in this paper is that you can achieve tractability (for computing another solution concept than NE/CCE relating to a modified game) independently of the knowledge of payoff functions (their bounds I guess). More precisely, I think bounded rationality is already enough to allow for tractability but it should require some assumptions on the payoffs which this paper avoids. Please correct me if I have missed something here. The paper does not exactly present the contribution this way.
> > -- l. 111-112: ‘RQE not only incorporates features of human decision-making but are also more amenable to computation than QRE …’ Why? There is no extended discussion about this whereas I believe it is essential to clarify the contributions.
>     -- l. 235-238: While risk aversion might not be enough, I think bounded rationality should be enough (e.g. QRE as demonstrated in a number of works) under relevant assumptions on the structure of the game (mainly conditions on the bounded rationality level compared to the magnitude of the utilities/rewards). I find the presentation order confusing here. You start by introducing risk aversion and show its shortcomings as for the tractability of computing equilibria in such risk averse games to motivate the need of something else but equilibria should become tractable only with bounded rationality.
>     -- From the title and the abstract (and throughout the paper), the emphasis is on tractability, as if both are actually needed to ‘achieve’ tractability. How does this condition compare to the one for say QREs? What’s the price to pay and advantages (if any) for RQNEs compared to QREs (beyond the advantage of incorporating risk aversion)?
>
> **A1**: Your intuition and understanding of our contributions are highly accurate, and your suggestions on presentation are really helpful.  In the revision we will adjust the presentation order to first introduce bounded rationality and why it is not enough to esnure computational tractabiltiy (more on this below). We will emphasize and give intuition for the following point: **What makes these games computationally tractable? The combination of both risk-aversion and bounded rationality.**
>
> * **Bounded rationality alone is insufficient for computational tractability.** The reviewer has accurately explained this aspect. If we consider only bounded rationality (e.g., Quantal Response Equilibria, QREs), additional assumptions about the level of bounded rationality are required, which depend on the structure of the original game (e.g., the dimensionality of pure strategies and the magnitude of payoffs).
>
> For instance, in a simple two-player game where the agents aim to solve objectives such as $\min_{\pi_1} \pi_1^\top A \pi_2  - \epsilon H(\pi_1)$ and $\min_{\pi_2} \pi_1^\top B \pi_2  - \epsilon H(\pi_2)$ with d-dimensional pure strategies $\pi_1,\pi_2$, a sufficient condition for computational tractability would require the bounded rationality parameter **$\epsilon^2 \geq d^2 ||A+B||^2$**. This requires $\epsilon$ to be excessively large, which may not always be practical or realistic. This point is central to the clarification in lines 111–112 of our manuscript and we will expand on this extensively in the revision to drive home this point.
>
> * **The contribution of this work: achieving computational tractability independently of the original game structure.** In contrast, the condition for tractability of RQE only depends on the relative degrees of risk-aversion and bounded rationality **and not on the underlying game structure**. This allows one to choose these parameters independent of the game/task. This is achieved by the incorporation of risk in its dual form which allows the extended game to be essentially a nonlinear n-player zero-sum game (no matter the underlying payoffs). We will make this clearer as well.

---

> ### Author Response · Authors · 2024-11-24
> **Reply to Reviewer BGux: Part Two**
>
> > Q2 Equilibrium collapse is briefly mentioned in l. 294. I would like to see a more in-depth description of how the results of this paper connect to the prior work regarding this phenomenon. It seems that the extended game you introduce seems to have some zero sum structure under a perhaps more stringent assumption than what you set (\epsilon_1 = \xi_{2,1} and \epsilon_2 = \xi_{1,2}, a condition satisfying your condition), under which J_1 + J_2 + \bar{J}_1 + \bar{J}_2 = 0. This can be seen from setting \lambda = 1/2 in your proof of Theorem 7 p. 21, l. 1133-1146. Is this an observation that connects to prior work about zero sum games and equilibrium collapse? It is also worth mentioning that setting \lambda = 1/2 leads to a condition that is consistent with your extension to the n-player setting (l. 1226-1232).
>
> **A2**:
>
> You are again correct that equilibrium collapse is key to our results, which is a phenomenon that is known to occur in n-player zero-sum matrix games and certain Markov games. For example, (Cai et al, 2016) and (Kalogiannis and Panageas, 2023) show that the set of CCE collapses to the set of NE for zero-sum polymatrix normal-form and Markov games, resp. In our work the introduction of risk-aversion gives rise to such a zero-sum structure if a stringent assumption such as $\epsilon_1 = \xi_{2,1}$ and $\epsilon_2 = \xi_{1,2}$ holds. The zero-sum game however  is nonlinear due to the risk metrics. Thus, the zero-sum structure by itself is not enough to guarantee collapse and we require more convexity/concavity on the loss functions which we achive through the careful introduction of bounded rationality. In the two-player setting we allow for slight deviations away from an exact zero-sum structure by introducing the parameter $\lambda$ and we believe that the requirements for tractability can be relaxed in n-player games but presented the $\lambda=1/n$ case for simplicity. All told, we *are* making use of the fact that n-player zero-sum games are conducive to equilibrium collapse, but due to the nonlinearities in our games we generalize previous proofs.
>
> > **Q3:** I think the introduction and motivation of (6) and (7) needs some improvement. Could you give more motivation and intuition?
>
> **A3:**
> Motivation for the studied $2n$-player game (6,7) can be summarized as follows. Incorporating risk aversion into the normal-form games is challenging. As illustrated in Example 1 (l.977), we show that even in two-player zero-sum games, the risk-adjusted game loses any zero-sum structure and may cease to even be strictly competitive (one player's gain is the other's loss). However, by using the dual form of a risk metric we show that the risk-adjusted game can be strategically equivalent (in that the eq. coincide) to a $2n$-player game which has a almost $zero-sum$-like structure that we can exploit using bounded rationality. We plan to add this discussion to the main text (adjusted to page-limits) making it clearer.
>
> > **Q4:** The notations $\xi_1^*,\xi_2^*$ are quite confusing, especially that you also use the notations $\tau_1,\tau_2$ in Fig. 1 without defining them in the main part. What do you mean precisely by satisfy the conditions of Theorem 2 (l. 328)? How do the \tau s and \xi s relate?I believe one issue in the presentation is that you chose to hide \tau in the definition of the regularizer D without putting it in front of D (as it appears from the appendix). Maybe a solution is just to add the examples to the main part to clarify this and further comment on the relationship between the \xi s and \tau s.
>
> **A4:**
>
> The $\tau$'s are special instances developed in Cor.7.1-7.2 as $\xi^*=1/\tau$. Thank you for the editing suggestions. We plan to add the following informal corollary to the main text (adjusted to page-limits), pointing the Fig.1 to this statement making it clearer:
>
> **Corollary:** Suppose the players are risk-averse using some risk metric (for e.g., entropic risk, also see Table 1) with parameters $\tau_1$ and $\tau_2$ respectively. If they respond in the space of some associated quantal responses (for e.g., Kullback-Leibler divergence, also see Table 1) with parameters $\epsilon_1$ and $\epsilon_2$ respectively, then the players can compute a risk-averse QRE by using no-regret learning when the parameters satisfy $\epsilon_1 \tau_1 \ge \frac{1}{\epsilon_2\tau_2}.$

---

> > ### Author Response · Authors · 2024-11-24
> > **Reply to Reviewer BGux: Part Three**
> >
> > > **Q5:** Comments about experiments: While the paper focuses on RQREs, experiments do not seem to illustrate the concept. What are the RQREs in this game and why? Are they different from NEs? Why would they be more reasonable?
> >
> > **A5:**
> >
> > We emphasize that the Fig.1 markers *GHP: Game 4* and *SC: Game 1-12* represent the values of $\epsilon$'s and $\tau$ required such that the resulting RQE of the game is close to the average strategy played by people in various $2$-player games. In the original behavioral economics papers in which these games are investigated, they made people play the game repeatedly and then searched for parameter values/equilibrium notions that recreated the observed behaviors. These games were designed so that if people are risk-averse, they would have to deviate from Nash at equilibrium. This was indeed seen in the experimental data. Interesting, in the original paper GHP, the combination of risk aversion and bounded rationality yielded the best fits to the experimental data---a finding that is particularly pertinent to our paper.
> >
> > In our paper we perform a similar excercise as these papers, and fit the parameter values of $\epsilon$'s and $\tau$'s by trial and error to match the average strategy played by people in these games. We found that RQE could capture these behaviors.  To emphasize, the average strategy played by people and the NEs for the matching pennies games (described in l.1512-l.1518) are *different*.  We also highlight similar need for risk-averse solutions in the cliff-walking example that are necessarily different risk-neutral solutions. We will expand on these experiments in the revision.
> >
> > > **Q6:** The introduction (and some other parts of the work e.g. l. 772-780), highlights the limitations of CCEs. Then Theorem 2 assumes access to a CCE of the 4-player game (which can be computed with a no-regret algorithm). While this CCE is for the modified (regularized) game and not the original one, why would this alleviate all the limitations that are inherent to CCEs in general and that you highlight in the paper? More specifically, the facts that the set of CCEs can be large, may have support on strictly dominated strategies and that stationary CCEs are PPAD-hard to compute in dynamic general sum games (l. 47-51). For the last point, does it mean that your output is also a nonstationary CCE since you also rely on no-regret algorithms (l. 8 of algorithm 1).
> >
> >
> > **A6:** Thank you for pointing this out, we will clarify this in the revision. We do not aleviate any limitations of CCEs or encounter the issues since all the games that we solve have the approximate $n$-player zero-sum structure we discussed above. We prove that all CCE of the $2n$-player games coincide with their Nash eq. and that all of their Nash eq. must be RQE of the original game. We also believe that when the assumption on parameter values on $\epsilon$ and $\xi$ is strict (i.e the inequality is strict) that the Nash is unique. While we do not prove this, we have observed this to be true experimentally and think it can be proved by exploiting links between equilibrium-collapse and monotone games.

---

> > > ### Author Response · Authors · 2024-11-24
> > > **Reply to Reviewer BGux: Part Four**
> > >
> > > > **Q7:** Dependence on the action space size: a host of theoretical results in the MARL literature have devoted efforts to ‘break’ the ‘curse of multi-agency’ and go from the product of action spaces to the sum. Your sample complexity is exponential in the number of players. While I understand that space might have been constraining, at least a comment is needed here on the theorem. The very last point of the conclusion mentions scalability but this is not clearly delineating the limitations of the result and why.
> > >
> > >
> > > **A7:** We sincerely appreciate the reviewer’s insightful suggestions. We fully agree that breaking the curse of multiagency is one of the big challenges in multi-agent reinforcement learning (MARL). We will incorporate the following discussion into the final version:
> > >
> > > "Breaking the curse of multiagency in RAMGs is crucial to achieving algorithms with data scalability. This challenge is well-recognized in MARL, where significant efforts and advancements have been made [1–3]. However, for risk-sensitive MARL or MARL frameworks that address robustness against uncertainty, this issue remains largely understudied, with only one related work utilizing a different game formulation [4] to our knowledge. These scenarios pose additional challenges compared to standard Markov games, as they often result in highly nonlinear or non-closed-form utility functions for each agent (as in this work). Addressing these challenges requires new techniques to balance adversarial online learning and statistical error. Moreover, the solutions may need to be tailored to specific risk-averse metrics or utilize quantal response approaches."
> > >
> > >
> > > >> [1]Jin, C., Liu, Q., Wang, Y., and Yu, T. (2021). V-learning-a simple, efficient, decentralized algorithm for multiagent RL. arXiv preprint arXiv:2110.14555\
> > > >>[2]Cui, Q., Zhang, K., and Du, S. (2023). Breaking the curse of multiagents in a large state space: Rl in markov games with independent linear function approximation. In The Thirty Sixth Annual Conference on Learning Theory, \
> > > >>[3]Wang, Y., Liu, Q., Bai, Y., and Jin, C. (2023). Breaking the curse of multiagency: Provably efficient decentralized multi-agent rl with function approximation. In The Thirty Sixth Annual Conference on Learning Theory\
> > > >> [4] Shi, Laixi, et al. "Breaking the Curse of Multiagency in Robust Multi-Agent Reinforcement Learning." arXiv preprint arXiv:2409.20067 (2024).
> > >
> > >
> > > > **Q8:** l. 47-48: ‘the set of CCE can be large (introducing an additional problem of equilibrium selection)’: This equilibrium solution problem is already largely relevant for Nash equilibria, I find the argument confusing here. This is even more confusing since you also rely on CCE computation (for a modified game though) in the paper.
> > >
> > > **A8:**
> > > Thank you for raising this question. In lines 47–48, we introduce correlated equilibria (CCEs) as a different type of equilibrium compared to Nash equilibria (NEs). The key point is that the set of CCEs is even larger than the set of NEs, which can sometimes lead to poor performance.
> > >
> > > However, there is no contradiction here. While we acknowledge the drawbacks of CCEs, such as their large solution set and potential for poor outcomes, we leverage their computational tractability. Importantly, the general property of CCEs having a large solution set is not relevant to our results, as we focus on NEs for our formulated risk-adjusted games. Nonetheless, we take advantage of the computational tractability of CCEs, particularly because we demonstrate that in our games, NEs and CCEs are identical, so we can compute CCEs to get NEs.

---

> > > > ### Author Response · Authors · 2024-11-24
> > > > **Reply to Reviewer BGux: Part Five**
> > > >
> > > > > **Q9:** Suggestions for notations and presentations
> > > > > * l. 305-307: H notations do not seem to be used in the main part.
> > > > >* l. 377, eq. (9): \tilde P V: the notation is a bit confusing here, I guess it is the usual dynamic programming notation but \tilde{P} \in \Delta_S and V \in R^S is a bit unclear, \tilde{P} is usually a matrix.
> > > > >* $\sigma$ is introduced as a probability distribution over the product space $P \times \bar{P}$, then you also use it only for $\pi$ (l. 317), perhaps a slight abuse of notation here.
> > > > >* I think Theorem 6 p. 18 should appear in the main part to make existence very clear. You report an existence result in theorem 1 for RNEs but you do not state it for the equilibrium which are the focus of the work. I understand that space imitation might have been the issue, but I think this is important. Suggestion: In that case maybe defer Theorem 1 to the appendix and move Theorem 6 to the main part?
> > > > >* l. 392: ‘non-linear sum over’ is this a sum? The terminology is confusing here, is it rather a composition or a recursion?
> > > > >* Please be precise in the statement of Theorem 4, this is a high probability result, the result holds with probability at least 1-\delta.
> > > >
> > > >
> > > > **A9:**
> > > >
> > > > We will add $\tilde P V$ -- represents the inner product depicting expectation of $V$ w.r.t $P$ -- in the notation block.
> > > >
> > > > As Theorem 1 and 6 are meant to be the same, we will update this accordingly in our revision.
> > > >
> > > > We will fix other suggested typos. Thanks.
> > > >
> > > >
> > > > > **Q10:** (24) l. 1045: for the argmax to be unique I guess you need D_i to be strictly convex w.r.t. p_i here.
> > > >
> > > >
> > > > **A10:**
> > > >
> > > > As Proposition 1 p. 20 does not require uniqueness of equilibrium solutions, we pick any argmax in l.1045 as $p^*_i$. We will fix this typo in our revision as $p^*_i \in \arg\max (\ldots)$ in Proposition 1 and its proof.
> > > >
> > > >
> > > > > **Q11:** Proposition 1 p. 20: bounded rationality does not seem to play any role in obtaining this result, this is probably worth mentioning.
> > > >
> > > >
> > > > **A11:**
> > > >
> > > > Thank you for observing this. We will highlight it in our revision in conjunction with our response **A3**.
> > > >
> > > > > **Q12:** Could you please elaborate on how do you obtain figure 1? l. 69-71: ‘The markers … represent the necessary parameter values required to recreate the average strategy played by people in various 2-player games in observational data…’ Could you please clarify this? How do you find the values of the bounded rationality and risk aversion levels (or their ratio) only looking at behavioral data? Are you saying that you test several such ratios, computing the equilibria that you define in the risk averse and bounded rationality game (which ones? There are not a priori unique?) and then the deployment of the obtained equilibrium policy cpincides with the ‘average strategy played by people’ (whatever that exactly means). As Fig. 1 plays an important role in motivating your approach (at least as a motivating illustration for the games you consider), I would expect a more rigorous and precise description here in order to strengthen the message and clarify to.
> > > >
> > > > **A12:**
> > > >
> > > > We refer to our response **A5** and l.1520-1528 for the implementation details (in addition to the shared anonymous code). We emphasize Fig.1 markers *GHP: Game 4* and *SC: Game 1-12* represent the average strategy played by people in various $2$-player games. The average strategy played by people and the NEs for the designed matching pennies game (described in l.1512-l.1518) are *different*. Our algorithm (gradient descent algorithm described in l.1520-1528) reaches different solutions for different $\epsilon$ and $\tau$ parameters. We arrive at the people strategies of the games via hyperparameter search for $\epsilon$ and $\tau$ parameters. Hence, the deployed equilibrium policy (for corresponding $\epsilon$ and $\tau$ axes in Fig.1) coincides with the average strategy played by people in these games. We observe that these parameters satisfy our Theorem 2 (or Corollary 7.1-7.2) conditions (this condition is depicted as the shaded region in Fig.1). We plan to add this discussion to the main text (adjusted to page-limits) making it clearer.
> > > >
> > > > > **Q13:**  Are both risk aversion formulations (2) and (3) related by Jensen’s inequality? Any implications?
> > > >
> > > > **A13:**  The reviewer is correct that the two risk-averse formulation in (2) and (3) can be related with Jensen's inequality, since $\rho_{i,\pi_{-i}}(\cdot)$ is a convex function. So using the same $\rho_{i,\pi_{-i}}(\cdot)$，the Aggregate Risk Aversion formulation (2) is more conservative (always not bigger) than Action-dependent Risk Aversion formulation (3). The reason is that the Aggregate Risk Aversion formulation in (2) considering the regularized worst-case when the agent i let the 'adversarial' know its strategy when doing the attacking. This additional information can help the 'adversarial' to choose a harder case for agent i than the latter formulation in (3) where the 'adversarial' won't know agent i's policy.

---

> > > > > ### Author Response · Authors · 2024-11-24
> > > > > **Reply to Reviewer BGux: Part Six**
> > > > >
> > > > > > **Q14:**
> > > > > > * Where is continuity w.r.t. both arguments of D needed? Does this exclude the examples of D considered in this work? It seems to exclude the ones considered in the appendix (because of continuity w.r.t. second argument that is likely not needed). While I understand that the space limitation is constraining, I think examples should appear in the main part.
> > > > > > * l. 373: You put a continuity assumption for matrix games (l. 207) and consider lower-semicontinuity in l. 373. Any specific reason for this discrepancy? Which ones of the examples you provide in table 1 satisfies continuity w.r.t. both arguments?
> > > > >
> > > > > **A14:**
> > > > >
> > > > > Thanks for raising this insight. Yes, for Theorem 6, we do indeed require continuity of $D$ w.r.t. both arguments to invoke convex game (Rosen, 1965) Nash equilibrium existence. The examples provided in Table.1 are lower-semi-continuous functions with respect to both arguments in general cases (for e.g. f-divergence [1], shortfall risk measure [2], mean-upper-semideviation [2], expectiles [2], quantiles [2]). We emphasize that each player i has access to a (finite) action set with $A_i$ decisions. As our decision space of $n$-player game for Theorem 6 is finite and discrete (so, every point has a neighborhood containing only itself), the examples provided in Table.1 are furthermore continuous w.r.t. both arguments. We will add this insight in our revision.
> > > > >
> > > > > > [1] Ambrosio, Luigi, Nicola Fusco, and Diego Pallara. Functions of bounded variation and free discontinuity problems. Courier Corporation, 2000.
> > > > > > [2] A. Shapiro, D. Dentcheva and A.Ruszczynski, Lectures on stochastic programming: modeling and theory, MPS-SIAM series on optimization, 2009.
> > > > >
> > > > >
> > > > > > **Q15:** Could you rather state your condition as $\epsilon_1 \epsilon_2 \geq \xi_1^* \xi_2^*$ to include zero values of the parameters? This comes from your specific choice of $\lambda$ that cancels one of the terms in your proof. Note that a symmetric choice for lambda is also possible (cancelling the other term). Otherwise the condition looks non-symmetric (I can set $\epsilon_1 =0$ but not  $\epsilon_2 =0$, similarly for $\xi_1^*, \xi_2^*)$ whereas the game is symmetric. Is positivity of all these parameters needed? Comments about the corner cases would be welcome here.
> > > > >
> > > > > **A15:**
> > > > > Thanks for raising this interesting questions. There are two insights for setting $\epsilon_1, \epsilon_2$.
> > > > > * **Whether $\epsilon_1,\epsilon_2$ can be zero?** No, we need $\epsilon_1>0, \epsilon_2 >0$. Since by definition, $\xi_1^* >0, \xi_2^*>0$ (see line 305-307), so $\epsilon_1\epsilon_2 \geq \xi_1^* \xi_2^*>0$. By observing Figure 1, we can see that  $\epsilon=0$ only yeilds tractability when $\xi$ is 0 which occurs when the player is so risk averse that they are entirely adversarially robust. We will add discussion of this to the paper.
> > > > > * **$\epsilon_1, \epsilon_2$ are prescribed before determining $\lambda$.** We want to recall that $\epsilon_1, \epsilon_2$ are actually fixed when we specify the risk-adjusted games as the level of bounded rationality. So they can be chosen as non-symmatric according to people's own preferences.
> > > > >
> > > > >
> > > > >
> > > > >
> > > > >
> > > > >
> > > > > > **Q16:** More generally, it seems that one of the key argument in the proof of the main result (theorem 7) (if not the only key argument) is the joint convexity invoked. Under which condition do CCEs and NEs coincide more generally? How does your result fit with such known existing results.
> > > > >
> > > > > **A16:**  Thanks for raising this constructive comments on the technical contribution. The reviewer is correct that Theorem 7 is one of the critical technical arguments. The key of the proof is that we show the coincidence between the CCEs and NEs of an auxiliary game that is constructed for proof.
> > > > > The technical contributions compared to the prior arts (also show coincidence between CCEs and NEs) are as follows:
> > > > > * **When considering any n-player general sum game, we construct an auxiliary 2n-player _approximate_ _zero-sum_ auxiliary game for proof.** The approach is indeed inspired by prior works that a class of zero-sum polymatrix games have the equilibrium collaps --- CCEs and NEs are identical [1]. While in this paper, we consider risk-adjusted games for n-player (equation (5)), which is a general-sum games. So to resort to the prior technical, we construct an auxiliary new 2n-player approximate zero-sum game by introducing an adversarial for each original player.
> > > > > * **Our risk-adjusted game have different structure that need tailored proof to get to this coincidence.** Regarding the auxiliary new approximate zero-sum game, we also need to proof specifically according to its structure since we are not exactly zero-sum and can't apply the prior arts [1] for zero-sum polymatrix game directly.
> > > > >
> > > > > >> [1] Kalogiannis, Fivos, and Ioannis Panageas. "Zero-sum polymatrix markov games: Equilibrium collapse and efficient computation of nash equilibria." Advances in Neural Information Processing Systems 36 (2024).

---

> > > > > > ### Author Response · Authors · 2024-11-24
> > > > > > **Reply to Reviewer BGux: Part Seven**
> > > > > >
> > > > > > > **Q17:** Definition 7: Why is there a deviation w.r.t. the horizon step and not just $\pi_i, \pi_{-i}$? Is that standard? If I set $\epsilon_i = 0$ and remove the regularizers, do I recover the standard definition of NEs in finite horizon Markov games?
> > > > > >
> > > > > > **A17:**
> > > > > > Thanks for raising this insightful question. Yes, it is a standard definition for finite-horizon Markov games --- If we remove all the regularizers (both $D_{env,i}, D_{pol,i}$ and $\nu_i$), we will reduce to finite-horizon Markov games. The deviation is w.r.t. the horizon and also $\pi_{i,h},\pi_{-i,h}$ (the policy at time step $h$) is due to, at each time step $h$, we need to find the RQE of the matrix game at time step $h$, by finding agents'policy $\pi_{i,h},\pi_{-i,h}$ at this time step $h$ (independent from other steps' policy).
> > > > > >
> > > > > >
> > > > > > > **Q18:** Theorem 3: The condition you obtain for RAMG is not a generalization of the matrix games (two-player) one. Does this show some fundamental difference between the two player and multiplayer (\geq 3) settings?
> > > > > >
> > > > > > **A18:**
> > > > > >
> > > > > > The reviewer is correct at this finding. The 2-player condition is less stringent compared to the $n$-player condition. The weighted combination proof technique we used in l.1129 (for 2-player setting) cannot be directly extended to the $n$-player setting due to combinatorial play between all $n$ players. This is an interesting open question to be addressed in future directions. Please also see our response **A2**.
> > > > > >
> > > > > > > **Q19:** Eq. 19 p. 17: If you exclude the action a_i of player i from being subject to \rho_i, \pi_{-i}, I guess you would get a less stringent assumption than what you currently have. Is this right? Essentially, excluding k=i in (31).
> > > > > >
> > > > > > **A19:**
> > > > > >
> > > > > > Thanks for this observation. It will be interesting to investigate this in future works. As of now, excluding $k=i$ in (31) will break the necessary zero-sum structure evident in l.1321-1334.
> > > > > >
> > > > > > > **Q20:** Can you provide a proof of Corollary 7.1?
> > > > > >
> > > > > > **A20:**
> > > > > >
> > > > > > The proof is similar as in in l.1188 p.23 by noting the function $H(p,\pi)=\frac{1}{\tau}KL(p,\pi)-\xi \nu(\pi)$ is concave in $\pi$ for all $p$ if $\xi\ge\frac{1}{\tau}$ and the log-barrier regularizer $\nu$. We will add this in our revised appendix. Thanks.
> > > > > >
> > > > > >
> > > > > > > **Q21:** Proof of Theorem 4: Do the examples you provide satisfy the L lipschitzness of D wrt the l_1 norm for the 2nd argument, the first one being fixed?
> > > > > >
> > > > > >
> > > > > > **A21:**
> > > > > > Thanks for raising this question. Yes, we have provided examples for $L$-Lipschitz $D_{env,i}$ in line 482-484. Namely, we can choose $L_p$ norm for any $p\geq 1$, including widely used total variation (TV) when people considering environmental shift in reinforcement learning [1].
> > > > > >
> > > > > > >> [1] Pan, Yuxin, Yize Chen, and Fangzhen Lin. "Adjustable robust reinforcement learning for online 3d bin packing." Advances in Neural Information Processing Systems 36 (2023): 51926-51954.
> > > > > >
> > > > > >
> > > > > > > **Q22:** You cite Tversky and Kahneman (1992) in l. 154-155: ‘a preponderance of empirical evidence suggests that people do not play their Nash strategies …’ To the best of my knowledge, there is no single occurrence of the word ‘Nash equilibrium’ in this paper. The reference is relevant though for the risk aversion discussion at least for single agents and beyond as a reference in behavioral economics.
> > > > > >
> > > > > >
> > > > > > **A22:**
> > > > > >
> > > > > > Thanks for noting this. We included Luce (1925) and Tversky and Kahneman (1992) as a reference to back up the fact that people deviate from rational decision-making by bying risk averse or boundedly rational. In the papers and indeed their more general lines of work, they only studied individual decisions and not games.  Goeree et al. (2003); McKelvey and Palfrey (1995)) extend these findings to games. We are happy to make the statement more precise.
> > > > > >
> > > > > > > **Q23:** Other quetions:
> > > > > > > * l. 515 ‘and goal states for agents as well as goal states of agents’, meaning?
> > > > > > > * l. 518: ‘To introduce multi-agent effects’, what do you mean exactly by these effects? Collisions? Are they possible in your implementation?
> > > > > > > * l. 988: what do you mean by a strictly competitive game?
> > > > > > > * p. 21, l. 1132- …: The 1/2 does not seem to be useful in the proof.
> > > > > > > * p. 29: ‘to arrive at the CCE \sigma of the four player game’, is there a unique one?
> > > > > >
> > > > > > **A23:**
> > > > > >
> > > > > > Thanks for these other questions. We will revise our manuscript accordingly:
> > > > > > * l.515: this is a typo where we mean to say ‘and goal states for agents'
> > > > > > * l.518: this was incorporated to arrive at risk-averse solutions -- there may be other possible ways to do this -- as we have described in Fig.2 results.
> > > > > > * l.988: In these examples we mean it no longer holds players' are competitive and a possibility of preferring unilateral deviation solutions.
> > > > > > * p. 21, l.1132: Thanks for this suggestion. It indeed is not useful; we used it for maintaining magnitudes (e.g. $J_1 + \bar{J}_1$) which is unnecessary.
> > > > > > * p. 29: You are right; this is a typo. Thanks.

---

> > > > > > > ### Author Response · Authors · 2024-11-26
> > > > > > >
> > > > > > > Dear Reviewer BGux,
> > > > > > >
> > > > > > > We extend our gratitude for dedicating your valuable time to reviewing our paper. Your insightful suggestions have greatly contributed to enhancing the quality of our work, and we eagerly anticipate the possibility of receiving further feedback from you. **As the reviewers-authors discussion period has been graciously extended by the ICLR committee (Dec 2/3 AoE), we kindly request your reply to our responses to ensure that all your concerns have been adequately addressed in our initial rebuttal response.** Moreover, we remain enthusiastic about engaging in further discussions to address any additional questions or considerations you may have.
> > > > > > >
> > > > > > > Best regards,
> > > > > > >
> > > > > > > Authors

---

> > > > > > > > ### Comment · Reviewer_BGux · 2024-12-01
> > > > > > > > **Thank you for your response and follow-up**
> > > > > > > >
> > > > > > > > Thank you for your detailed response to all my questions. I still have a clarification question before updating my score (I apologize for the delayed answer and follow-up given the upcoming deadline for authors though).
> > > > > > > >
> > > > > > > > The advantage of RQE (proposed equilibrium notion) over the standard QRE in terms of computational tractability is yet not clear to me and this relates to the main contribution of the paper even from the title (at least as it is presented now throughout the paper). I am not questioning the correctness of the result here (which I have checked and I do not see any issue to the best of my knowledge) and I think that the paper brings some interesting novelty but I think this point is worth clarifying given the central motivation of the paper.
> > > > > > > >
> > > > > > > > QREs should always exist under mild assumptions independently of the values of the payoff and the level of bounded rationality if I am not mistaken (by an application of the fixed point theorem).  Similarly RQEs do exist.
> > > > > > > > Now when it comes to computation:
> > > > > > > > - Can you elaborate more on why QREs would only be computationally tractable under assumptions on the level of bounded rationality involving the payoffs? Any reference for the result you mention in the rebuttal? What does it state exactly if the level of bounded rationality is larger than the quantity that you mention? How can we compute them efficiently?
> > > > > > > > - Are you claiming that equilibrium collapse also occurs for QREs but only under a condition that involves conditions on the level of bounded rationality (i.e. regularization of the game) and payoffs, maybe using similar arguments as yours? In this case, I think this result should appear in the paper and should be discussed to clarify the contribution of the paper, or at least a comment about it should be welcome. What do we know about computational tractability of QREs for general normal-form games? There are results for zero-sum games for instance (e.g. Cen et al. 2024. Fast Policy Extragradient Methods for Competitive Games with Entropy Regularization) but I don't know about results for general sum normal form games which are the focus of this work.
> > > > > > > > - This might not be known and this is fine but since the paper argues that they introduce both bounded rationality and risk aversion to achieve tractability because risk aversion is not enough, I think it is important to explain precisely why bounded rationality might or not be enough as well.
> > > > > > > >
> > > > > > > > As for the computation, even for RQEs (which you define), your algorithm requires running a no-regret learning algorithm and the step size of this algorithm will depend on the payoffs and the levels of bounded rationality and risk aversion (as the step size of such no-regret algorithms do usually depend on the bounds on the payoff values, some of the papers assume that the payoffs are bounded by 1 which hides this dependence). This is not the central question about computational tractability though.
> > > > > > > >
> > > > > > > >
> > > > > > > > **Minor:**
> > > > > > > >
> > > > > > > > **Q8:** Thank you for the response. l. 47-48: ‘the set of CCE can be large (introducing an additional problem of equilibrium selection)’: I just want to emphasize here that the formulation can be improved. As it is written, it might give the impression that the fact that the set of CCEs can be large does introduce an additional problem of equilibrium selection. This problem is already important for NEs in general, so the selection problem is not specific to CCE. The fact that the set of CCEs can be much larger than the set of NEs might make the selection problem even worse which is maybe what you wanted to say. This is not that clear from the formulation and that was one of my points related to that comment. As for the rest of the comment about CCE collapse and so on, I agree with your response.
> > > > > > > >
> > > > > > > > **Q14:** The decision space is the set of joint policies which you define in l. 140.  I don’t think the following is a solution to bypass continuity: ‘As our decision space of $n$-player game for Theorem 6 is finite and discrete (so, every point has a neighborhood containing only itself), the examples provided in Table.1 are furthermore continuous w.r.t. both arguments.’ Otherwise you are not even considering mixed strategies, existence of NEs using Rosen (1965) is based on Kakutani’s fixed point and existence of such a point has no reason to lie at the corners of the simplex. Maybe you just need to exclude the boundary where some risk measures (in your examples of Table 1) might not be defined on the entire simplex for both variables of the risk measure (e.g. entropic risk with KL w.r.t. its second variable).  I think this (secondary) technical point might be better addressed (sometimes you use continuity, sometimes lower-semicontinuity, it is not very clear to me). What is clear is that you use Rosen’s result that requires continuity w.r.t. both variables as you mention it in your response.

---

> > > > > > > > > ### Author Response · Authors · 2024-12-01
> > > > > > > > > **Further Follow-Up**
> > > > > > > > >
> > > > > > > > > Thank you for your response and for continuing to engage! Your point on making clear why bounded rationality is not enough for the kind of computational tractability result we want is a very good one and we will emphasize this in the revision.
> > > > > > > > >
> > > > > > > > > **Questions about QRE**
> > > > > > > > >
> > > > > > > > >  To answer your central question on the tractability of QREs, I think the most succinct explanation is that---to our understanding--- there is no similar result for QREs showing that **the same** class of QRE can be computed in all 2-player games or n-player games using a given algorithm or class of algorithms.
> > > > > > > > >
> > > > > > > > > The only condition that we know of or are able to derive for QRE is the one provided in our previous response which is the result of a straightforward calculation of the amount of regularization required to turn an arbitrarity 2-player matrix game into a monotone game (or strictly diagonally concave game a la Rosen) through regularization.
> > > > > > > > >
> > > > > > > > > This is done by giving sufficient conditions under which the game Jacobian becomes positive definite on the simplex. Given the monotone game you could compute the QRE using no-regret algorithms or extragradient/optimistic gradient methods (see e.g., Tight Last-Iterate Convergence Rates for No-Regret Learning in Multiplayer-Games, Golowich et al 2020).  We are happy to incorporate this result into our paper to highlight the differences between RQE and QRE.
> > > > > > > > >
> > > > > > > > > Without sufficient regularization the game does not have enough structure to allow computation of the QRE using no-regret learning to the best of our knowledge, and one could have cycles or chaos emerge in their learning dynamics (see e.g., Poincare Recurrance, Cycles, and Spurious Equilibria...., Flokas et al. 2019).
> > > > > > > > >
> > > > > > > > > To the best of my knowledge, we do not know of a general result for equilibrium collapse to occur for QREs. The structure that allows for equilibrium collapse in our transformed games stems in large part from the introduction of risk aversion.
> > > > > > > > >
> > > > > > > > > **Question on Computation**
> > > > > > > > >
> > > > > > > > > You are completely correct that many algorithms rely on parameter values that implicitly depend on game-dependent properties. However, one could also choose diminishing stepsizes and still retain their convergence properties (albeit asymptotically or at less fast rates).
> > > > > > > > > An interesting side-note is that because of risk-aversion one cannot simply normalize payoffs to a constant since relative magnitudes are well known to play a role in risk-aversion.
> > > > > > > > >
> > > > > > > > >
> > > > > > > > > **Other Comments**
> > > > > > > > >
> > > > > > > > > Q8: Thanks for this, that is the point we are trying to make and we are happy to clarify this.
> > > > > > > > > Q14: Thanks for being careful with this. The only condition we use in our paper is continuity of the penality metrics---since we rely on Rosen's result for existence.

---

> > > > > > > > > > ### Comment · Reviewer_BGux · 2024-12-01
> > > > > > > > > > **Thank you**
> > > > > > > > > >
> > > > > > > > > > Thank your for your answer that clarifies my concerns, I am increasing my score, I think this paper provides some interesting insights. The limitations of RQEs are not really discussed compared to existing solution concepts but that might be for future work. I believe that the paper might benefit from incorporating this discussion about QRE/RQE and adjusting the presentation accordingly along the lines of the rebuttal of the authors and the discussion.

---

> > > > > > > > > > > ### Author Response · Authors · 2024-12-01
> > > > > > > > > > >
> > > > > > > > > > > Dear Reviewer BGux,
> > > > > > > > > > >
> > > > > > > > > > > Thanks so much for your efforts in reviewing our paper and your positive score/feedback! We will further improve our paper following your suggestions during revision.
> > > > > > > > > > >
> > > > > > > > > > > Thanks,
> > > > > > > > > > >
> > > > > > > > > > > Authors

---

### Author Response · Authors · 2024-11-24
**General response to all reviewers**

We thank all the reviewers for their careful reading of the paper and their insightful and valuable feedback and suggestions. Below we provide a general response to highlight the technical contributions of this work and discuss individual points below.

**Key Contributions**
Our goal in this work was to understand the impact of introducing features of human decision-making from behavioral economics into problems of learning in games and multi-agent reinforcement learning. Surprisingly, we were able to show that the **combination** of bounded rationality and risk-aversion gives rise to a single class of equilibria that are computationally tractable across all n-player matrix games and finite-horizon Markov games.

Crucially this result requires both bounded rationality and risk aversion to go through. Risk aversion on its own gives rise to a convex game but cannot guarantee computational tractabiltiy of the resulting equilibrium notion (risk-averse Nash equilibria). Bounded rationality on its own is well known to help with computational tractability (as is pointed out by reviewer BGUx), but the degree of bounded rationality required to ensure this holds depends on the game structure (i.e., the relationship between players' payoffs and their magnitude). In contrast, the class of RQE we prove is computationally tractable is completely independent of the game structure. Furthermore, this class is expressive in that it can recreate peoples' patterns of play in experimental data as observed in Figure 1.


These two benefits distinguish RQEs from previous other equilibrium notions such as Nash equlibrium and Quantal Response Equilibriums (only consider bounded rationality), offering a fresh perspective on tractable solutions for learning games. **To the best of our knowledge, this is the first result of its kind and while we only scratch the surface of their analysis we are excited at the protential that these equilibria represent for designing principled algorithms for multi-agent reinforcement learning.**

---

### Meta-Review · Area_Chair_obKh · 2024-12-20

**Metareview:**

This paper incorporates elements and concepts from behavioral economics - namely risk aversion and bounded rationality - into multi-agent reinforcement learning (MARL). In particular, the authors define a class of risk-averse quantal response equilibria (RQE) which, under certain adjustments, are no-regret learnable in both $n$ -player matrix games and finite-horizon Markov games. Importantly, the tractability of RQE is determined by the agents' degree of risk aversion and bounded rationality rather than the underlying game structure. Finally, the authors provide an analysis of the sample complexity for computing these equilibria in finite-horizon Markov games assuming access to a generative/simulation oracle.

This paper was discussed extensively with the reviewers, and Reviewer BGux in particular provided a real service to the authors in this regard. Overall, the reviewers appreciated the paper's innovative integration of behavioral economics into multi-agent reinforcement learning, and they appreciated the fresh perspective that it brings to the field. Some concerns were raised regarding the paper's assumptions on the agents' risk aversion and bounded rationality levels, and their generalizability across different game settings but, for the most part, these were addressed convincingly in the authors' rebuttal.

As pointed out by Reviewer BGux, one particular point that merits further discussion is why bounded rationality without additional assumptions is not enough for computational tractability: the authors' model incorporates both risk aversion *and* bounded rationality, so insisting only on why the former is an obstacle to tractability seems somewhat incomplete. The authors committed on updating this part as well as some other (more minor) points in the paper to improve the presentation and delineate the limitations of their results.

Modulo the above, the consensus among reviewers was that the paper provides a fresh and solid contribution to the field of multi-agent reinforcement learning, and unanimously recommended acceptance.

**Additional Comments On Reviewer Discussion:**

To conform to ICLR policy, I am repeating here the relevant part of the metareview considering the reviewer discussion.

> This paper was discussed extensively with the reviewers, and Reviewer BGux in particular provided a real service to the authors in this regard. Overall, the reviewers appreciated the paper's innovative integration of behavioral economics into multi-agent reinforcement learning, and they appreciated the fresh perspective that it brings to the field. Some concerns were raised regarding the paper's assumptions on the agents' risk aversion and bounded rationality levels, and their generalizability across different game settings but, for the most part, these were addressed convincingly in the authors' rebuttal.
>
> Reviewer BGux pointed out the following issue that merits further discussion: why does bounded rationality without additional assumptions is enough for computational tractability? The authors' model incorporates both risk aversion *and* bounded rationality, so insisting only on why the former is an obstacle to tractability seems somewhat incomplete. The authors committed on updating this part as well as some other (more minor) points in the paper to improve the presentation and delineate the limitations of their results.

---

### Decision · Program_Chairs · 2025-01-22

Accept (Oral)